# The genomes of polyextremophilic cyanidiales contain 1% horizontally transferred genes with diverse adaptive functions

Alessandro W Rossoni[1], Dana C Price[2], Mark Seger[3], Dagmar Lyska[1], Peter Lammers[3], Debashish Bhattacharya[4], Andreas PM Weber[1]*

[1]Institute of Plant Biochemistry, Cluster of Excellence on Plant Sciences (CEPLAS), Heinrich Heine University, Düsseldorf, Germany; [2]Department of Plant Biology, Rutgers University, New Brunswick, United States; [3]Arizona Center for Algae Technology and Innovation, Arizona State University, Mesa, United States; [4]Department of Biochemistry and Microbiology, Rutgers University, New Brunswick, United States

**Abstract** The role and extent of horizontal gene transfer (HGT) in eukaryotes are hotly disputed topics that impact our understanding of the origin of metabolic processes and the role of organelles in cellular evolution. We addressed this issue by analyzing 10 novel Cyanidiales genomes and determined that 1% of their gene inventory is HGT-derived. Numerous HGT candidates share a close phylogenetic relationship with prokaryotes that live in similar habitats as the Cyanidiales and encode functions related to polyextremophily. HGT candidates differ from native genes in GC-content, number of splice sites, and gene expression. HGT candidates are more prone to loss, which may explain the absence of a eukaryotic pan-genome. Therefore, the lack of a pan-genome and cumulative effects fail to provide substantive arguments against our hypothesis of recurring HGT followed by differential loss in eukaryotes. The maintenance of 1% HGTs, even under selection for genome reduction, underlines the importance of non-endosymbiosis related foreign gene acquisition.

DOI: https://doi.org/10.7554/eLife.45017.001

*For correspondence:
andreas.weber@uni-duesseldorf.de

**Competing interests:** The authors declare that no competing interests exist.

## Introduction

Eukaryotes transmit their nuclear and organellar genomes from one generation to the next in a vertical manner. As such, eukaryotic evolution is primarily driven by the accumulation, divergence (e.g., due to mutation, insertion, duplication), fixation, and loss of gene variants over time. In contrast, horizontal (also referred to as lateral) gene transfer (HGT) is the inter- and intraspecific transmission of genes from parents to their offspring. HGT in Bacteria (*Doolittle, 1999*; *Ochman et al., 2000*; *Boucher et al., 2003*) and Archaea (*Nelson-Sathi et al., 2012*) is widely accepted and recognized as an important driver of evolution leading to the formation of pan-genomes (*Tettelin et al., 2005*; *Vernikos et al., 2015*). A pan-genome comprises all genes shared by any defined phylogenetic clade and includes the so-called core (shared) genes associated with central metabolic processes, dispensable genes present in a subset of lineages often associated with the origin of adaptive traits, and lineage-specific genes (*Vernikos et al., 2015*). This phenomenon is so pervasive that it has been questioned whether prokaryotic genealogies can be reconstructed with any confidence using standard phylogenetic methods (*Philippe and Douady, 2003*; *Doolittle and Brunet, 2016*). In contrast, as eukaryote genome sequencing has advanced, an increasing body of data has pointed towards

| Species | Origin | Country | Habitat | Habitat pH | Habitat Temp (°C) | Source |
|---|---|---|---|---|---|---|
| *C. merolae* 10D* | Sardinia | Italy | Acidic Hot Spring | 1.5 | Up to 45°C | ATCC®, T. Kuroiwa |
| *C. merolae* Soos | Soos National Park | CZ | Diatom field | 0.8 - 2 | < 0° - 30°C | W. Gross, M. Seger |
| *G. phlegrea* DBV009* | Nepi | Italy | Sulphuar Spring | 0.8 | 12°C | G. Pinto, ACUF |
| *G. phlegrea* Soos | Soos National Park | CZ | Diatom field | 0.8 - 2 | < 0° - 30°C | W. Gross, M. Seger |
| *G. sulphuraria* 002 (S) | La Solfatara | Italy | na | 1 | 36°C | G. Pinto |
| *G. sulphuraria* 074W* | Mount Lawu | Indonesia | Fumaroles | na | 35°C | W. Gross, P. De Luca |
| *G. sulphuraria* 5572 | Norris Basin, YNP | USA | Acidic soil | 1 | 55°C | M. Seger, R. W. Castenholz |
| *G. sulphuraria* Azora | Azores | Portugal | Porous sandstone, endolithic | 2.1 | na | W. Gross, A. Flechner |
| *G. sulphuraria* MS1 | Contaminant | USA | Ronust contaminant of YNP cultures | na | na | M. Seger, P. Lammers |
| *G. sulphuraria* MtSh | Mount Shasta | USA | Soil, close to mountain peak (4300m) | 2.2 | na | W. Gross, R. R. Pausewein |
| *G. sulphuraria* RT22 | Rio Tinto, Berrocal | Spain | Riverbank, endolithic | 2.5 | na | W. Gross, R. R. Pausewein |
| *G. sulphuraria* SAG21.92 | Yangmingshan | Taiwan | Hot spring | na | na | J. T. |
| *G. sulphuraria* YNP5578.1 | Nymph Creek, YNP | USA | Acid stream | 3 | 42°C | M. Seger, R. W. Castenholz |

**Figure 1.** Geographic origin and habitat description of the analyzed Cyanidiales strains. Available reference genomes are marked with an asterisk (*), whereas 'na' indicates missing information.

DOI: https://doi.org/10.7554/eLife.45017.002

the existence of HGT in these taxa, but at much lower rates than in prokaryotes (*Danchin, 2016*). The frequency and impact of eukaryotic HGT outside the context of endosymbiosis and pathogenicity however, remain hotly debated topics in evolutionary biology. Opinions range from the existence of ubiquitous and regular occurrence of eukaryotic HGT (*Husnik and McCutcheon, 2018*) to the almost complete dismissal of any eukaryotic HGT outside the context of endosymbiosis as being Lamarckian, thus false, and resulting from analysis artefacts (*Martin, 2018*; *Martin, 2017*). HGT skeptics favor the alternative hypothesis of differential loss (DL) to explain the current data. DL imposes strict vertical inheritance (eukaryotic origin) on all genes outside the context of pathogenicity and endosymbiosis, including putative HGTs. Therefore, all extant genes have their root in LECA, the last eukaryotic common ancestor. Patchy gene distributions are the result of multiple ancient paralogs in LECA that have been lost over time in some eukaryotic lineages but retained in others. Under this view, there is no eukaryotic pan-genome, there are no cumulative effects (e.g., the evolution of eukaryotic gene structures and accrual of divergence over time), and therefore, mechanisms for the uptake and integration of foreign DNA in eukaryotes are unnecessary.

A comprehensive analysis of the frequency of eukaryotic HGT was recently done by *Ku and Martin (2016)*. These authors reported the absence of eukaryotic HGT candidates sharing over 70% protein identity with their putative non-eukaryotic donors (for very recent HGTs, this figure could be as high as 100%). Furthermore, no continuous sequence identity distribution was detected for HGT candidates across eukaryotes and the 'the 70% rule' was proposed ('*Coding sequences in eukaryotic genomes that share more than 70% amino acid sequence identity to prokaryotic homologs are most likely assembly or annotation artifacts.*') (*Ku and Martin, 2016*). However, as noted by others (*Richards and Monier, 2016*; *Leger, 2018*), this result was obtained by categorically dismissing all

eukaryotic HGT singletons located within non-eukaryotic branches as assembly/annotation artefacts, as well as those remaining that exceeded the 70% threshold. In addition, all genes that were presumed to be of organellar origin were excluded from the analysis, leaving a small dataset extracted from already under-sampled eukaryotic genomes.

Given these uncertainties, the aim of our work was to systematically analyze eukaryotic HGT using the Cyanidiales (known as Cyanidiophytina in some taxonomic schemes) as model organisms. The Cyanidiales comprise a monophyletic clade of polyextremophilic, unicellular red algae (Rhodophyta) that thrive in acidic and thermal habitats worldwide (e.g., volcanoes, geysers, acid mining sites, acid rivers, urban wastewaters, geothermal plants) (*Castenholz and McDermott, 2010*). With a divergence age estimated to be around 1.92–1.37 billion years (*Yoon et al., 2004*), the Cyanidiales are the earliest split within Rhodophyta and define one of the oldest surviving eukaryotic lineages. They are located near the root of the supergroup Archaeplastida, whose ancestor underwent the primary plastid endosymbiosis with a cyanobacterium that established photosynthesis in eukaryotes (*Reyes-Prieto et al., 2007*; *Price et al., 2012*). In the context of HGT, the Cyanidiales became more broadly known after publication of the genome sequences of *Cyanidioschyzon merolae* 10D (*Matsuzaki et al., 2004*; *Nozaki et al., 2007*), *Galdieria sulphuraria* 074W (*Schönknecht et al., 2013*), and *Galdieria phlegrea* DBV009 (*Qiu et al., 2013*). The majority of putative HGTs in these taxa was hypothesized to have provided selective advantages during the evolution of polyextremophily, contributing to the ability of *Galdieria*, *Cyanidioschyzon*, and *Cyanidium* to cope with extremely low pH values, temperatures up to 56°C, as well as high salt and toxic heavy metal ion concentrations (*Castenholz and McDermott, 2010*; *Doemel and Brock, 1971*; *Reeb and Bhattacharya, 2010*; *Hsieh et al., 2018*). In such environments, they can represent up to 90% of the total biomass, competing with specialized Bacteria and Archaea (*Seckbach, 1972*), although some Cyanidiales strains also occur in more temperate environments (*Qiu et al., 2013*; *Gross et al., 2002*; *Ciniglia et al., 2004*; *Barcyté et al., 2018*; *Iovinella et al., 2018*). The integration and maintenance of HGT-derived genes, in spite of strong selection for genome reduction in these taxa (*Qiu et al., 2015*) underlines the potential ecological importance of this process to niche specialization (*Schönknecht et al., 2013*; *Qiu et al., 2013*; *Raymond and Kim, 2012*; *Bhattacharya et al., 2013*; *Foflonker et al., 2018*; *Schönknecht et al., 2014*). For this reason, we chose the Cyanidiales as a model lineage for studying eukaryotic HGT.

It should be appreciated that the correct identification of HGTs based on sequence similarity and phylogeny is rarely trivial and unambiguous, leaving much space for interpretation and erroneous assignments. In this context, previous findings regarding HGT in Cyanidiales were based on single genome analyses and have therefore been questioned (*Ku and Martin, 2016*).

Many potential sources of error need to be excluded during HGT analysis, such as possible bacterial contamination in the samples, algorithmic errors during genome assembly and annotation, phylogenetic model misspecification, and unaccounted for gene paralogy (*Richards and Monier, 2016*). In addition, eukaryotic HGT reports based on single gene tree analysis are prone to misinterpretation and may be a product of deep branching artefacts and low genome sampling. Indeed, false claims of prokaryote-to-eukaryote HGT have been published (*Boothby et al., 2015*; *Crisp et al., 2015*) which were later corrected (*Koutsovoulos et al., 2016*; *Salzberg, 2017*).

Here, we used multi-genomic analysis with 13 Cyanidiales lineages (including 10 novel, long-read, draft genome sequences) from nine geographically isolated habitats. This approach increased phylogenetic resolution within Cyanidiales to allow more accurate assessment of HGT while avoiding many of the above-mentioned sources of error. The following questions were addressed by our research: (i) did HGT have a significant impact on Cyanidiales evolution? (ii) Are previous HGT findings in the sequenced Cyanidiales genomes an artefact of short read assemblies, limited genome databases, and uncertainties associated with single gene trees, or do they hold up with more extensive sampling? (iii) And, assuming that evidence of eukaryotic HGT is found across multiple Cyanidiales species, are cumulative effects observable, or is DL the better explanation for these results?

**Table 1.** Summary of the 13 analyzed Cyanidiales genomes.

The existing genomes of *Galdieria sulphuraria* 074W, *Cyanidioschyzon merolae* 10D, and *Galdieria phlegrea* are marked with '#'. The remaining 10 genomes are novel. Genome Size (Mb): size of the genome assembly in Megabases. Contigs: number of contigs produced by the genome assembly. Contig N50 (kb): Contig N50. %GC Content: GC content of the genome given in percent. Genes: transcriptome size of species. Orthogroups: All Cyanidiales genes were clustered into a total of 9075 OGs. Here we show how many OGs there are per species. HGT Orthogroups: Number of OGs derived from HGT. HGT Genes: Number of HGT gene candidates found in species. %GC Native: GC content of the native transcriptome given in percent. %GC HGT: GC content of the HGT gene candidates given in percent % Multiexon Native: % of multiallelic genes in the native transcriptome. % Multiexon HGT: percent of multiallelic genes in the HGT gene candidates. S/M Native: Ratio of Multiexonic vs Singleexonic genes in native transcriptome. S/M HGT: Ratio of Multiexonic vs Singleexonic genes in HGT candidates. Asterisks (*) denote a significant difference (p<=0.05) between native and HGT gene subsets. EC, PFAM, GO, KEGG: Number of species-specific annotations in EC, PFAM, GO, KEGG.

| Strain | Genome features | | | | Gene and OG counts | | HGTs | | HGT vs native gene subsets | | | | | | Annotations | | | |
|---|---|---|---|---|---|---|---|---|---|---|---|---|---|---|---|---|---|---|
| | Genome Size (Mb) | Contigs | Contig N50 (kb) | %GC Content | Genes | Ortho groups | HGT ortho groups | HGT genes | %GC Native | %GC HGT | (%) Multi exon Native | (%) Multi exon HGT | Exon/ Gene Native | Exon/ Gene HGT | EC | PFAM | KEGG | GO |
| G. sulphuraria 074W# | 13.78 | 433 | 172.3 | 36.89 | 7174 | 5265 | 51 | 55 | 38.99 | 39.62* | 73.6 | 47.3* | 2.25 | 3.2* | 938 | 3073 | 3241 | 6572 |
| G. sulphuraria MS1 | 14.89 | 129 | 172.1 | 37.62 | 7441 | 5389 | 54 | 58 | 39.59 | 40.79* | 83.4 | 62.1* | 2.5 | 3.88* | 930 | 3077 | 3178 | 6564 |
| G. sulphuraria RT22 | 15.62 | 118 | 172.9 | 37.43 | 6982 | 5186 | 51 | 54 | 39.54 | 40.85* | 74.7 | 51.9* | 2.63 | 3.95* | 941 | 3118 | 3223 | 6504 |
| G. sulphuraria SAG21 | 14.31 | 135 | 158.2 | 37.92 | 5956 | 4732 | 44 | 47 | 40.04 | 41.47* | 84.8 | 83.0 | 4.02 | 5.03* | 931 | 3047 | 3143 | 6422 |
| G. sulphuraria MtSh | 14.95 | 101 | 186.6 | 40.04 | 6160 | 4746 | 46 | 47 | 41.33 | 42.48* | 79.7 | 63.8* | 3.15 | 4.32* | 939 | 3114 | 3244 | 6450 |
| G. sulphuraria Azora | 14.06 | 127 | 162.3 | 40.10 | 6305 | 4905 | 49 | 58 | 41.34 | 42.57* | 84.5 | 75.9* | 2.68 | 4.03* | 934 | 3072 | 3181 | 6474 |
| G. sulphuraria YNP5587.1 | 14.42 | 115 | 170.8 | 40.05 | 6118 | 4846 | 46 | 46 | 41.33 | 42.14* | 74.5 | 54.3* | 2.61 | 3.65* | 938 | 3084 | 3206 | 6516 |
| G. sulphuraria 5572 | 14.28 | 108 | 229.7 | 37.99 | 6472 | 5009 | 46 | 53 | 39.68 | 40.5* | 78.4 | 45.3* | 2.15 | 3.53* | 936 | 3108 | 3252 | 6540 |
| G. sulphuraria 002 | 14.11 | 107 | 189.3 | 39.16 | 5912 | 4701 | 46 | 52 | 40.76 | 41.35* | 97.1 | 50.0* | 2.37 | 3.73* | 927 | 3060 | 3184 | 6505 |
| G. phlegrea DBV009# | 11.41 | 9311 | 2.0 | 37.86 | 7836 | 5562 | 54 | 62 | 39.97 | 40.58* | na | na | na | na | 935 | 3018 | 3125 | 6512 |
| G. phlegrea Soos | 14.87 | 108 | 201.1 | 37.52 | 6125 | 4624 | 44 | 47 | 39.57 | 40.73* | 77.5 | 43.2* | 2.19 | 3.33* | 929 | 3034 | 3197 | 6493 |
| C. merolae 10D# | 16.73 | 22 | 859.1 | 54.81 | 4803 | 3980 | 33 | 33 | 56.57 | 56.57 | 0.5 | 0.0 | 1 | 1.01 | 883 | 2811 | 2832 | 6213 |
| C. merolae Soos | 12.33 | 35 | 567.5 | 54.33 | 4406 | 3574 | 34 | 34 | 54.84 | 54.26 | 9.4 | 2.9 | 1.06 | 1.1 | 886 | 2787 | 2823 | 6188 |

DOI: https://doi.org/10.7554/eLife.45017.003

## Results

### Features of the newly sequenced cyanidiales genomes

Genome sizes of the 10 targeted Cyanidiales (*Figure 1*) range from 12.33 Mbp - 15.62 Mbp, similar to other members of this red algal lineage (*Matsuzaki et al., 2004*; *Schönknecht et al., 2013*; *Qiu et al., 2013*) (*Table 1*). PacBio sequencing yielded 0.56 Gbp – 1.42 Gbp of raw sequence reads with raw read N50 ranging from 7.9 kbp – 14.4 kbp, which translated to a coverage of 28.91x – 70.99x at the unitigging stage (39.46x – 91.20x raw read coverage) (Appendix 1). We predicted a total of 61,869 novel protein coding sequences which, together with the protein data sets of the already published Cyanidiales species (total of 81,682 predicted protein sequences), capture 295/ 303 (97.4%) of the highly conserved eukaryotic BUSCO dataset. Each species, taken individually, scored an average of 92.7%. In spite of massive gene losses observed in the Cyanidiales (*Qiu et al., 2015*), these results corroborate previous observations that genome reduction has had a minor influence on the core eukaryotic gene inventory in free-living organisms (*Qiu et al., 2016*). Even *C. merolae* Soos, the species with the most limited coding capacity (4406 protein sequences), includes 89.5% of the eukaryotic BUSCO dataset. The number of contigs obtained from the *Galdieria* genomes ranged between 101–135. *G. sulphuraria* 17.91 (a strain different from the ones sequenced) was reported to have 40 chromosomes, and strains isolated from Rio Tinto (Spain), 47 or 57 chromosomes (*Moreira et al., 1994*). Pulsed-field gel electrophoresis indicates that *G. sulphuraria* 074W has approximately 42 chromosomes that are between 100 kbp and 1 Mbp in size (*Weber, 2007*). The genome assembly of *C. merolae* Soos produced 35 contigs, which approximates the 22 chromosomes (including plastid and mitochondrion) of the *C. merolae* 10D telomere-to-telomere assembly. Whole genome alignments indicate that a portion of the assembled contigs represent complete chromosomes.

### Orthogroups and phylogeny

The 81,682 predicted protein sequences from all 13 genomes clustered into a total of 9075 orthogroups and phylogenetic trees were built for each orthogroup. The reference species tree was constructed using 2,090 OGs that contained a single-copy gene in at least 12 of the 17 taxa (*Porphyra umbilicalis* (*Brawley et al., 2017*), *Porphyridium purpureum* (*Bhattacharya et al., 2013*), *Ostreococcus tauri* RCC4221 (*Blanc-Mathieu et al., 2014*), and *Chlamydomonas reinhardtii* (*Merchant et al., 2007*) were added to the dataset as outgroups). As a result, the species previously named *G. sulphuraria* Soos and *C. merolae* MS1 were reannotated as *G. phlegrea* Soos and *G. sulphuraria* MS1. Given these results, we sequenced a second genome of *C. merolae* and a representative of the *G. phlegrea* lineage. The species tree reflects previous findings that suggest more biodiversity exists within the Cyanidiales (*Ciniglia et al., 2004*) (*Figure 2*).

### Analysis of HGTs

The most commonly used approach to identify HGT candidates is to determine the position of eukaryotic and non-eukaryotic sequences in a maximum likelihood tree. Using this approach, 96 OGs were identified in which Cyanidiales genes shared a monophyletic descent with prokaryotes, representing 1.06% of all OGs. A total of 641 individual Cyanidiales sequences are considered as HGT candidates (*Table 1*). The amount of HGT per species varied considerably between members of the *Cyanidioschyzon* (33–34 HGT events, all single copy genes) and *Galdieria* lineages with 44–54 HGT events (52.6 HGT origins on average, 47–62 HGT gene candidates). In comparison to previous studies (*Schönknecht et al., 2013*; *Qiu et al., 2013*), no evidence of massive gene family expansion regarding HGT genes was found because the maximum number of gene copies in HGT orthogroups was three. We note, however, that one large gene family of STAND-type ATPases that was previously reported to originate from an archaeal HGT (*Schönknecht et al., 2013*) did not meet the criteria used in our restrictive Blast searches; that is the $10^{-5}$ e-value cut-off for consideration and a minimum of three different non-eukaryotic donors. This highly diverged family requires more sophisticated comparative analyses that were not done here (Appendix 2).

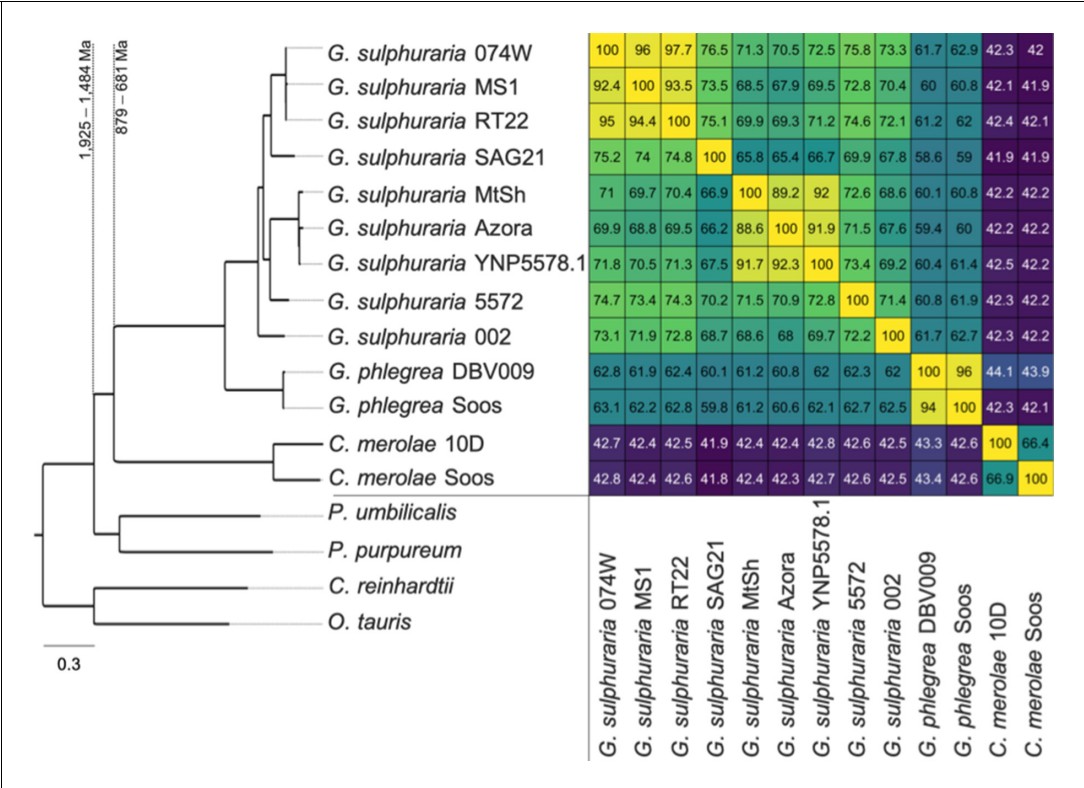

**Figure 2.** Species tree of the 13 analyzed extremophilic Cyanidiales genomes using mesophilic red (*Porphyra umbilicalis, Porphyridium purpureum*) and green algae (*Ostreococcus tauri, Chlamydomonas reinhardtii*) as outgroups. IQTREE was used to infer a single maximum-likelihood phylogeny based on orthogroups containing single-copy representative proteins from at least 12 of the 17 taxa (13 Cyanidiales + 4 Others). Each orthogroup alignment represented one partition with unlinked models of protein evolution chosen by IQTREE. Consensus tree branch support was determined by 2000 rapid bootstraps. All nodes in this tree had 100% bootstrap support, and are therefore not shown. Divergence time estimates are taken from *Yang et al. (2016)*. Similarity is derived from the average one-way best blast hit protein identity (minimum protein identity threshold = 30%). The minimal protein identity between two strains was 65.4%, measured between *g. sulphuraria* SAG21.92, which represent the second most distant sampling locations (12,350 km). Similar lineage boundaries were obtained for the *C. merolae* samples (66.4% protein identity), which are separated by only 1150 km.
DOI: https://doi.org/10.7554/eLife.45017.004

## Gene co-localization on raw sequence reads

One major issue associated with previous HGT studies is the incorporation of contaminant DNA into the genome assembly, leading to incorrect results (*Boothby et al., 2015*; *Crisp et al., 2015*; *Koutsovoulos et al., 2016*; *Salzberg, 2017*). Here, we screened for potential bacterial contamination in our tissue samples using PCR analysis of extracted DNA with the *rbcL* and 18S rRNA gene markers prior to sequencing. No instances of contamination were found. Furthermore, our work relied on PacBio RSII long-read sequencing technology, whereby single reads frequently exceed 10 kbp of DNA. Given these robust data, we also tested for co-occurrence of HGT gene candidates and 'native' genes in the same read. The protein sequences of each species were queried with tblastn ($10^{-5}$ e-value, 75 bitscore) against a database consisting of the uncorrected PacBio RSII long reads. This analysis showed that 629/641 (98.12%) of the HGT candidates co-localize with native red algal genes on the same read (38,297 reads in total where co-localization of native genes and HGT candidates was observed). It should be noted that the 10 novel genomes we determined share HGT candidates with *C. merolae* 10D, *G. sulphuraria* 074W, and *G. phlegrea* DBV009, which were sequenced in different laboratories, at different points in time, using different technologies, and assembly pipelines. Hence, we consider it highly unlikely that these HGT candidates result from bacterial contamination. As the accuracy of long read sequencing technologies further increases, we believe this criterion for excluding bacterial contamination provides an additional piece of evidence that should be added to the guidelines for HGT discovery (*Richards and Monier, 2016*).

## Differences in molecular features between native and HGT-derived genes

One of the main consequences of HGT is that horizontally acquired genes may have different structural characteristics when compared to native genes (cumulative effects). HGT-derived genes initially retain characteristics of the genome of the donor lineage. Consequently, the passage of time is required (and expected) to erase these differences. Therefore, we searched for differences in genomic features between HGT candidates and native Cyanidiales genes with regard to: (1) GC-content, (2) the number of spliceosomal introns and the exon/gene ratio, (3) differential transcription, (4) percent protein identity between HGT genes and their non-eukaryotic donors, and (5) cumulative effects as indicators of their non-eukaryotic origin (*Danchin, 2016*; *Ku and Martin, 2016*; *Schönknecht et al., 2013*).

### GC-content

All 11 *Galdieria* species showed significant differences (GC-content of transcripts is normally distributed, Student's *t*-test, two-sided, p≤0.05) in percent GC-content between native sequences and HGT candidates (*Table 1*). Sequences belonging to the *Galdieria* lineage have an exceptionally low GC-content (39%–41%) in comparison to the majority of thermophilic organisms that exhibit higher values (~55%). On average, HGT candidates in *Galdieria* display 1% higher GC-content in comparison to their native counterparts. No significant differences were found for *C. merolae* 10D and *C. merolae* Soos in this respect. Because native *Cyanidioschyzon* genes have an elevated GC-content (54%–56%), this makes it difficult to distinguish between them and HGT-derived genes (Appendix 3).

### Spliceosomal introns and exon/Gene

Bacterial genes lack spliceosomal introns and therefore the spliceosomal machinery. Consequently, genes acquired through HGT are initially single-exons and may acquire introns over time due to the invasion of existing intervening sequences. We detected significant discrepancies in the ratio of single-exon to multi-exon genes between HGT candidates and native genes in the *Galdieria* lineage. On average, 42% of the *Galdieria* HGT candidates are single-exon genes, whereas only 19.2% of the native gene set are comprised of single-exons. This difference is significant (categorical data, 'native' vs 'HGT' and 'single exon' vs. 'multiple exon', Fisher's exact test, p≤0.05) in all *Galdieria* species except *G. sulphuraria* SAG21.92 (*Table 1*). The *Cyanidioschyzon* lineage contains a highly reduced spliceosomal machinery (*Qiu et al., 2018*), therefore only ~10% of native genes are multi-exonic in *C. merolae* Soos and only 1/34 HGT candidates has gained an intron. *C. merolae* 10D has only 26 multi-exonic genes (~0.5% of all transcripts) and none of its HGT candidates has gained an intron. Enrichment testing is not possible with these small sample sizes (Appendix 4).

We analyzed the number of exons that are present in multi-exonic genes and obtained similar results for the *Galdieria* lineage (*Table 1*). All *Galdieria* species show significant differences regarding the exon/gene ratio between native and HGT genes (non-normal distribution regarding the number of exons per gene, Wilcoxon-Mann-Whitney-Test, 1000 bootstraps, p<=0.05). HGT candidates in *Galdieria* have 0.97–1.36 fewer exons per gene in comparison to their native counterparts. Because the multi-exonic HGT subset in both *Cyanidioschyzon* species combined includes only one multi-exonic HGT candidate, no further analysis was performed (Appendix 4).

### Differential transcription

Several RNA-Seq datasets are publicly available for *G. sulphuraria* 074W (*Rossoni, 2018*) and *C. merolae* 10D (*Rademacher et al., 2016*). We aligned (*Kim et al., 2015*) the transcriptome reads to the respective genomes, using an identical data processing pipeline (*Robinson et al., 2010*) for both datasets to exclude potential algorithmic errors (*Figure 3*). The average read count per gene (measured as counts per million, CPM), of native genes was 154 CPM in *G. sulphuraria* 074W and 196 CPM *C. merolae* 10D. The average read counts for HGT candidates in *G. sulphuraria* 074W and *C. merolae* 10D were 130 CPM and 184 CPM, respectively. No significant differences in RNA abundance between native genes and HGT candidates were observed for these taxa (non-normal distribution of CPM, Wilcoxon-Mann-Whitney-Test, p<0.05).

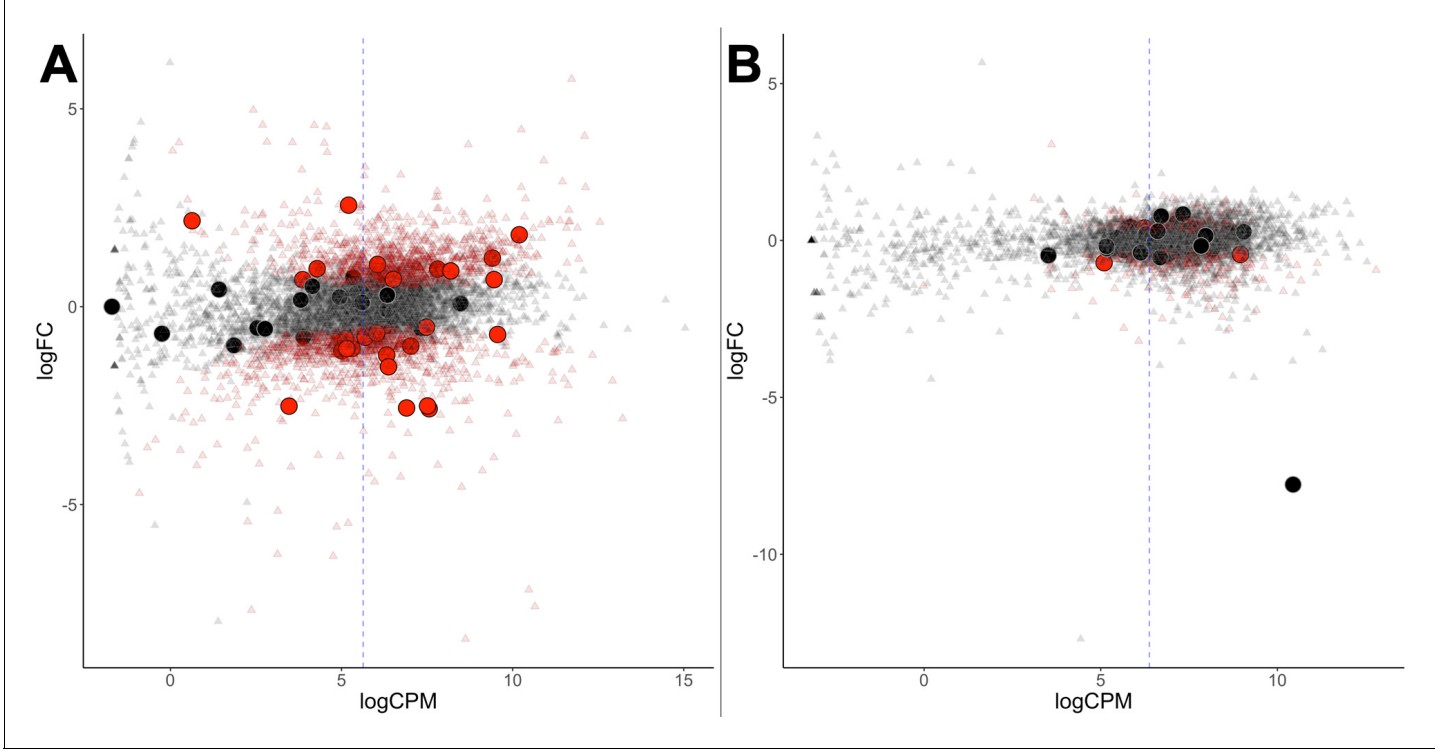

**Figure 3.** Differential gene expression of *G. sulphuraria* 074W. (**A**) and *C. merolae* 10D (**B**), here measured as log fold change (logFC) vs transcription rate (logCPM). Differentially expressed genes are colored red (quasi-likelihood (QL) F-test, Benjamini-Hochberg, $p <= 0.01$). HGT candidates are shown as large circles. The blue dashes indicate the average logCPM of the dataset. Although HGT candidates are not significantly more or less expressed than native genes, they react significantly stronger to temperature changes in *G. sulphuraria* 074W ('more red than black dots'). This is not the case in high $CO_2$ treated *C. merolae* 10D.

DOI: https://doi.org/10.7554/eLife.45017.005

### Gene function – not passage of time – explains percent protein identity (PID) between Cyanidiales HGT candidates and their non-eukaryotic donors

Once acquired, any HGT-derived gene may be fixed in the genome and propagated across the lineage. The PID data can be further divided into different subsets depending on species composition of the OG. Of the total 96 OGs putatively derived from HGT events, 60 are exclusive to the *Galdieria* lineage (62.5%), 23 are exclusive to the *Cyanidioschyzon* lineage (24%), and 13 are shared by both lineages (13.5%) (*Figure 4A*). Consequently, either a strong prevalence for lineage specific DL exists, or both lineages underwent individual sets of HGT events because they share their habitat with other non-eukaryotic species (which is what the HGT theory would assume). The 96 OGs in question are affected by gene loss or partial fixation. Once acquired only 8/13 of the 'Cyanidiales' (including 'Multiple HGT' and 'Uncertain') OGs and 20/60 of the *Galdieria* specific OGs are encoded by all species. Once acquired by the *Cyanidioschyzon* ancestor, the HGT candidates were retained by both *C. merolae* Soos and *C. merolae* 10D in 22/23 *Cyanidioschyzon* specific OGs. It is not possible to verify whether the only *Cyanidioschyzon* OG containing one HGT candidate is the result of gene loss, individual acquisition, or due to erroneously missing this gene model during gene prediction. The average percent PID between HGT gene candidates of the 13 OGs shared by all Cyanidiales and their non-eukaryotic donors is 41.2% (min = 24.4%; max = 65.4%) (*Figure 4B*). From the HGT perspective, these OGs are derived from ancient HGT events that occurred at the root of the Cyanidiales, well before the split of the *Galdieria* and *Cyanidioschyzon* lineages. The OGs were retained over time in all Cyanidiales, although evidence of subsequent gene loss is observed. Under the DL hypothesis, this group of OGs contains genes that have been lost in all other eukaryotic lineages except the Cyanidiales. Similarly, the average PID between HGT candidates their non-eukaryotic donors in OGs

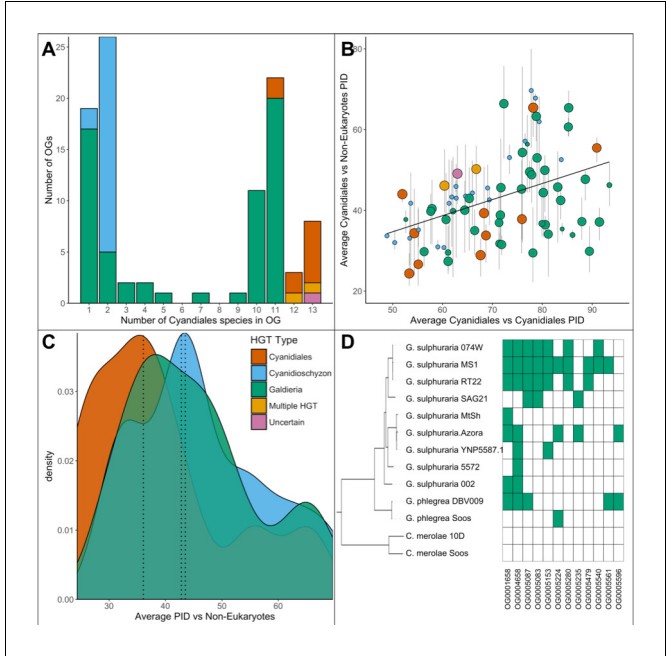

**Figure 4.** Comparative analysis of the 96 OGs potentially derived from HGT. (**A**) OG count vs. the number of Cyanidiales species contained in an OG (=OG size). Only genes from the sequenced genomes were considered (13 species). A total of 60 OGs are exclusive to the *Galdieria* lineage (11 species), 23 OGs are exclusive to the *Cyanidioschyzon* lineage (two species), and 13 OGs are shared by both lineages. A total of 46/96 HGT events seem to be affected by later gene erosion/partial fixation. (**B**) OG-wise PID between HGT candidates vs. their potential non-eukaryotic donors. Point size represents the number of sequenced species contained in each OG. Because only two genomes of *Cyanidioschyzon* were sequenced, the maximum point size for this lineage is 2. The whiskers span minimum and maximum shared PID of each OG. The PID within Cyanidiales HGTs vs. PID between Cyanidiales HGTs and their potential non-eukaryotic donors is positively correlated (Kendall's tau coefficient, p=0.000747), showing evolutionary constraints that are gene function dependent, rather than time-dependent. (**C**) Density curve of average PID towards potential non-eukaryotic donors. The area under each curve is equal to 1. The average PID of HGT candidates found in both lineages ('ancient HGT', left dotted line) is ~5% lower than the average PID of HGT candidates exclusive to *Galdieria* or *Cyandioschyzon* ('recent HGT', right dotted lines). This difference is not significant (pairwise Wilcoxon rank-sum test, Benjamini-Hochberg, p>0.05). (**D**) Presence/Absence pattern (green/white) of Cyanidiales species in HGT OGs. Some patterns strictly follow the branching structure of the species tree. They represent either recent HGTs that affect a monophyletic subset of the *Galdieria* lineage, or are the last eukaryotic remnants of an ancient gene that was eroded through differential loss. In other cases, the presence/absence pattern of *Galdieria* species is random and conflicts with the *Galdieria* lineage phylogeny. HGT would assume either multiple independent acquisitions of the same HGT candidate, or a partial fixation of the HGT candidate in the lineage, while still allowing for gene erosion. According to DL, these are the last existing paralogs of an ancient gene, whose erosion within the eukaryotic kingdom is nearly complete.
DOI: https://doi.org/10.7554/eLife.45017.006

exclusive to the *Cyanidioschyzon* lineage is 46.4% (min = 30.8%; max = 69.7%) and 45.1% (min = 27.4%; max = 69.5%) for those OGs exclusive to the *Galdieria* lineage. According to the HGT view, these subsets of candidates were horizontally acquired either in the *Cyanidioschyzon* lineage, or in the *Galdieria* lineage after the split between *Galdieria* and *Cyanidioschyzon*. DL would impose gene loss on all other eukaryotic lineages except *Galdieria* or *Cyanidioschyzon*. Over time, sequence similarity between the HGT candidate and the non-eukaryotic donor is expected to decrease at a rate that reflects the level of functional constraint. The average PID of 'ancient' HGT candidates shared by both lineages (before the split into *Galdieria* and *Cyanidioschyzon* approximately 800 Ma years ago [*Yang et al., 2016*]) is ~5% lower than the average PID of HGT candidates exclusive to one lineage which, according to HGT would represent more recent HGT events because their acquisition occurred only after the split (thus lower divergence) (*Figure 4C*). However, no significant difference between *Galdieria*-exclusive HGTs, *Cyandioschyzon*-exclusive HGTs, and HGTs shared by

both lineages was found (non-normal distribution of percent protein identity, Shapiro-Wilk normality test, W = 0.95, p=0.002; Pairwise Wilcoxon rank-sum test, Benjamini-Hochberg, all comparisons p>0.05). Therefore, neither *Cyanidioschyzon* nor *Galdieria* specific HGTs, or HGTs shared by all Cyanidiales, are significantly more, or less, similar to their potential prokaryotic donors. We also addressed the differences in PID within the three groups. The average PID within HGT gene candidates of the 13 OGs shared by all Cyanidiales is 75.0% (min = 51.9%; max = 90.9%) (*Figure 4B*). Similarly, the average PID within HGT candidates in OGs exclusive to the *Cyanidioschyzon* lineage is 65.1% (min = 48.9%; max = 83.8%) and 75.0% (min = 52.6%; max = 93.4%) for those OGs exclusive to the *Galdieria* lineage. Because we sampled only two *Cyanidioschyzon* species in comparison to 11 *Galdieria* lineages that are also much more closely related (*Figure 2A*), a comparison between these two groups was not done. However, a significant positive correlation (non-normal distribution of PID across all OGs, Kendall's tau coefficient, p=0.000747) exists between the PID within Cyanidiales HGTs versus PID between Cyanidiales HGTs and their non-eukaryotic donors (*Figure 4B*). Hence, the more similar Cyanidiales sequences are to each other, the more similar they are to their non-eukaryotic donors, showing gene function dependent evolutionary constraints.

## Complex origins of HGT-impacted orthogroups

While comparing the phylogenies of HGT candidates, we also noted that not all Cyanidiales genes within one OG necessarily originate via HGT (phylogenetic trees of each HGT-OG are included in *Figure 5—figure supplements 1–96*). Among the 13 OGs that contain HGT candidates present in both *Galdieria* and *Cyanidioschyzon*, we found two cases (*Figure 4A*, 'Multiple HGT'), OG0002305 and OG0003085, in which *Galdieria* and *Cyanidioschyzon* HGT candidates cluster in the same orthogroup. However, these have different non-eukaryotic donors and are located on distinct phylogenetic branches that do not share a monophyletic descent (*Figure 5A*). This is potentially the case for OG0002483 as well, but we were uncertain due to low bootstrap values (*Figure 4A*, 'Uncertain'). These OGs either represent two independent acquisitions of the same function or, according to DL, the LECA encoded three paralogs of the same gene which were propagated through evolutionary time. One of these was retained by the *Galdieria* lineage (and shares sequence similarity with one group of prokaryotes), the second was retained by *Cyanidioschyzon* (and shares sequence similarity with a different group of prokaryotes), and a third paralog was retained by all other eukaryotes. It should be noted that the 'other eukaryotes' do not always cluster in one uniformly eukaryotic clade which increases the number of required paralogs in LECA to explain the current pattern. Furthermore, some paralogs could also have already been completely eroded and do not exist in extant eukaryotes. Similarly, 6/60 *Galdieria* specific OGs also contain *Cyanidioschyzon* genes (OG0001929, OG0001938, OG0002191, OG0002574, OG0002785 and OG0003367). Here, they are nested within other eukaryote lineages and would not be derived from HGT (*Figure 5B*). Also, eight of the 23 *Cyanidioschyzon* specific HGT OGs contain genes from *Galdieria* species (OG0001807, OG0001810, OG0001994, OG0002727, OG0002871, OG0003539, OG0003929 and OG0004405) which cluster within the eukaryotic branch and are not monophyletic with *Cyanidioschyzon* HGT candidates (*Figure 5C*). According to the HGT view, this subset of candidates was horizontally acquired in either the *Cyanidioschyzon* lineage, or the *Galdieria* lineage only after the split between *Galdieria* and *Cyanidioschyzon*, possibly replacing the ancestral gene or functionally complementing a function that was lost due to genome reduction. According to DL, the LECA would have encoded two paralogs of the same gene. One was retained by all eukaryotes, red algae, and *Galdieria* or *Cyanidioschyzon*, the other exclusively by *Cyanidioschyzon* or *Galdieria* together with non-eukaryotes.

## Stronger erosion of HGT genes impedes assignment to HGT or DL

As already noted above, only 50/96 of the sampled HGT-impacted OGs do not appear to be affected by erosion. Dense sampling of 11 taxa within the *Galdieria* lineage allowed a more in-depth analysis of this issue. Here, a bimodal distribution is observed regarding the number of species per OG in the native and HGT dataset (*Figure 6C*). Only 52.5% of the native gene set is present in all *Galdieria* strains (defined as 10 and 11 strains in order to account for potential misassemblies and missed gene models during prediction). Approximately 1/3 of the native OGs (36.1%) has been affected by gene erosion to such a degree that it is present in only one, or two *Galdieria* strains. In comparison, 26.7% of the candidate HGT-impacted OGs are encoded in >10 *Galdieria* strains,

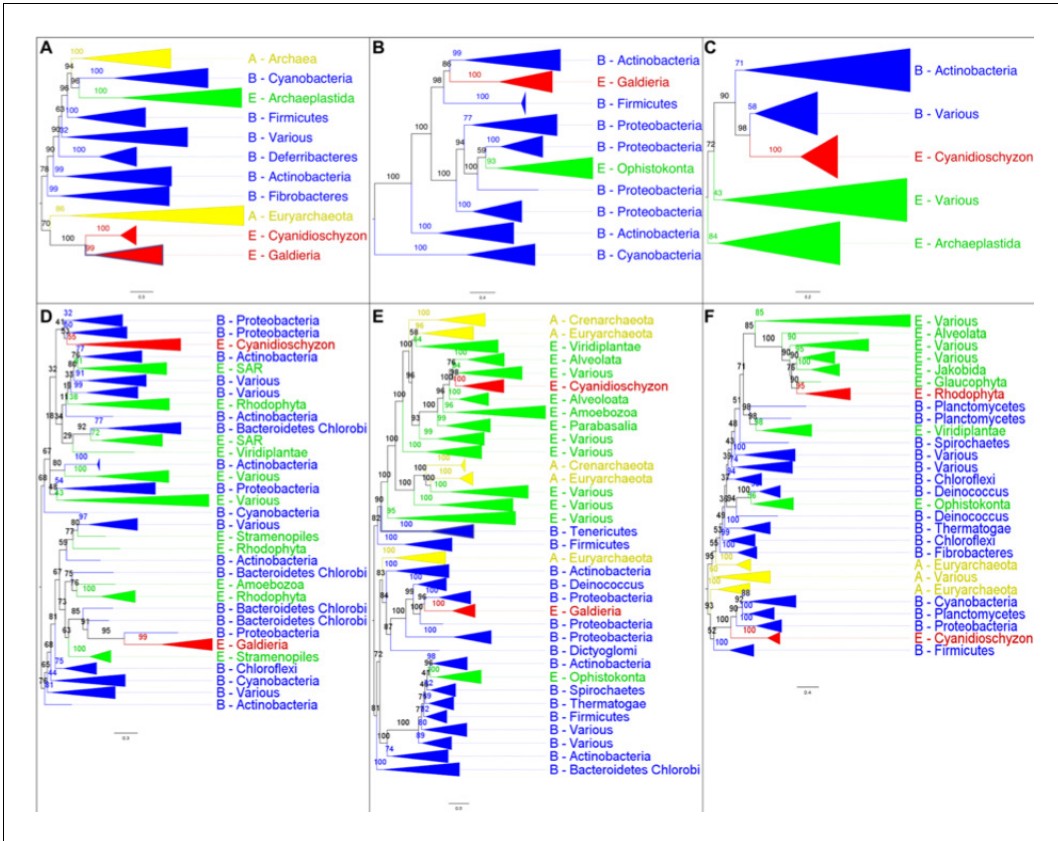

**Figure 5.** The analysis of OGs containing HGT candidates revealed different patterns of HGT acquisition. Some OGs contain genes that are shared by all Cyanidiales, whereas others are unique to the *Galdieria* or *Cyanidioschyzon* lineage. In some cases, HGT appears to have replaced the eukaryotic genes in one lineage, whereas the other lineage maintained the eukaryotic ortholog. Here, some examples of OG phylogenies are shown, which were simplified for ease of presentation. The first letter of the tip labels indicates the kingdom. A = Archaea (yellow), B = Bacteria (blue), E = Eukaryota (green). Branches containing Cyanidiales sequences are highlited in red. (A) Example of an ancient HGT that occurred before *Galdieria* and *Cyanidioschyzon* split into separate lineages. As such, both lineages are monophyletic (e.g., OG0001476). (B) HGT candidates are unique to the *Galdieria* lineage (e.g. OG0001760). (C) HGT candidates are unique to the *Cyanidioschyzon* lineage (e.g. OG0005738). (D) *Galdieria* and *Cyanidioschyzon* HGT candidates are derived from different HGT events and share monophyly with different non-eukaryotic organisms (e.g., OG0003085). (E) *Galdieria* HGT candidates cluster with non-eukaryotes, whereas the *Cyanidioschyzon* lineage clusters with eukaryotes (e.g., OG0001542). (F) *Cyanidioschyzon* HGT candidates cluster with non-eukaryotes, whereas the *Galdieria* lineage clusters with eukaryotes (e.g., OG0006136).

DOI: https://doi.org/10.7554/eLife.45017.007

The following figure supplements are available for figure 5:

**Figure supplement 1.** Sequence tree of orthogroup OG0001476.
DOI: https://doi.org/10.7554/eLife.45017.008

**Figure supplement 2.** Sequence tree of orthogroup OG0001486.
DOI: https://doi.org/10.7554/eLife.45017.009

**Figure supplement 3.** Sequence tree of orthogroup OG0001509.
DOI: https://doi.org/10.7554/eLife.45017.010

**Figure supplement 4.** Sequence tree of orthogroup OG0001513.
DOI: https://doi.org/10.7554/eLife.45017.011

**Figure supplement 5.** Sequence tree of orthogroup OG0001542.
DOI: https://doi.org/10.7554/eLife.45017.012

**Figure supplement 6.** Sequence tree of orthogroup OG0001613.
DOI: https://doi.org/10.7554/eLife.45017.013

**Figure supplement 7.** Sequence tree of orthogroup OG0001658.

*Figure 5 continued*

DOI: https://doi.org/10.7554/eLife.45017.014
**Figure supplement 8.** Sequence tree of orthogroup OG0001760.
DOI: https://doi.org/10.7554/eLife.45017.015
**Figure supplement 9.** Sequence tree of orthogroup OG0001807.
DOI: https://doi.org/10.7554/eLife.45017.016
**Figure supplement 10.** Sequence tree of orthogroup OG0001810.
DOI: https://doi.org/10.7554/eLife.45017.017
**Figure supplement 11.** Sequence tree of orthogroup OG0001929.
DOI: https://doi.org/10.7554/eLife.45017.018
**Figure supplement 12.** Sequence tree of orthogroup OG0001938.
DOI: https://doi.org/10.7554/eLife.45017.019
**Figure supplement 13.** Sequence tree of orthogroup OG0001955.
DOI: https://doi.org/10.7554/eLife.45017.020
**Figure supplement 14.** Sequence tree of orthogroup OG0001976.
DOI: https://doi.org/10.7554/eLife.45017.021
**Figure supplement 15.** Sequence tree of orthogroup OG0001994.
DOI: https://doi.org/10.7554/eLife.45017.022
**Figure supplement 16.** Sequence tree of orthogroup OG0002036.
DOI: https://doi.org/10.7554/eLife.45017.023
**Figure supplement 17.** Sequence tree of orthogroup OG0002051.
DOI: https://doi.org/10.7554/eLife.45017.024
**Figure supplement 18.** Sequence tree of orthogroup OG0002191.
DOI: https://doi.org/10.7554/eLife.45017.025
**Figure supplement 19.** Sequence tree of orthogroup OG0002305.
DOI: https://doi.org/10.7554/eLife.45017.026
**Figure supplement 20.** Sequence tree of orthogroup OG0002337.
DOI: https://doi.org/10.7554/eLife.45017.027
**Figure supplement 21.** Sequence tree of orthogroup OG0002431.
DOI: https://doi.org/10.7554/eLife.45017.028
**Figure supplement 22.** Sequence tree of orthogroup OG0002483.
DOI: https://doi.org/10.7554/eLife.45017.029
**Figure supplement 23.** Sequence tree of orthogroup OG0002574.
DOI: https://doi.org/10.7554/eLife.45017.030
**Figure supplement 24.** Sequence tree of orthogroup OG0002578.
DOI: https://doi.org/10.7554/eLife.45017.031
**Figure supplement 25.** Sequence tree of orthogroup OG0002609.
DOI: https://doi.org/10.7554/eLife.45017.032
**Figure supplement 26.** Sequence tree of orthogroup OG0002676.
DOI: https://doi.org/10.7554/eLife.45017.033
**Figure supplement 27.** Sequence tree of orthogroup OG0002727.
DOI: https://doi.org/10.7554/eLife.45017.034
**Figure supplement 28.** Sequence tree of orthogroup OG0002785.
DOI: https://doi.org/10.7554/eLife.45017.035
**Figure supplement 29.** Sequence tree of orthogroup OG0002871.
DOI: https://doi.org/10.7554/eLife.45017.036
**Figure supplement 30.** Sequence tree of orthogroup OG0002896.
DOI: https://doi.org/10.7554/eLife.45017.037
**Figure supplement 31.** Sequence tree of orthogroup OG0002999.
DOI: https://doi.org/10.7554/eLife.45017.038
**Figure supplement 32.** Sequence tree of orthogroup OG0003085.
DOI: https://doi.org/10.7554/eLife.45017.039
**Figure supplement 33.** Sequence tree of orthogroup OG0003250.
DOI: https://doi.org/10.7554/eLife.45017.040
**Figure supplement 34.** Sequence tree of orthogroup OG0003367.
DOI: https://doi.org/10.7554/eLife.45017.041
**Figure supplement 35.** Sequence tree of orthogroup OG0003441.

*Figure 5 continued*

DOI: https://doi.org/10.7554/eLife.45017.042

**Figure supplement 36.** Sequence tree of orthogroup OG0003539.

DOI: https://doi.org/10.7554/eLife.45017.043

**Figure supplement 37.** Sequence tree of orthogroup OG0003777.

DOI: https://doi.org/10.7554/eLife.45017.044

**Figure supplement 38.** Sequence tree of orthogroup OG0003782.

DOI: https://doi.org/10.7554/eLife.45017.045

**Figure supplement 39.** Sequence tree of orthogroup OG0003834.

DOI: https://doi.org/10.7554/eLife.45017.046

**Figure supplement 40.** Sequence tree of orthogroup OG0003846.

DOI: https://doi.org/10.7554/eLife.45017.047

**Figure supplement 41.** Sequence tree of orthogroup OG0003856.

DOI: https://doi.org/10.7554/eLife.45017.048

**Figure supplement 42.** Sequence tree of orthogroup OG0003901.

DOI: https://doi.org/10.7554/eLife.45017.049

**Figure supplement 43.** Sequence tree of orthogroup OG0003905.

DOI: https://doi.org/10.7554/eLife.45017.050

**Figure supplement 44.** Sequence tree of orthogroup OG0003907.

DOI: https://doi.org/10.7554/eLife.45017.051

**Figure supplement 45.** Sequence tree of orthogroup OG0003929.

DOI: https://doi.org/10.7554/eLife.45017.052

**Figure supplement 46.** Sequence tree of orthogroup OG0003954.

DOI: https://doi.org/10.7554/eLife.45017.053

**Figure supplement 47.** Sequence tree of orthogroup OG0004030.

DOI: https://doi.org/10.7554/eLife.45017.054

**Figure supplement 48.** Sequence tree of orthogroup OG0004102.

DOI: https://doi.org/10.7554/eLife.45017.055

**Figure supplement 49.** Sequence tree of orthogroup OG0004142.

DOI: https://doi.org/10.7554/eLife.45017.056

**Figure supplement 50.** Sequence tree of orthogroup OG0004203.

DOI: https://doi.org/10.7554/eLife.45017.057

**Figure supplement 51.** Sequence tree of orthogroup OG0004258.

DOI: https://doi.org/10.7554/eLife.45017.058

**Figure supplement 52.** Sequence tree of orthogroup OG0004339.

DOI: https://doi.org/10.7554/eLife.45017.059

**Figure supplement 53.** Sequence tree of orthogroup OG0004392.

DOI: https://doi.org/10.7554/eLife.45017.060

**Figure supplement 54.** Sequence tree of orthogroup OG0004405.

DOI: https://doi.org/10.7554/eLife.45017.061

**Figure supplement 55.** Sequence tree of orthogroup OG0004486.

DOI: https://doi.org/10.7554/eLife.45017.062

**Figure supplement 56.** Sequence tree of orthogroup OG0004658.

DOI: https://doi.org/10.7554/eLife.45017.063

**Figure supplement 57.** Sequence tree of orthogroup OG0005083.

DOI: https://doi.org/10.7554/eLife.45017.064

**Figure supplement 58.** Sequence tree of orthogroup OG0005087.

DOI: https://doi.org/10.7554/eLife.45017.065

**Figure supplement 59.** Sequence tree of orthogroup OG0005153.

DOI: https://doi.org/10.7554/eLife.45017.066

**Figure supplement 60.** Sequence tree of orthogroup OG0005224.

DOI: https://doi.org/10.7554/eLife.45017.067

**Figure supplement 61.** Sequence tree of orthogroup OG0005235.

DOI: https://doi.org/10.7554/eLife.45017.068

**Figure supplement 62.** Sequence tree of orthogroup OG0005280.

DOI: https://doi.org/10.7554/eLife.45017.069

**Figure supplement 63.** Sequence tree of orthogroup OG0005479.

*Figure 5 continued*

DOI: https://doi.org/10.7554/eLife.45017.070

**Figure supplement 64.** Sequence tree of orthogroup OG0005540.
DOI: https://doi.org/10.7554/eLife.45017.071

**Figure supplement 65.** Sequence tree of orthogroup OG0005561.
DOI: https://doi.org/10.7554/eLife.45017.072

**Figure supplement 66.** Sequence tree of orthogroup OG0005596.
DOI: https://doi.org/10.7554/eLife.45017.073

**Figure supplement 67.** Sequence tree of orthogroup OG0005683.
DOI: https://doi.org/10.7554/eLife.45017.074

**Figure supplement 68.** Sequence tree of orthogroup OG0005694.
DOI: https://doi.org/10.7554/eLife.45017.075

**Figure supplement 69.** Sequence tree of orthogroup OG0005738.
DOI: https://doi.org/10.7554/eLife.45017.076

**Figure supplement 70.** Sequence tree of orthogroup OG0005963.
DOI: https://doi.org/10.7554/eLife.45017.077

**Figure supplement 71.** Sequence tree of orthogroup OG0005984.
DOI: https://doi.org/10.7554/eLife.45017.078

**Figure supplement 72.** Sequence tree of orthogroup OG0006136.
DOI: https://doi.org/10.7554/eLife.45017.079

**Figure supplement 73.** Sequence tree of orthogroup OG0006143.
DOI: https://doi.org/10.7554/eLife.45017.080

**Figure supplement 74.** Sequence tree of orthogroup OG0006191.
DOI: https://doi.org/10.7554/eLife.45017.081

**Figure supplement 75.** Sequence tree of orthogroup OG0006251.
DOI: https://doi.org/10.7554/eLife.45017.082

**Figure supplement 76.** Sequence tree of orthogroup OG0006252.
DOI: https://doi.org/10.7554/eLife.45017.083

**Figure supplement 77.** Sequence tree of orthogroup OG0006435.
DOI: https://doi.org/10.7554/eLife.45017.084

**Figure supplement 78.** Sequence tree of orthogroup OG0006482.
DOI: https://doi.org/10.7554/eLife.45017.085

**Figure supplement 79.** Sequence tree of orthogroup OG0006498.
DOI: https://doi.org/10.7554/eLife.45017.086

**Figure supplement 80.** Sequence tree of orthogroup OG0006623.
DOI: https://doi.org/10.7554/eLife.45017.087

**Figure supplement 81.** Sequence tree of orthogroup OG0006670.
DOI: https://doi.org/10.7554/eLife.45017.088

**Figure supplement 82.** Sequence tree of orthogroup OG0007051.
DOI: https://doi.org/10.7554/eLife.45017.089

**Figure supplement 83.** Sequence tree of orthogroup OG0007123.
DOI: https://doi.org/10.7554/eLife.45017.090

**Figure supplement 84.** Sequence tree of orthogroup OG0007346.
DOI: https://doi.org/10.7554/eLife.45017.091

**Figure supplement 85.** Sequence tree of orthogroup OG0007383.
DOI: https://doi.org/10.7554/eLife.45017.092

**Figure supplement 86.** Sequence tree of orthogroup OG0007550.
DOI: https://doi.org/10.7554/eLife.45017.093

**Figure supplement 87.** Sequence tree of orthogroup OG0007551.
DOI: https://doi.org/10.7554/eLife.45017.094

**Figure supplement 88.** Sequence tree of orthogroup OG0007596.
DOI: https://doi.org/10.7554/eLife.45017.095

**Figure supplement 89.** Sequence tree of orthogroup OG0008189.
DOI: https://doi.org/10.7554/eLife.45017.096

**Figure supplement 90.** Sequence tree of orthogroup OG0008334.
DOI: https://doi.org/10.7554/eLife.45017.097

**Figure supplement 91.** Sequence tree of orthogroup OG0008335.

*Figure 5 continued*

DOI: https://doi.org/10.7554/eLife.45017.098

**Figure supplement 92.** Sequence tree of orthogroup OG0008579.

DOI: https://doi.org/10.7554/eLife.45017.099

**Figure supplement 93.** Sequence tree of orthogroup OG0008680.

DOI: https://doi.org/10.7554/eLife.45017.100

**Figure supplement 94.** Sequence tree of orthogroup OG0008822.

DOI: https://doi.org/10.7554/eLife.45017.101

**Figure supplement 95.** Sequence tree of orthogroup OG0008898.

DOI: https://doi.org/10.7554/eLife.45017.102

**Figure supplement 96.** Sequence tree of orthogroup OG0008996.

DOI: https://doi.org/10.7554/eLife.45017.103

whereas 53.0% are present in less than three. The latter number might be an underestimation due to the strict threshold for HGT discovery which led to the removal of HGT candidates that were singletons. The HGT distribution is therefore skewed towards OGs containing only a few or one *Galdieria* species as the result of recent HGT events that occurred; for example after the split of *G. sulphuraria* and *G. phlegrea*. In spite of the strong erosion which would also lead to partial fixation of presumably recent HGT events, we analyzed whether the distribution patterns of HGT candidates across the sequenced genomes reflect the branching pattern of the species trees (*Figure 4C*). This is true for all HGT candidates that are exclusive to the *Cyanidioschyzon* or *Galdieria* lineage. Either the HGT candidates were acquired after the split of the two lineages (according to HGT), or differentially lost in one of the two lineages (according to DL). In the 60 *Galdieria* specific OGs we found 12 OGs containing less than 10 and more than one *Galdieria* species (*Figure 4C*). In 5/12 of the cases, the presence absence pattern reflects the species tree (OG0005087, OG0005083, GO0005479, OG0005540). Here, the potential HGT candidates are not found in any other eukaryotic species. According to HGT, they were acquired by a monophyletic sub-clade of the *Galdieria* lineage. According to DL, they were lost in all eukaryotes with the exception of this subset of the *Galdieria* lineage (e.g., OG0005280 and OG0005083 were potentially acquired or maintained exclusively by the last common ancestor of *G. sulphuraria* 074W, *G. sulphuraria* MS1, *G. sulphuraria* RT22, and *G. sulphuraria* SAG21). In the remaining OGs, the HGT gene candidate is distributed across the *Galdieria* lineage and conflicts with the branching pattern of the species tree. HGT would assume either multiple independent acquisitions of the same HGT candidate, or partial fixation of the HGT candidate in the lineage, while still allowing for gene erosion. According to DL, these are the last existing paralogs of an ancient gene, whose erosion within the eukaryotic kingdom is nearly complete. However, it must be considered that in some cases, DL must have occurred independently across multiple species in a brief of time after the gene was maintained for hundreds of millions of years across the lineage (e.g., OG0005224 contains *G. phlegrea* Soos, *G. sulphuraria* Azora and *G. sulphuraria* MS1). This implies that the gene was present in the ancestor of the *Galdieria* lineage and also in the last common ancestor of closely related *G. sulphuraria* MS1, *G. sulphuraria* 074W and *G. sulphuraria* RT22 (as well as *G. sulphuraria* SAG21) and the last common ancestor of closely related *G. sulphuraria* MtSh, *G. sulphuraria* Azora and *G. sulphuraria* YNP5587.1 (as well as *G. sulphuraria* 5572). A gene that was encoded and maintained since LECA, was lost independently in 6/8 species within the past few million years.

## The seventy percent rule

In their analysis of eukaryotic HGT (*Ku and Martin, 2016*), Ku and co-authors reach the conclusion that prokaryotic homologs of genes in eukaryotic genomes that share >70% PID are not found outside individual genome assemblies (unless derived from endosymbiotic gene transfer, EGT). Hence, they are considered as assembly artifacts. We analyzed whether our dataset supports this rule, or alternatively, it is arbitrary and a byproduct of the analysis approach used, combined with low eukaryotic sampling (*Richards and Monier, 2016*; *Leger, 2018*). In addition to the 96 OGs potentially acquired through HGT, 2134 of the 9075 total OGs contained non-eukaryotic sequences, in which the Cyanidiales sequences cluster within the eukaryotic kingdom, but are similar enough to non-eukaryotic species to produce blast hits. Based on the average PID, no OG contains HGT

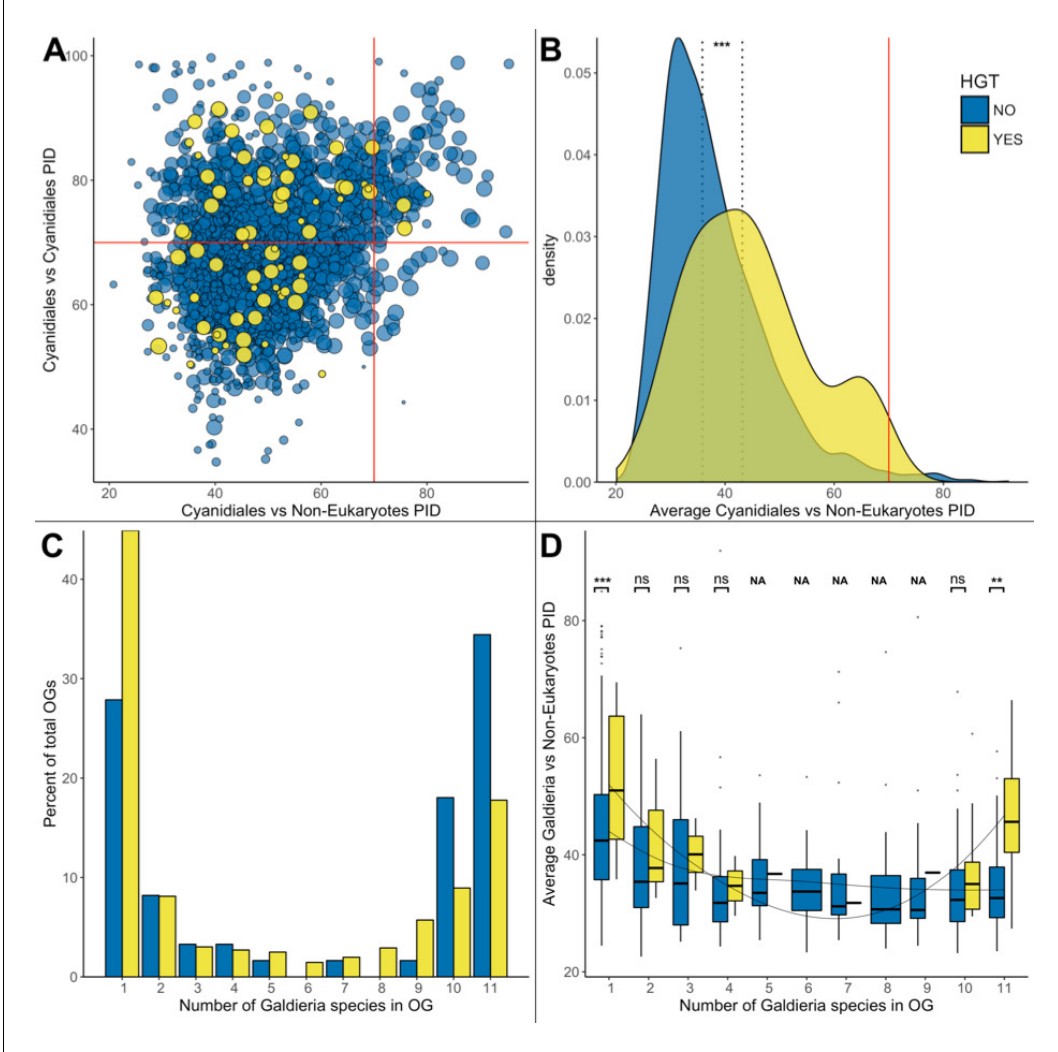

**Figure 6.** HGT vs. non-HGT orthogroup comparisons. **(A)** Maximum PID of Cyanidiales genes in native (blue) and HGT (yellow) orthogroups when compared to non-eukaryotic sequences in each OG. The red lines denote the 70% PID threshold for assembly artifacts according to 'the 70% rule'. Dots located in the top-right corner depict the 73 OGs that appear to contradict this rule, plus the 5 HGT candidates that score higher than 70%. 18/73 of those OGs are not derived from EGT or contamination within eukaryotic assemblies. **(B)** Density curve of average PID towards non-eukaryotic species in the same orthogroup (potential non-eukaryotic donors in case of HGT candidates). The area under each curve is equal to 1. The average PID of HGT candidates (left dotted line) is 6.1% higher than the average PID of native OGs also containing non-eukaryotic species (right dotted line). This difference is significant (Wilcoxon rank-sum test, p>0.01). **(C)** Distribution of OG-sizes (=number of *Galdieria* species present in each OG) between the native and HGT dataset. A total of 80% of the HGT OGs and 89% of the native OGs are present in either ≤10 species, or ≤2 species. Whereas 52.5% of the native gene set is conserved in ≤10 *Galdieria* strains, only 36.1% of the HGT candidates are conserved. In contrast, about 50% of the HGT candidates are present in only one *Galdieria* strain. **(D)** Pairwise OG-size comparison between HGT OGs and native OGs. A significantly higher PID when compared to non-eukaryotic sequences was measured in the HGT OGs at OG-sizes of 1 and 11 (Wilcoxon rank-sum test, BH, p<0.01). No evidence of cumulative effects was detected in the HGT dataset. However, the fewer *Galdieria* species that are contained in one OG, the higher the average PID when compared to non-eukaryotic species in the same tree (Jonckheere-Terpstra, p<0.01) in the native dataset.

DOI: https://doi.org/10.7554/eLife.45017.104

candidates that share over 70% PID to their non-eukaryotic donors with OG0006191 having the highest average PID (69.68%). However, 5/96 HGT-impacted OGs contain one or more individual HGT candidates that exceed this threshold (5.2% of the HGT OGs) (*Figure 6A*). These sequences are found in OG0001929 (75.56% PID, 11 *Galdieria* species), OG0002676 (75.76% PID, 11 *Galdieria* species), OG0006191 (80.00% PID, both *Cyanidioschyzon* species), OG0008680 (72.37% PID, 1 *Galdieria* species), and OG0008822 (71.17% PID, 1 *Galdieria* species). Moreover, we find 73 OGs with eukaryotes as sisters sharing over 70% PID to non-eukaryotic sequences (0.8% of the native OGs) (*Figure 6A*). On closer inspection, the majority are derived from endosymbiotic gene transfer (EGT): 16/73 of the OGs are of proteobacterial descent and 33/73 OGs are phylogenies with gene origin in Cyanobacteria and/or Chlamydia. These annotations generally encompass mitochondrial/plastid components and reactions, as well as components of the phycobilisome, which is exclusive to Cyanobacteria, red algae, and red algal derived plastids. Of the remaining 24 OGs, 18 cannot be explained through EGT or artifacts alone unless multiple eukaryotic genomes would share the same artifact (and also assuming all gene transfers from Cyanobacteria, Chlamydia, and Proteobacteria are derived from EGT). A total of 6/24 OGs are clearly cases of contamination within the eukaryotic assemblies. Although 'the 70% rule' captures a large proportion of the dataset, increasing the sampling resolution within eukaryotes increased the number of exceptions to the rule. This number is likely to increase as more high-quality eukaryote nuclear genomes are determined. Considering the paucity of these data across the eukaryotic tree of life and the rarity of eukaryotic HGT, the systematic dismissal of eukaryotic singletons located within non-eukaryotic branches as assembly/annotation artifacts (or contamination) may come at the cost of removing true positives.

## Cumulative effects

We assessed our dataset for evidence of cumulative effects within the candidate HGT-derived OGs. If cumulative effects were present, then recent HGT candidates would share higher similarity to their non-eukaryotic ancestors than genes resulting from more ancient HGT. Hence, the fewer species that are present in an OG, the higher likelihood of a recent HGT (unless the tree branching pattern contradicts this hypothesis, such as in OG 0005224, which is limited to 3 *Galdieria* species, but is ancient due to its presence in *G. sulphuraria* and *G. phlegrea*). In the case of DL, no cumulative effects as well as no differences between the HGT and native dataset are expected because the PID between eukaryotes and non-eukaryotes is irrelevant to this issue because all genes are native and occurred in the LECA. According to DL, the monophyletic position of Cyanidiales HGT candidates with non-eukaryotes is determined by the absence of other eukaryotic orthologs (given the limited current data) and may be the product of deep branching effects.

First, we tested for general differences in PID with regard to non-eukaryotic sequences between the native and HGT datasets (*Figure 6B*). Neither the PID with non-eukaryotic species in the same OG for the native dataset, nor the PID with potential non-eukaryotic donors in the same OG for the HGT dataset was normally distributed (Shapiro-Wilk normality test, p=2.2e-16/0.00765). Consequently, exploratory analysis was performed using non-parametric testing. On average, the PID with non-eukaryotic species in OGs containing HGT candidates is higher by 6.1% in comparison to OGs with eukaryotic descent. This difference is significant (Wilcoxon rank-sum test, p=0.000008).

Second, we assessed if OGs containing fewer *Galdieria* species would have a higher PID with their potential non-eukaryotic donors in the HGT dataset. We expected a lack of correlation with OG size in the native dataset because the presence/absence pattern of HGT candidates within the *Galdieria* lineage is dictated by gene erosion and thus independent of which non-eukaryotic sequences also cluster in the same phylogeny. Jonckheere's test for trends revealed a significant trend within the native subset: the fewer *Galdieria* species are contained in one OG, the higher the average PID with non-eukaryotic species in the same tree (Jonckheere-Terpstra, p=0.002). This was not the case in the 'HGT' subset. Here, no general trend was observed (Jonckheere-Terpstra, p=0.424).

Third, we compared the PID between HGT-impacted OGs and native OGs of the same size (OGs containing the same number of *Galdieria* species). This analysis revealed a significantly higher PID with non-eukaryotic sequences in favor of the HGT subset in OGs containing either one *Galdieria* sequence, or all 11 *Galdieria* sequences (Wilcoxon rank-sum test, Benjamini-Hochberg, p=2.52e-08| 3.39e-03) (*Figure 6D*). Hence, the 'most recent' and 'most ancient' HGT candidates share the highest identity with their non-eukaryotic donors, which is also significantly higher when compared to native genes in OGs of the same size.

## Natural habitat of extant prokaryotes with closely related orthologs

We next set out to explore the natural habitats of extant prokaryotes that harbor the closest orthologs with candidate HGTs in the Cyanidiales. To this end, we counted the frequency at which any non-eukaryotic species shared monophyly with Cyanidiales (*Table 2*). A total of 568 non-eukaryotic species (19 Archaea, 549 Bacteria), from 365 different genera representing 24 divisions share monophyly with the 96 OGs containing HGT candidates. Most prominent are Proteobacteria that are sister phyla to 53/96 OGs. This group is followed by Firmicutes (28), Actinobacteria (19), Chloroflexi (12), and Bacteroidetes/Chlorobi (10). The only frequently occurring archaeal orthologs were found in Euryarchaeota (6 OGs). Interestingly, the closest orthologs often occurred in extremophilic prokaryotes that share similar (current) habitats with Cyanidiales. We hypothesize that potential non-eukaryotic HGT donors might share similar habitats because proximity is thought to favor HGT. However, we have no direct evidence of what the environment might have been at the time of HGT, or whether a third organism acted as the vector and has not been sampled in our analyses. It is worth noting that the phylogenetic data clearly demonstrate that Cyanidiales have been extremophiles for hundreds

**Table 2.** Natural habitats of extant prokaryotes harboring the closest orthologs to Cyanidiales HGTs.
Numbers in brackets represent how many times HGT candidates from Cyanidiales shared monophyly with non-eukaryotic organisms; for example Proteobacteria were found in 53/96 of the OG monophylies. **Kingdom**: Taxon at kingdom level. **Species**: Scientific species name. **Habitat**: habitat description of the original sampling site. **pH**: pH of the original sampling site. **Temp**: Temperature in Celsius of the sampling site. **Salt**: Ion concentration of the original sampling site. **na**: no information available.

| | Phylogeny | | | Natural habitat of closest non-eukaryotic ortholog | | | |
|---|---|---|---|---|---|---|---|
| Kingdom | Division | Species | Habitat description | pH | Max. temp | Salt |
| Bacteria | Proteobacteria (53) | *Acidithiobacillus thiooxidans (4)* | Mine drainage/Mineral ores | 2.0–2.5 | 30°C | 'hypersaline' |
| | | *Carnimonas nigrificans (4)* | Raw cured meat | 3.0 | 35°C | 8% NaCl |
| | | *Methylosarcina fibrata (4)* | Landfill | 5.0–9.0 | 37°C | 1% NaCl |
| | | *Sphingomonas phyllosphaerae (3)* | Phyllosphere of Acacia caven | na | 28°C | na |
| | | *Gluconacetobacter diazotrophicus (3)* | Symbiont of various plant species | 2.0–6.0 | na | 'high salt' |
| | | *Gluconobacter frateurii (3)* | na | na | na | na |
| | | *Luteibacter yeojuensis (3)* | River | na | na | na |
| | | *Thioalkalivibrio sulfidiphilus (3)* | Soda lake | 8.0–10.5 | 40°C | 15% total salts |
| | | *Thiomonas arsenitoxydans (3)* | Disused mine site | 3.0–8.0 | 30°C | 'halophilic' |
| | Firmicutes (28) | *Sulfobacillus thermosulfidooxidans (6)* | Copper mining | 2.0–2.5 | 45°C | 'salt tolerant' |
| | | *Alicyclobacillus acidoterrestris (4)* | Soil sample | 2.0–6.0 | 53°C | 5% NaCl |
| | | *Gracilibacillus lacisalsi (3)* | Salt lake | 7.2–7.6 | 50°C | 25% total salts |
| | Actinobacteria (19) | *Amycolatopsis halophila (3)* | Salt lake | 6.0–8.0 | 45°C | 15% NaCl |
| | | *Rubrobacter xylanophilus (3)* | Thermal industrial runoff | 6.0–8.0 | 60°C | 6.0% NaCl |
| | Chloroflexi (12) | *Caldilinea aerophila (4)* | Thermophilic granular sludge | 6.0–8.0 | 65°C | 3% NaCl |
| | | *Ardenticatena maritima (3)* | Coastal hydrothermal field | 5.5–8.0 | 70°C | 6% NaCl |
| | | *Ktedonobacter racemifer (3)* | Soil sample | 4.8–6.8 | 33°C | >3% NaCl |
| | Bacteroidetes Chlorobi (10) | *Salinibacter ruber (4)* | Saltern crystallizer ponds | 6.5–8.0 | 52°C | 30% total salts |
| | | *Salisaeta longa (3)* | Experimental mesocosm (Salt) | 6.5–8.5 | 46°C | 20% NaCl |
| | Nitrospirae (7) | *Leptospirillum ferriphilum (4)* | Arsenopyrite biooxidation tank | 0–3.0 | 40°C | 2% NaCl |
| | Fibrobacteres (6) | *Acidobacteriaceae bacterium TAA166 (3)* | na | na | na | na |
| | Deinococcus (5) | *Truepera radiovictrix (3)* | Hot spring runoffs | 7.5–9.5 | na | 6% NaCl |
| Archaea | Euryarchaeota (6) | *Ferroplasma acidarmanus (3)* | Acid mine drainage | 0–2.5 | 40°C | 'halophilic' |

DOI: https://doi.org/10.7554/eLife.45017.105

of millions of years. It is however conceivable that the HGTs may have occurred when these cells were being dispersed (they have a worldwide distribution) from one extreme site to another and would have encountered mesophilic donors at these times. Given these caveats, it is interesting to note that *Sulfobacillus thermosulfidooxidans* (Firmicutes), a mixotrophic, acidophilic (pH 2.0), and moderately thermophilic (45°C) bacterium that was isolated from acid mining environments in northern Chile (where *Galdieria* is also present) was most prominent amongst the prokaryotic orthogroups. *Sulfobacillus thermosulfidooxidans* shares monophyly in 6/96 HGT-derived OGs and is followed in frequency by several species that are either thermophiles, acidophiles, or halophiles and share habitats in common with Cyanidiales (*Table 2*).

## Functions of horizontally acquired genes in cyanidiales

We analyzed the putative molecular functions and processes acquired through HGT. Annotations were curated using information gathered from blast, GO-terms, PFAM, KEGG, and EC. A total of 72 GO annotations occurred more than once within the 96 HGT-impacted OGs. Furthermore, 37/72 GO annotations are significantly enriched (categorical data, 'native' vs 'HGT', Fisher's exact test, Benjamini-Hochberg, p≤0.05). The most frequent terms were: 'decanoate-CoA ligase activity' (5/72 GOs, p=0), 'oxidation-reduction process' (16/72 GOs, p=0.001), 'transferase activity' (14/72 GOs, p=0.009), 'carbohydrate metabolic process' (5/72 GOs, p=0.01), 'oxidoreductase activity' (9/72 GOs, p=0.012), 'methylation '(6/72 GOs, p=0.013), 'methyltransferase activity' (5/72 GOs, p=0.023), 'transmembrane transporter activity' (4/72 GOs, p=0.043), and 'hydrolase activity' (9/72 GOs, p=0.048). In comparison to previous studies, our analysis did not report a significant enrichment of membrane proteins in the HGT dataset ('membrane', 11/72 OGs, p=0.699; 'integral component of membrane', 22/72 GOs, p=0.416. The GO annotation 'extracellular region' was absent in the HGT dataset) (*Schönknecht et al., 2013*). As such, we report a strong bias for metabolic functions among HGT candidates (*Figure 7*).

## Metal and xenobiotic resistance/detoxification

Geothermal environments often contain high arsenic (Ar) concentrations, up to a several g/L as well as high levels of mercury (Hg), such as >200 μg/g in soils of the Norris Geyser Basin (Yellowstone National Park) and volcanic waters in southern Italy (*Stauffer and Thompson, 1984*; *Aiuppa, 2003*), both known Cyanidiales habitats (*Castenholz and McDermott, 2010*; *Ciniglia et al., 2004*; *Toplin et al., 2008*; *Pinto, 1975*). Studies with *G. sulphuraria* have shown an increased efficiency and speed regarding the biotransformation of $HgCl_2$ compared to eukaryotic algae (*Kelly et al., 2007*). Orthologs of OG0002305, which are present in all 13 Cyanidiales genomes, encode mercuric reductase that catalyzes the critical step in $Hg^{2+}$ detoxification, converting cytotoxic $Hg^{2+}$ into the less toxic metallic mercury, $Hg^0$. Arsenate (As(V)) is imported into the cell by high-affinity $P_i$ transport systems (*Meharg and MacNair, 1992*; *Catarecha et al., 2007*), whereas aquaporins regulate arsenite (As(III)) uptake (*Zhao et al., 2010*). *Galdieria* and *Cyanidioschyzon* possess a eukaryotic gene-set for the chemical detoxification and extrusion of As through biotransformation and direct efflux (*Schönknecht et al., 2013*). Arsenic tolerance was expanded in the *Galdieria* lineage through the acquisition (OG0001513) of a bacterial **arsC** gene, thus enabling the reduction of As(V) to As(III) using thioredoxin as the electron acceptor. It is known that As(III) can be converted into volatile dimethylarsine and trimethylarsine through a series of reactions, exported, or transported to the vacuole in conjugation with glutathione. Two separate acquisitions of a transporter annotated as ArsB are present in *G. sulphuraria* RT22 and *G. sulphuraria* 5572 (OG0006498, OG0006670), as well as a putative cytoplasmic heavy metal binding protein (OG0006191) in the *Cyanidioschyzon* lineage.

In the context of xenobiotic detoxification, we found an aliphatic nitrilase (OG0001760) involved in styrene degradation and three (OG0003250, OG0005087, OG0005479) *Galdieria* specific 4-nitrophenylphosphatases likely involved in the bioremediation of highly toxic hexachlorocyclohexane (HCH) (*van Doesburg et al., 2005*), or more generally other cyclohexyl compounds, such as cyclohexylamine. In this case, bioremediation can be achieved through the hydrolysis of 4-nitrophenol to 4-nitrophenyl phosphate coupled with phosphoesterase/metallophosphatase activity. The resulting cyclohexyl compounds serve as multifunctional intermediates in the biosynthesis of various heterocyclic and aromatic metabolites. A similar function in the *Cyanidioschyzon* lineage could be taken up by OG0006252, a cyclohexanone monooxygenase (*Chen et al., 1988*) oxidizing phenylacetone to

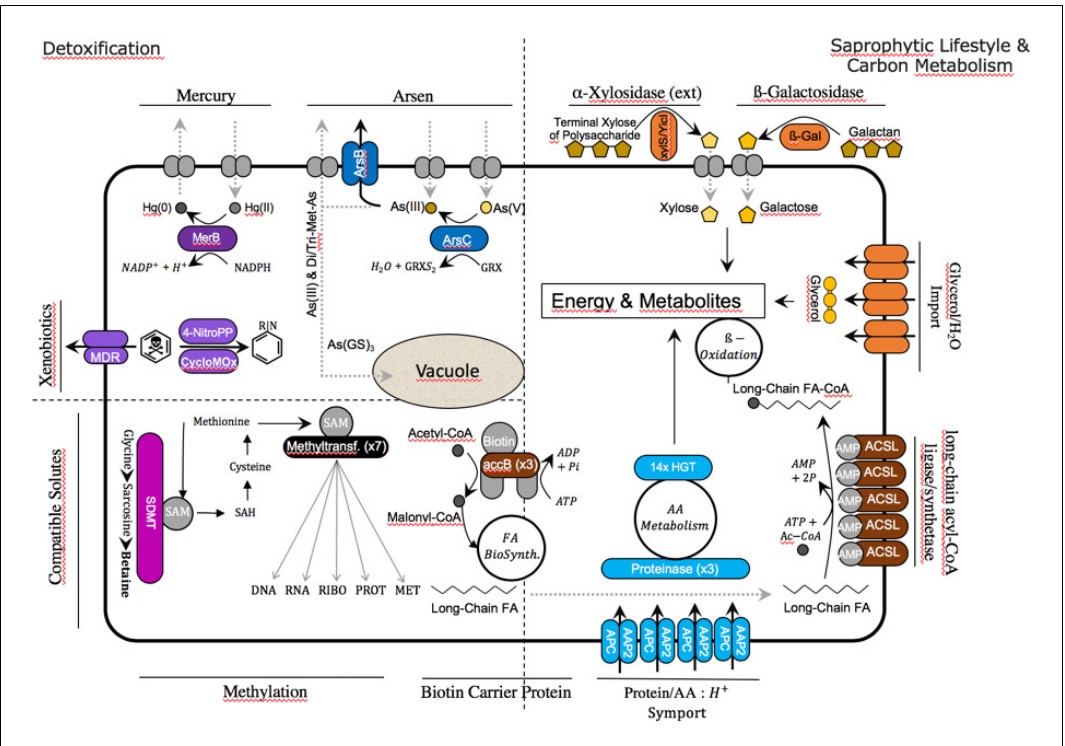

**Figure 7.** Cyanidiales live in hostile habitats, necessitating a broad range of adaptations to polyextremophily. The majority of the 96 HGT-impacted OGs were annotated and putative functions identified (in the image, colored fields are from HGT, whereas gray fields are native functions). The largest number of HGT candidates is involved in carbon and amino acid metabolism, especially in the *Galdieria* lineage. The excretion of lytic enzymes and the high number of importers (protein/AA symporter, glycerol/H2O symporter) within the HGT dataset suggest a preference for import and catabolic function.

DOI: https://doi.org/10.7554/eLife.45017.106

benzyl acetate that can also oxidize various aromatic ketones, aliphatic ketones (e.g., dodecan- 2-one) and sulfides (e.g., 1-methyl-4-(methylsulfanyl)benzene). In this context, a probable multidrug-resistance/quaternary ammonium compound exporter (OG0002896), which is present in all Cyanidiales, may control relevant efflux functions whereas a phosphatidylethanolamine (penicillin?) binding protein (OG0004486) could increase the stability of altered peptidoglycan cell walls. If these annotations are correct, then *Galdieria* is an even more promising target for industrial bioremediation applications than previously thought (*Henkanatte-Gedera et al., 2017*; *Fukuda et al., 2018*).

## Cellular oxidant reduction

Increased temperature leads to a higher metabolic rate and an increase in the production of endogenous free radicals (FR), such as reactive oxygen species (ROS) and reactive nitrogen species (RNS), for example during cellular respiration (*Phaniendra et al., 2015*). Furthermore, heavy metals such as lead and mercury, as well as halogens (fluorine, chlorine, bromine, iodine) stimulate formation of FR (*Dietz et al., 1999*). FR are highly biohazard and cause damage to lipids (*Ylä-Herttuala, 1999*), proteins (*Stadtman and Levine, 2000*) and DNA (*Marnett, 2000*). In the case of the superoxide radical ($^{\bullet}O^{2-}$), enzymes such as superoxide dismutase enhance the conversion of 2 x $^{\bullet}O^{2-}$, into hydrogen peroxide ($H_2O_2$) which is in turn reduced to $H_2O$ through the glutathione-ascorbate cycle. Other toxic hydroperoxides (R-OOH) can be decomposed various peroxidases to $H_2O$ and alcohols (R-OH) at the cost of oxidizing the enzyme, which is later recycled (re-reduced) through oxidation of thioredoxin (*Rouhier et al., 2008*). The glutathione and thioredoxin pools and their related enzymes are thus factors contributing to a successful adaptation to geothermal environments. Here, we found a cytosolic and/or extracellular peroxiredoxin-6 (OG0005984) specific to the *Cyanidioschyzon* lineage and two peroxidase-related enzymes (probable alkyl hydroperoxide reductases acting on

carboxymuconolactone) in the *Galdieria* lineage (OG0004203, OG0004392) (*Chae et al., 1994*). In addition, a thioredoxin oxidoreductase related to alkyl hydroperoxide reductases (OG0001486) as well as a putative glutathione-specific gamma-glutamylcyclotransferase 2 (OG0003929) are present in all Cyanidiales. The latter has been experimentally linked to the process of heavy metal detoxification in *Arabidopsis thaliana* (*Paulose et al., 2013*).

## Carbon metabolism

*G. sulphuraria* is able to grow heterotrophically using a large variety of different carbon sources and compounds released from dying cells (*Gross et al., 1995*; *Gross, 1998*). In contrast, *C. merolae* is strictly photoautotrophic (*De Luca et al., 1978*). *G. sulphuraria* can be maintained on glycerol as the sole carbon source (*Gross et al., 1995*) making use of a family of glycerol uptake transporters likely acquired via HGT (*Schönknecht et al., 2013*). We confirm the lateral acquisition of glycerol transporters in *G. sulphuraria* RT22 (OG0006482), *G. sulphuraria* Azora and *G. sulphuraria* SAG21 (OG0005235). The putative HGT glycerol transporters found in *G. sulphuraria* 074W did not meet the required threshold of two Cyanidiales sequences (from different strains) in one OG. In addition, another MIP family aquaporin, permeable to $H_2O$, glycerol and other small uncharged molecules (*Liu et al., 2007*) is encoded by *G. sulphuraria* Azora (OG0007123). This could be an indication of a very diverse horizontal acquisition pattern regarding transporters. OG0003954 is the only exception to this rule, because it is present in all *Galdieria* lineages and is orthologous to AcpA|SatP acetate permeases involved with the uptake of acetate and succinate (*Robellet et al., 2008*; *Sá-Pessoa et al., 2013*).

We found evidence of saprophytic adaptations in *Galdieria* through the potential horizontal acquisition of an extracellular beta-galactosidase enzyme (*Rojas et al., 2004*; *Rico-Díaz et al., 2014*). This enzyme contains all five bacterial beta-galactosidase domains (OG0003441) involved in the catabolism of glycosaminoglycans, a polysaccharide deacetylase/peptidoglycan-N-acetylglucosamine deacetylase (OG0004030) acting on glucosidic (but note peptide bonds) that may degrade chitooligosaccharides, chitin, and/or xylan (*Psylinakis et al., 2005*; *Lee et al., 2002*) as well as an α-amylase (OG0004658) converting starch/glycogen to dextrin/maltose (*Diderichsen and Christiansen, 1988*) which is missing only in *G. sulphuraria* SAG21. All other HGT OGs involved in sugar metabolism are involved in the intercellular breakdown and interconversions of sugar carbohydrates. OG0006623 contains a non-phosphorylating glyceraldehyde-3-phosphate dehydrogenase found in hyperthermophile archaea (*Ettema et al., 2008*) (*G. sulphuraria* 002). The OG0005153 encodes a glycosyl transferase family one protein involved in carbon metabolism (*G. sulphuraria* 074W, *G. sulphuraria* MS1, *G. sulphuraria* RT22, *G. sulphuraria* YNP5587.1). All *Galdieria* have an alpha-xylosidase resembling an extremely thermo-active and thermostable α-galactosidase (OG0001542) (*van Lieshout et al., 2003*; *Okuyama et al., 2004*). The only horizontal acquisition in this category present in all Cyanidiales is a cytoplasmic ribokinase involved in the D-ribose catabolic process (OG0001613).

The irreversible synthesis of malonyl-CoA from acetyl-CoA through acetyl-CoA carboxylase (ACCase) is the rate limiting and step in fatty acid biosynthesis. The bacterial ACCase complex consists of three separate subunits, whereas the eukaryotic ACCase is composed of a single multifunctional protein. Plants contain both ACCase isozymes. The eukaryotic enzyme is located in the cytosol and a bacterial-type enzyme consisting of four subunits is plastid localized. Three of the HGT orthogroups (OG0002051, OG0007550 and OG0007551) were annotated as bacterial biotin carboxyl carrier proteins (AbbB/BCCP), which carry biotin and carboxybiotin during the critical and highly regulated carboxylation of acetyl-CoA to form malonyl-CoA [ATP +Acetyl CoA + $HCO^{3-} \rightleftharpoons$ ADP + Orthophosphate + Malonyl-CoA]. Whereas OG0002051 is present in all Cyanidiales and located in the cytoplasm, OG0007550 and OG0007551 are unique to *C. merolae* Soos and annotated as 'chloroplastic'. Prior to fatty acid (FA) beta-oxidation, FAs need to be transformed to a FA-CoA before entering cellular metabolism as an exogenous or endogenous carbon source (eicosanoid metabolism is the exception). This process is initiated by long-chain-fatty-acid-CoA ligases/acyl-CoA synthetases (ACSL) (*Mashek et al., 2007*) [ATP + long-chain carboxylate + CoA $\rightleftharpoons$ AMP + diphosphate + Acyl-CoA]. Five general non-eukaryotic ACSL candidates were found (OG0001476, OG0002999, OG0005540, OG0008579, OG0008822). Only OG0001476 is present in all species, whereas OG0002999 is present in all *Galdieria,* OG0005540 in *G. sulphuraria* 074W and *G.*

*sulphuraria* MS1, and OG0008579 and OG0008822 are unique to *G. phlegrea* DBV009. The GO annotation suggests moderate specificity to decanoate-CoA. However, OG0002999 also indicates involvement in the metabolism of linoleic acid, a $C_{18}H_{32}O_2$ polyunsaturated acid found in plant glycosides. ACSL enzymes share significant sequence identity but show partially overlapping substrate preferences in terms of length and saturation as well as unique transcription patterns. Furthermore, ACSL proteins play a role in channeling FA degradation to various pathways, as well as enhancing FA uptake and FA cellular retention. Although an annotation of the different ACSL to their specific functions was not possible, their involvement in the saprophytic adaptation of *Cyanidioschyzon* and especially *Galdieria* appears to be plausible.

## Amino acid metabolism

Oxidation of amino acids (AA) can be used as an energy source. Once AAs are deaminated, the resulting α-ketoacids ('carbon backbone') can be used in the tricarboxylic acid cycle for energy generation, whereas the remaining $NH_4^+$ can be used for the biosynthesis of novel AAs, nucleotides, and ammonium containing compounds, or dissipated through the urea cycle. In this context, we confirm previous observations regarding a horizontal origin of the urease accessory protein UreE (OG0003777) present in the *Galdieria* lineage (*Qiu et al., 2013*) (the other urease genes reported in *G. phlegrea* DBV009 appear to be unique to this species and were thus removed from this analysis as singletons; for example *ureG*, OG0008984). AAs are continuously synthesized, interconverted, and degraded using a complex network of balanced enzymatic reactions (e.g., peptidases, lyases, transferases, isomerases). Plants maintain a functioning AA catabolism that is primarily used for the interconversion of metabolites because photosynthesis is the primary source of energy. The Cyanidiales, and particularly the *Galdieria* lineage is known for its heterotrophic lifestyle. We assigned 19/96 HGT-impacted OGs to this category. In this context, horizontal acquisition of protein|AA:proton symporter AA permeases (OG0001658, OG0005224, OG0005596, OG0007051) may be the first indication of adaptation to a heterotrophic lifestyle in *Galdieria*. Once a protein is imported, peptidases cleave single AAs by hydrolyzing the peptide bonds. Although no AA permeases were found in the *Cyanidioschyzon* lineage, a cytoplasmic threonine-type endopeptidase (OG0001994) and a cytosolic proline iminopeptidase involved in arginine and proline metabolism (OG0006143) were potentially acquired through HGT. At the same time, the *Galdieria* lineage acquired a Clp protease (OG0007596). The remaining HGT candidates are involved in various amino acid metabolic pathways. The first subset is shared by all Cyanidiales, such as a cytoplasmic imidazoleglycerol-phosphate synthase involved in the biosynthetic process of histidine (OG0002036), a phosphoribosyltransferase involved in phenylalanine/tryptophan/tyrosine biosynthesis (OG0001509) and a peptydilproline peptidyl-prolyl cis-trans isomerase acting on proline (OG0001938) (*Dilworth et al., 2012*). The second subset is specific to the *Cyanidium* lineage. It contains a glutamine/leucine/phenylalanine/valine dehydrogenase (OG0006136) (*Kloosterman et al., 2006*), a glutamine cyclotransferase (OG0006251) (*Dahl et al., 2000*), a cytidine deaminase (OG0003539) as well as an adenine deaminase (OG0005683) and a protein binding hydrolase containing a NUDIX domain (OG0005694). The third subset is specific to the *Galdieria* lineage and contains an ornithine deaminase, a glutaryl-CoA dehydrogenase (OG0007383) involved in the oxidation of lysine, tryptophan, and hydroxylysine (*Rao et al., 2006*), as well as an ornithine cyclodeaminase (OG0004258) involved in arginine and/or proline metabolism. Finally, a lysine decarboxylase (OG0007346), a bifunctional ornithine acetyltransferase/N-acetylglutamate synthase (*Martin and Mulks, 1992*) involved in the arginine biosynthesis (OG0008898) and an aminoacetone oxidase family FAD-binding enzyme (OG0007383), probably catalytic activity against several different L-amino acids were found as unique acquisitions in *G. sulphuraria* SAG21, *G. phlegrea* DBV009 and *G. sulphuraria* YNP5587.1 respectively.

## One carbon metabolism and methylation

One-carbon (1C) metabolism based on folate describes a broad set of reactions involved in the activation and transfer C1 units in various processes including the synthesis of purine, thymidine, methionine, and homocysteine re-methylation. C1 units can be mobilized using tetrahydrofolate (THF) as a cofactor in enzymatic reactions, vitamin B12 (cobalamin) as a co-enzyme in methylation/rearrangement reactions and S-adenosylmethionine (SAM) (*Ducker and Rabinowitz, 2017*). In terms of purine biosynthesis, OG0005280 encodes an ortholog of a bacterial FAD-dependent thymidylate (dTMP)

synthase converting dUMP to dTMP by oxidizing THF present in *G. sulphuraria* 074W, *G. sulphuraria* MS1, and *G. sulphuraria* RT22. In terms of vitamin B12 biosynthesis, an ortholog of the cobalamin biosynthesis protein CobW was found in the *Cyanidioschyzon* lineage (OG0002609). Much of the methionine generated through C1 metabolism is converted to SAM, the second most abundant cofactor after ATP, which is a universal donor of methyl (-CH$_3$) groups in the synthesis and modification of DNA, RNA, hormones, neurotransmitters, membrane lipids, proteins and also play a central role in epigenetics and posttranslational modifications. Within the 96 HGT-impacted dataset we found a total of 9 methyltransferases (OG0003901, OG0003905, OG0002191, OG0002431, OG0002727, OG0003907, OG0005083 and OG0005561) with diverse functions, 8 of which are SAM-dependent methyltransferases. OG0002431 (Cyanidiales), OG0005561 (*G. sulphuraria* MS1 and *G. phlegrea* DBV009) and OG0005083 (*G. sulphuraria* SAG21) encompass rather unspecific SAM-dependent methyltransferases with a broad range of possible methylation targets. OG0002727, which is exclusive to *Cyanidioschyzon*, and OG0002191, which is exclusive to *Galdieria*, both methylate rRNA. OG0002727 belongs to the Erm rRNA methyltransferase family that methylate adenine on 23S ribosomal RNA (*Yu et al., 1997*). Whether it confers macrolide-lincosamide-streptogramin (MLS) resistance, or shares only adenine methylating properties remains unclear. The OG0002191 is a 16S rRNA (cytidine1402-2'-O)-methyltransferase involved the modulation of translational fidelity (*Kimura and Suzuki, 2010*).

## Osmotic resistance and salt tolerance

Cyanidiales withstand salt concentrations up to 10% NaCl (*Albertano, 2000*). The two main strategies to prevent the accumulation of cytotoxic salt concentrations and to withstand low water potential are the active removal of salt from the cytosol and the production of compatible solutes. Compatible solutes are small metabolites that can accumulate to very high concentrations in the cytosol without negatively affecting vital cell functions while keeping the water potential more negative in relation to the saline environment, thereby avoiding loss of water. The *G. sulphuraria* lineage produces glycine/betaine as compatible solutes under salt stress in the same manner as halophilic bacteria (*Imhoff and Rodriguez-Valera, 1984*) through the successive methylation of glycine via sarcosine and dimethylglycine to yield betaine using S-adenosyl methionine (SAM) as a cofactor (*Lu et al., 2006*; *Waditee et al., 2003*; *Nyyssola et al., 2000*). This reaction is catalyzed by the enzyme sarcosine dimethylglycine methyltransferase (SDMT), which has already been characterized in *Galdieria* (*McCoy et al., 2009*). Our results corroborate the HGT origin of this gene, supporting two separate acquisitions of this function (OG0003901, OG0003905). In this context, a inositol 2-dehydrogenase possibly involved in osmoprotective functions (*Kingston et al., 1996*) in *G. phlegrea* DBV009 was also found in the HGT dataset (OG0008335).

## Non-Metabolic functions

Outside the context of HGT involving enzymes that perform metabolism related functions, we found 6/96 OGs that are annotated as transcription factors, ribosomal components, rRNA, or fulfilling functions not directly involved in metabolic fluxes. Specifically, two OGs associated with the bacterial 30S ribosomal subunit were found, whereas OG0002627 (*Galdieria*) is orthologous to the tRNA binding translation initiation factor eIF1a which binds the fMet-tRNA(fMet) start site to the ribosomal 30S subunit and defines the reading frame for mRNA translation (*Simonetti et al., 2009*), and OG0004339 (*Galdieria*) encodes the S4 structural component of the S30 subunit. Three genes functioning as regulators were found in *Cyanidioschyzon*, a low molecular weight phosphotyrosine protein phosphatase with an unknown regulator function (OG0002785), a SfsA nuclease (*Takeda et al., 2001*), similar to the sugar fermentation stimulation protein A and (OG0002871) a MRP family multidrug resistance transporter connected to parA plasmid partition protein, or generally involved in chromosome partitioning (mrp). Additionally, we found a *Cyanidioschyzon*-specific RuvX ortholog (OG0002578) involved in chromosomal crossovers with endonucleolytic activity (*Nautiyal et al., 2016*) as well as a likely Hsp20 heat shock protein ortholog (OG0004102) unique to the *Galdieria* lineage.

## Various functions and uncertain annotations

The remaining OGs were annotated with a broad variety of functions. For example, OG0001929, OG0001810, OG0004405, and OG0001087 are possibly connected to the metabolism of cell wall precursors and components and OG0001929 (*Galdieria*) is an isomerizing glutamine-fructose-6-phosphate transaminase most likely involved in regulating the availability of precursors for N- and O-linked glycosylation of proteins, such as for peptidoglycan. In contrast, OG0004405 (*Cyanidioschyzon*) synthesizes exopolysaccharides on the plasma membrane and OG0001087 (*Cyanidiales*) and OG0001810 (*Cyanidioschyzon*) are putative undecaprenyl transferases (UPP) which function as lipid carrier for glycosyl transfer in the biosynthesis of cell wall polysaccharide components in bacteria (*Apfel et al., 1999*). The OGs OG0002483 and OG0001955 are involved in purine nucleobase metabolic processes, probably in cAMP biosynthesis (*Galperin, 2005*) and IMP biosynthesis (*Schrimsher et al., 1986*). A *Cyanidioschyzon* specific 9,15,9'-tri-cis-zeta-carotene isomerase (OG0002574) may be involved in the biosynthesis of carotene (*Chen et al., 2010*). Two of the 96 HGT OGs obtained the tag 'hypothetical protein' and could not be further annotated. Others had non-specific annotations, such as 'selenium binding protein' (OG0003856) or contained conflicting annotations.

## Discussion

Making an argument for the importance of HGT in eukaryote (specifically, Cyanidiales) evolution, as we do here, requires that three major issues are addressed: a mechanism for foreign gene uptake and integration, the apparent absence of eukaryotic pan-genomes, and the lack of evidence for cumulative effects (*Martin, 2017*). The latter two arguments are dealt with below but the first concern no longer exists. For example, recent work has shown that red algae harbor naturally occurring plasmids, regions of which are integrated into the plastid DNA of a taxonomically wide array of species (*Lee et al., 2016*). Genetic transformation of the unicellular red alga *Porphyridium purpureum* has demonstrated that introduced plasmids accumulate episomally in the nucleus and are recognized and replicated by the eukaryotic DNA synthesis machinery (*Li and Bock, 2018*). These results suggest that a connection can be made between the observation of bacterium-derived HGTs in *P. purpureum* (*Bhattacharya et al., 2013*) and a putative mechanism of bacterial gene origin *via* long-term plasmid maintenance. Other proposed mechanisms for the uptake and integration of foreign DNA in eukaryotes are well-studied, observed in nature, and can be successfully recreated in the lab (*Leger, 2018*; *Li and Bock, 2018*).

## HGT- the eukaryotic pan-genome

Eukaryotic HGT is rare and affected by gene erosion. Within the 13 analyzed genomes of the polyextremophilic Cyanidiales (*Foflonker et al., 2018*; *Schönknecht et al., 2014*), we identified and annotated 96 OGs containing 641 single HGT candidates. Given an approximate age of 1,400 Ma years and ignoring gene erosion, on average, one HGT event occurs every 14.6 Ma years in Cyanidiales. This figure ranges from one HGT every 33.3 Ma years in *Cyanidioschyzon* and one HGT every 13.3 Ma in *Galdieria*. Still, one may ask, given that eukaryotic HGT exists, what comprises the eukaryotic pan-genome and why does it not increase in size as a function of time due to HGT accumulation? In response, it should be noted that evolution is 'blind' to the sources of genes and selection does not act upon native genes in a manner different from those derived from HGT. In our study, we report examples of genes derived from HGT that are affected by gene erosion and/or partial fixation (*Figure 4A*). As such, only 8/96 of the HGT-impacted OGs (8.3%) are encoded by all 13 Cyanidiales species. Looking at the *Galdieria* lineage alone (*Figure 6C*), 28 of the 60 lineage-specific OGs (47.5%) show clear signs of erosion (HGT orthologs are present in ≤10 *Galdieria* species), to the point where a single ortholog of an ancient HGT event may remain.

When considering HGT in the Cyanidiales it is important to keep in mind the ecological boundaries of this group, the distance between habitats, the species composition of habitats, and the mobility of Cyanidiales within those borders that control HGT. Hence, we would not expect the same HGT candidates derived from the same non-eukaryotic donors to be shared between Cyanidiales and marine/freshwater red algae (unless they predate the split between Cyanidiales and other red algae), but rather between Cyanidiales and other polyextremophilic organisms. In this context,

inspection of the habitats and physiology of potential HGT donors revealed that the vast majority is extremophilic and, in some cases, shares the same habitat as Cyanidiales (*Table 2*). A total of 84/96 of the inherited gene functions could be connected to ecologically important traits such as heavy metal detoxification, xenobiotic detoxification, ROS scavenging, and metabolic functions related to carbon, fatty acid, and amino acid turnover. In contrast, only 6/96 OGs are related to methylation and ribosomal functions. We did not find HGTs contributing other traits such as ultrastructure, development, or behavior (*Figure 7*). If cultures were exposed to abiotic stress, the HGT candidates were significantly enriched within the set of differentially expressed genes (*Figure 3*). These results not only provide evidence of successful integration into the transcriptional circuit of the host, but also support an adaptive role of HGT as a mechanism to acquire beneficial traits. Because eukaryotic HGT is the exception rather than the rule, its number in eukaryotic genomes does not need to increase as a function of time and may have reached equilibrium in the distant past between acquisition and erosion.

## HGT vs. DL

Ignoring the cumulative evidence from this and many other studies, one may still dismiss the phylogenetic inference as mere assembly artefact and overlook all the significant differences and trends between native genes and HGT candidates. This could be done by superimposing vertical inheritance (and thus eukaryotic origin) on all HGT events outside the context of pathogenicity and endosymbiosis. Under this extreme view, all extant genes would have their roots in LECA. Consequently, patchy phylogenetic distributions are the result of multiple putative ancient paralogs existing in the LECA followed by mutation, gene duplication, and gene loss. Following this line of reasoning, all HGT candidates in the Cyanidiales would be the product of DL acting on all other eukaryotic species, with the exception of the Cyanidiales, *Galdieria* and/or *Cyanidioschyzon* (*Figure 5A–C*). However, we found cases where either *Galdieria* HGT candidates (six orthogroups), or *Cyanidioschyzon* HGT candidates (eight orthogroups) show non-eukaryotic origin, whereas the others cluster within the eukaryotic branch (*Figure 5E–F*). In addition, we find two cases in which *Galdieria* and *Cyanidioschyzon* HGT candidates are located in different non-eukaryotic branches (*Figure 5D*). DL would require LECA to have encoded three paralogs of the same gene, one of which was retained by *Cyanidioschyzon*, another by *Galdieria*, whereas the third by all other eukaryotes. The number of required paralogs in the LECA would be further increased when taking into consideration that some ancient paralogs of LECA may have been eroded in all eukaryotes and that eukaryote phylogenies are not always monophyletic which would additionally increase the number of required paralogs in the LECA in order to explain the current pattern. The strict superimposition of vertical inheritance would thus require a complex LECA, an issue known as 'the genome of Eden'.

Cumulative effects are observed when genes derived from HGT increasingly diverge as a function of time. Hence, a gradual increase in protein identity towards their non-eukaryotic donor species is expected the more recent an individual HGT event is. The absence of cumulative effects in eukaryotic HGT studies has been used as argument in favor of strict vertical inheritance followed by DL. Here, we also did not find evidence for cumulative effects in the HGT dataset. 'Recent' HGT events that are exclusive to either the *Cyanidioschyzon* or *Galdieria* lineage shared 5% higher PID with their potential non-eukaryotic donors in comparison to ancient HGT candidates that predate the split, but this difference was not significant (*Figure 4C*). We also tested for cumulative effects between the number of species contained in orthogroups compared to the percent protein identity shared with potential non-eukaryotic donors under the assumption that recent HGT events would be present in fewer species in comparison to ancient HGT events that occurred at the root of *Galdieria* (*Figure 6D*). Neither a gradual increase in protein identity for potentially recent HGT events, nor a general trend could be determined. Only orthogroups containing one *Galdieria* species reported a statistically significant higher protein identity to their potential non-eukaryotic donors which could be an indication of 'most recent' HGT.

Whereas the absence of cumulative effects may speak against HGT, this does not automatically argue in favor of strict vertical inheritance followed by DL. Here, the null hypothesis would be that no differences exist between HGT genes and native genes because all genes are descendants of LECA. This null hypothesis is rejected on multiple levels. At the molecular level, the HGT subset differs significantly from native genes with respect to various genomic and molecular features (e.g., GC-content, frequency of multiexonic genes, number of exons per gene, responsiveness to

temperature stress) (*Table 1*, *Figure 3*). Furthermore, HGT candidates in *Galdieria* are significantly more similar (6.1% average PID) to their potential non-eukaryotic donors when compared to native genes and non-eukaryotic sequences in the same orthogroup (*Figure 6B*). This difference cannot be explained by the absence of eukaryotic orthologs. We also find significant differences in PID with regard to non-eukaryotic sequences between HGT and native genes in orthogroups containing either one *Galdieria* sequence, or all eleven *Galdieria* sequences regarding (*Figure 6D*). Hence, the 'most recent' and 'most ancient' HGT candidates share the highest resemblance to their non-eukaryotic donors, which is also significantly higher when compared to native genes in OGs of the same size. Intriguingly, a general trend towards 'cumulative effects' could be observed for native genes, highlighting the differences between these two gene sources in Cyanidiales.

Given these results and interpretations, we advocate the following view of eukaryotic HGT. Specifically, two forces may act simultaneously on HGT candidates in eukaryotes. The first is strong evolutionary pressure for adaptation of eukaryotic genetic features and compatibility with native replication and transcriptional mechanisms to ensure integration into existing metabolic circuits (e. g., codon usage, splice sites, methylation, pH differences in the cytosol). The second however is that key structural aspects of HGT-derived sequence cannot be significantly altered by the first process because they ensure function of the transferred gene (e.g., protein domain conservation, three-dimensional structure, ligand interaction). Consequently, HGT candidates may suffer more markedly from gene erosion than native genes due to these countervailing forces, in spite of potentially providing beneficial adaptive traits. This view suggests that we need to think about eukaryotic HGT in fundamentally different ways than is the case for prokaryotes, necessitating a taxonomically broad genome-based approach that is slowly taking hold.

In summary, we do not discount the importance of DL in eukaryotic evolution because it can impact ca. 99% of the gene inventory in Cyanidiales. What we strongly espouse is that strict vertical inheritance in combination with DL cannot explain all the data. HGTs in Cyanidiales are significant because the 1% (values will vary across different eukaryotic lineages) helps explain the remarkable evolutionary history of these extremophiles. Lastly, we question the validity of the premise regarding the applicability of cumulative effects in the prokaryotic sense to eukaryotic HGT. The absence of cumulative effects and a eukaryotic pan-genome are neither arguments in favor of HGT, nor DL.

## Materials and methods

### Cyanidiales strains used for draft genomic sequencing

Ten Cyanidiales strains (*Figure 1*) were sequenced in 2016/2017 using the PacBio RS2 (Pacific Biosciences Inc, Menlo Park, CA) technology (*Rhoads and Au, 2015*) and P6-C4 chemistry (the only exception being *C. merolae* Soos, which was sequenced as a pilot study using P4-C2 chemistry in 2014). Seven strains, namely *G. sulphuraria* 5572, *G. sulphuraria* 002, *G. sulphuraria* SAG21.92, *G. sulphuraria* Azora, *G. sulphuraria* MtSh, *G. sulphuraria* RT22 and *G. sulphuraria* MS1 were sequenced at the University of Maryland Institute for Genome Sciences (Baltimore, MD). The remaining three strains, *G. sulphuraria* YNP5587.1, *G. phlegrea* Soos, and *C. merolae* Soos were sequenced at the Max-Planck-Institut für Pflanzenzüchtungsforschung (Cologne, Germany). To obtain axenic and monoclonal genetic material for sequencing, single colonies of each strain were grown at 37°C in the dark on plates containing glucose as the sole carbon source (1% Gelrite mixed 1:1 with 2x Allen medium [*Allen, 1959*], 50 µM Glucose). The purity of single colonies was assessed using microscopy (Zeiss Axio Imager 2, 1000x) and molecular markers (18S, *rbcL*). Long-read compatible DNA was extracted using a genomic-tip 20/G column following the steps of the 'YEAST' DNA extraction protocol (QIAGEN N.V., Hilden, Germany). The size and quality of DNA were assessed via gel electrophoresis and the Nanodrop instrument (Thermo Fisher Scientific Inc, Waltham, MA).

### Assembly

All genomes (excluding the already published *G. sulphuraria* 074W, *G. phlegrea* DBV009 and *C. merolae* 10D) were assembled using canu version 1.5 (*Koren et al., 2017*). The genomic sequences were polished three times using the Quiver algorithm (*Chin et al., 2013*). Different versions of each genome were assessed using BUSCO v.3 (*Simão et al., 2015*) and the best performing genome was chosen as reference for gene model prediction. Each genome was queried against the National

Center for Biotechnology Information (NCBI) nr database (*Geer et al., 2010*) in order to detect contigs consisting exclusively of bacterial best blast hits (i.e., possible contamination). None were found.

## Gene prediction

Gene and protein models for the 10 sequenced Cyanidiales were predicted using MAKER v3 beta (*Cantarel et al., 2008*). MAKER was trained using existing protein sequences from *Cyanidioschyzon merolae* 10D and *Galdieria sulphuraria* 074W, for which we used existing RNA-Seq (*Rossoni, 2018*) data with expression values > 10 FPKM (*Rademacher et al., 2016*) combined with protein sequences from the UniProtKB/Swiss-Prot protein database (*UniProt Consortium T, 2018*). Augustus (*Stanke and Morgenstern, 2005*), GeneMark ES (*Borodovsky and Lomsadze, 2011*), and EVM (*Haas et al., 2008*) were used for gene prediction. MAKER was run iteratively and using various options for each genome. The resulting gene models were again assessed using BUSCO v.3 (*Simão et al., 2015*) and PFAM 31.0 (*Finn et al., 2016*). The best performing set of gene models was chosen for each species.

## Sequence annotation

The transcriptomes of all sequenced species and those of *Cyanidioschyzon merolae* 10D, *Galdieria sulphuraria* 074W, and *Galdieria phlegrea* DB10 were annotated (re-annotated) using BLAST2GO PRO v.5 (*Götz et al., 2008*) combined with INTERPROSCAN (*Jones et al., 2014*) in order to obtain the annotations, Gene Ontology (GO)-Terms (*Ashburner et al., 2000*), and Enzyme Commission (EC)-Numbers (*Bairoch, 2000*). KEGG orthology identifiers (KO-Terms) were obtained using KAAS (*Ogata et al., 1999*; *Moriya et al., 2007*) and PFAM annotations using PFAM 31.0 (*Finn et al., 2016*).

## Orthogroups and phylogenetic analysis

The 81,682 predicted protein sequences derived from the 13 genomes listed in *Table 1* were clustered into orthogroups (OGs) using OrthoFinder v. 2.2 (*Emms and Kelly, 2015*). We queried each OG member using DIAMOND v. 0.9.22 (*Buchfink et al., 2015*) to an in-house database comprising NCBI RefSeq sequences with the addition of predicted algal proteomes available from the JGI Genome Portal (*Nordberg et al., 2014*), TBestDB (*O'Brien et al., 2007*), dbEST (*Boguski et al., 1993*), and the MMETSP (Moore Microbial Eukaryote Transcriptome Sequencing Project) (*Keeling et al., 2014*). The database was partitioned into four volumes: Bacteria, Metazoa, remaining taxa, and the MMETSP data. To avoid taxonomic sampling biases due to under or overabundance of particular lineages in the database, each volume was queried independently with an expect (e-value) of $1 \times 10^{-5,}$ and the top 2000 hits were saved and combined into a single list that was then sorted by descending DIAMOND bitscore. Proteins containing one or more bacterial hits (and thus possible HGT candidates) were retained for further analysis, whereas those lacking bacterial hits were removed. A taxonomically broad list of hits was selected for each query (the maximum number of genera selected for each taxonomic phylum present in the DIAMOND output was equivalent to 180 divided by the number of unique phyla), and the corresponding sequences were extracted from the database and aligned using MAFFT v7.3 (*Katoh and Standley, 2013*) together with queries and hits selected in the same manner for remaining proteins assigned to the same OG (duplicate hits were removed). A maximum-likelihood phylogeny was then constructed for each alignment using IQTREE v7.3 (*Nguyen et al., 2015*) under automated model selection, with node support calculated using 2000 ultrafast bootstraps. Single-gene trees for the referenced HGT candidates from previous research regarding *G. sulphuraria* 074W (*Schönknecht et al., 2013*) and *G. phlegrea* DBV009 (*Qiu et al., 2013*) were constructed in the same manner, without assignment to OG. To create the algal species tree, the OG assignment was re-run with the addition of proteomes from outgroup taxa *Porphyra umbilicalis* (*Brawley et al., 2017*), *Porphyridium purpureum* (*Bhattacharya et al., 2013*), *Ostreococcus tauri* RCC4221 (*Blanc-Mathieu et al., 2014*), and *Chlamydomonas reinhardtii* (*Merchant et al., 2007*). Orthogroups were parsed and 2090 were selected that contained single-copy representative proteins from at least 12/17 taxa; those taxa with multi-copy representatives were removed entirely from the OG. The proteins for each OG were extracted and aligned with MAFFT, and IQTREE was used to construct a single maximum-likelihood phylogeny via a partitioned

analysis in which each OG alignment represented one partition with unlinked models of protein evolution chosen by IQTREE. Consensus tree branch support was determined by 2,000 UF bootstraps.

## Detection of HGTs

All phylogenies containing bacterial sequences were inspected manually. Only trees in which there were at least two different Cyanidiales sequences and at least three different non-eukaryotic donors were retained. The singleton HGT candidates in Cyanidiales are presented in the appendix (Appendix 5) and were not analyzed further here. Phylogenies with cyanobacteria and Chlamydiae as sisters were considered as EGT and excluded from the analysis. Genes that were potentially transferred from cyanobacteria were only accepted as HGT candidates when homologs were absent in other photosynthetic eukaryotes; that is the cyanobacterium was not the closest neighbor, and when the annotation did not include a photosynthetic function, to discriminate from EGT. Furthermore, phylogenies containing inconsistencies within the distribution patterns of species, especially at the root, or UF values below 70% spanning over multiple nodes, were excluded. Each orthogroup was queried against NCBI nr to detect eukaryotic homologs not present in our databases. The conservative approach to HGT assignment used here allowed identification of robust candidates for in-depth analysis. This may however have come at the cost of underestimating HGT at the single species level. Furthermore, some of the phylogenies that were rejected because <3 non-eukaryotic donors were found may have resulted from current incomplete sampling of prokaryotes. For example, OG0001817 is present in the sister species *G. sulphuraria* 074W and *G. sulphuraria* MS1 but has a single bacterial hit (*Acidobacteriaceae bacterium* URHE0068, CBS domain-containing protein, GI:651323331).

## Data deposit

The nuclear, plastid, and mitochondrial sequences of the 10 novel genomes, as well as gene models, ESTs, protein sequences, protein alignments, orthogroup and single gene trees, and gene annotations are available at http://porphyra.rutgers.edu. Raw PacBio RSII reads, and also the genomic, chloroplast and mitochondrial sequences, have been submitted to the NCBI and are retrievable via BioProject ID PRJNA512382.

## Acknowledgements

This work was supported by the Deutsche Forschungsgemeinschaft (under Germany´s Excellence Strategy – EXC-2048/1 – Project ID: 390686111 and EXC 1028, and WE 2231/21–1 to APMW) and by the Heine Research Academy (AWR). We thank Dr. Luke Tallon and Dr. Bruno Hüttel for the excellent technical assistance for PacBio sequencing. Dr. Marion Eisenhut for her fantastic Power Point templates. DCP and DB are grateful to the New Jersey Agricultural Experiment Station and the Rutgers University School of Environmental and Biological Sciences Genome Cooperative for supporting this research.

## Additional information

### Funding

| Funder | Grant reference number | Author |
| --- | --- | --- |
| Deutsche Forschungsge-meinschaft | EXC 1028 | Andreas P M Weber |
| Heinrich-Heine-Universität Düsseldorf | | Andreas P M Weber |
| Deutsche Forschungsge-meinschaft | WE 2231/21-1 | Andreas P M Weber |

The funders had no role in study design, data collection and interpretation, or the decision to submit the work for publication.

## Author contributions
Alessandro W Rossoni, Conceptualization, Resources, Data curation, Software, Formal analysis, Funding acquisition, Validation, Investigation, Visualization, Methodology, Writing—original draft, Project administration; Dana C Price, Resources, Data curation, Software, Formal analysis, Methodology, Writing—original draft; Mark Seger, Conceptualization, Resources, Investigation, Methodology, Writing—review and editing; Dagmar Lyska, Formal analysis, Validation, Methodology, Writing—original draft; Peter Lammers, Conceptualization, Resources, Funding acquisition, Writing—review and editing; Debashish Bhattacharya, Conceptualization, Resources, Data curation, Formal analysis, Supervision, Validation, Methodology, Writing—review and editing; Andreas PM Weber, Conceptualization, Resources, Data curation, Supervision, Funding acquisition, Methodology, Project administration, Writing—review and editing

## Author ORCIDs
Alessandro W Rossoni (iD) https://orcid.org/0000-0003-4716-6799
Debashish Bhattacharya (iD) https://orcid.org/0000-0003-0611-1273
Andreas PM Weber (iD) https://orcid.org/0000-0003-0970-4672

## Decision letter and Author response
Decision letter https://doi.org/10.7554/eLife.45017.151
Author response https://doi.org/10.7554/eLife.45017.152

# Additional files

## Supplementary files
• Transparent reporting form
DOI: https://doi.org/10.7554/eLife.45017.107

## Data availability
The genomic, chloroplast and mitochondrial sequences of the 10 novel genomes, as well as gene models, ESTs, protein sequences, and gene annotations are available at http://porphyra.rutgers.edu. These data have also be uploaded to Dryad doi:10.5061/dryad.m06n200. Raw PacBio RSII reads, and also the genomic, chloroplast and mitochondrial sequences, have been submitted to the NCBI and are retrievable via BioProject ID PRJNA512382.

The following datasets were generated:

| Author(s) | Year | Dataset title | Dataset URL | Database and Identifier |
|---|---|---|---|---|
| Alessandro W Rossoni, Dana C Price, Mark Seger, Dagmar Lyska, Peter Lammers, Debashish Bhattacharya, Andreas PM Weber | 2019 | Genome sequencing of 10 novel Cyanidiales strains | https://www.ncbi.nlm.nih.gov/bioproject/PRJNA512382/ | NCBI Sequence Read Archive, PRJNA512382 |
| Alessandro W Rossoni, Dana C Price, Mark Seger, Dagmar Lyska, Peter Lammers, Debashish Bhattacharya, Andreas PM Weber | 2019 | Data from: The genomes of polyextremophilic Cyanidiales contain 1% horizontally transferred genes with diverse adaptive functions | http://dx.doi.org/10.5061/dryad.m06n200 | Dryad Digital Repository, 10.5061/dryad.m06n200 |
| Alessandro W Rossoni, Dana C Price, Mark Seger, Dagmar Lyska, Peter Lammers, Debashish Bhattacharya, Andreas PM Weber | 2019 | Red Algal Resources to Promote Integrative Research in Algal Genomics | http://porphyra.rutgers.edu/bindex.php | Rutgers University, Red Algal |

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

## Appendix 1

DOI: https://doi.org/10.7554/eLife.45017.108

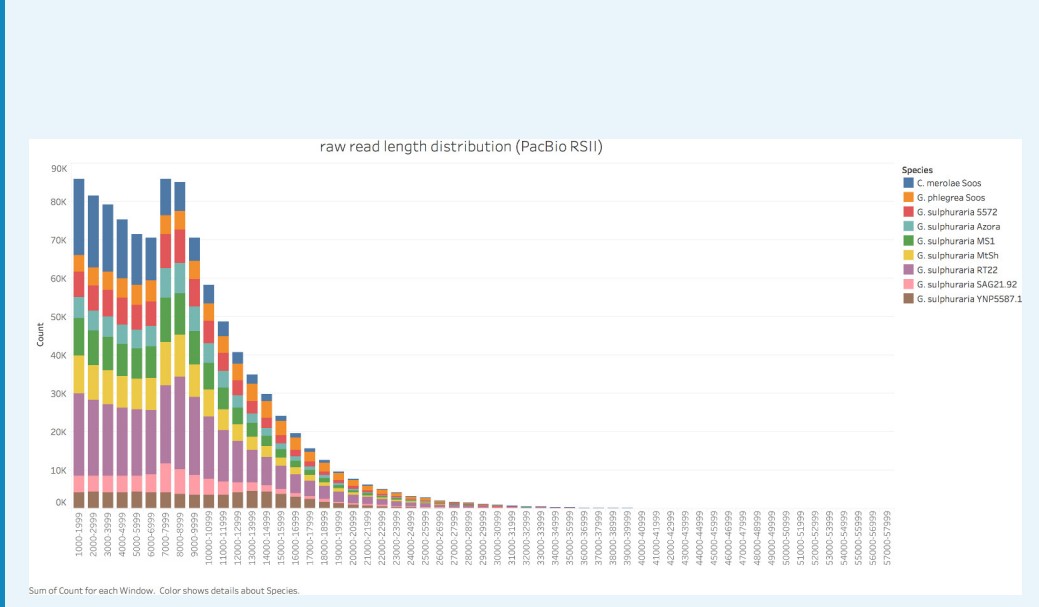

**Appendix 1—figure 1.** Raw read length distribution of the sequenced Cyanidiales strains. The strains were sequenced in 2016/2017 using PacBio's RS2 sequencing technology and P6-C4 chemistry (the only exception being *C. merolae Soos*, which was sequenced as pilot study using P4-C2 chemistry in 2014). Seven strains, namely *G. sulphuraria* 5572, *G. sulphuraria* 002, *G. sulphuraria* SAG21.92, *G. sulphuraria* Azora, *G. sulphuraria* MtSh, *G. sulphuraria* RT22 and *G. sulphuraria* MS1 were sequenced at the University of Maryland Institute for Genome Sciences (Baltimore, USA). The remaining three strains, *G. sulphuraria* YNP5578.1, *G. phlegrea* Soos and *C. merolae* Soos, were sequenced at the Max-Planck-Institut für Pflanzenzüchtungsforschung (Cologne, Germany).

DOI: https://doi.org/10.7554/eLife.45017.109

**Appendix 1—table 1.** Sequencing and Assembly stats.

The strains were sequenced using PacBio's RS2 sequencing technology and P6-C4 chemistry (the only exception being *C. merolae Soos*, which was sequenced using P4-C2 chemistry). For genome assembly, canu version 1.5 was used, followed by polishing three times using the Quiver algorithm. Genes were predicted with MAKER v3 beta(*Doolittle, 1999; Doolittle, 1999*). The performance of genome assemblies (not shown here) and gene prediction was assessed using BUSCO v.3. Raw Reads: Number of raw PacBio RSII reads. Raw Reads N50: 50% of the raw sequence is contained in reads with sizes greater than the N50 value. Raw Reads GC: GC content of the raw reads in percent. Raw Reads (bp): Total number of sequenced basepairs (nucleotides) per species. Raw Coverage (bp): Genomic coverage by raw reads. This figure was computed once the assembly was finished. Unitigging (bp): Total number of basepairs that survived read correction and trimming. This amount of sequence is what the assembler considered when constructing the genome. Unitigging Coverage: Genomic coverage by corrected and trimmed reads. Genome Size (bp): Size of the polished genome. Genome GC: GC content of the polished genome. Contigs: Number of contigs. Contig N50: 50% of the final genomic sequence is contained in contigs sizes greater than the N50 value. Genes: Number of genes predicted by Maker v3 beta. BUSCO (C): Percentage of complete gene models. BUSCO (C + F): Percentage of complete and fragmented gene models. BUSCO (D): Percentage of duplicated gene models. BUSCO (M): Percentage of missing gene models. Fragmented gene models are also somewhat present.

| Species | Raw reads | Raw reads N50 | Raw reads GC | Raw reads (bp) | Raw reads coverage | Unitigging (bp) | Unitigging coverage | Genome size (bp) | Genome GC | Contigs | Contig N50 | Genes | Busco (C) | Busco (C + F) | Busco (D) | Busco (M) |
|---|---|---|---|---|---|---|---|---|---|---|---|---|---|---|---|---|
| G. sulphuraria RT22 | 163764 | 12023 | 35.83% | 1424372481 | 91.20 | 1108677098 | 70.99 | 15617852 | 37.43% | 118 | 172878 | 6982 | 92.8% | 94.5% | 6.3% | 5.5% |
| G. sulphuraria 002 | 131978 | 10109 | 37.90% | 946093501 | 67.05 | 805608410 | 57.09 | 14110219 | 39.16% | 107 | 189293 | 5912 | 87.5% | 92.5% | 5.0% | 7.5% |
| G. sulphuraria 5572 | 101472 | 10449 | 36.45% | 802203307 | 56.19 | 664626554 | 46.55 | 14277368 | 37.99% | 108 | 229711 | 6472 | 91.5% | 93.5% | 5.0% | 6.5% |
| G. sulphuraria MS1 | 128294 | 9991 | 36.18% | 934546621 | 62.77 | 777587876 | 52.23 | 14887946 | 37.62% | 129 | 172087 | 7441 | 90.8% | 94.1% | 4.0% | 5.9% |
| G. sulphuraria MtSh | 158936 | 13617 | 39.19% | 1523875693 | 101.95 | 1235394614 | 82.65 | 14947614 | 40.04% | 101 | 186619 | 6160 | 87.4% | 91.7% | 6.9% | 8.3% |
| G. sulphuraria Azora | 82544 | 10244 | 37.09% | 651280930 | 46.31 | 551720524 | 39.23 | 14063793 | 40.10% | 127 | 162248 | 6305 | 88.4% | 92.0% | 2.3% | 8.0% |

Appendix 1—table 1 continued

| Species | Raw reads | Raw reads N50 | Raw reads GC | Raw reads (bp) | Raw reads coverage | Unitigging (bp) | Unitigging coverage | Genome size (bp) | Genome GC | Contigs | Contig N50 | Genes | Busco (C) | Busco (C + F) | Busco (D) | Busco (M) |
|---|---|---|---|---|---|---|---|---|---|---|---|---|---|---|---|---|
| G. sulphuraria SAG21.92 | 71480 | 10341 | 36.67% | 564874149 | 39.47 | 413793659 | 28.91 | 14312824 | 37.92% | 135 | 158217 | 5956 | 83.8% | 88.4% | 3.6% | 11.6% |
| G. sulphuraria YNP5587.1 | 77421 | 13842 | 36.69% | 769606723 | 53.38 | 613905250 | 42.58 | 14416547 | 40.05% | 115 | 170797 | 6118 | 91.8% | 93.5% | 5.0% | 6.5% |
| G. phlegrea Soos | 92263 | 14365 | 36.01% | 966702049 | 65.00 | 619580741 | 41.66 | 14872696 | 37.52% | 108 | 201071 | 6125 | 92.1% | 93.8% | 7.9% | 6.2% |
| C. merolae Soos | 154461 | 7924 | 52.92% | 848542698 | 68.82 | 570542830 | 46.27 | 12329961 | 54.33% | 35 | 567466 | 4406 | 85.2% | 89.5% | 2.0% | 10.5% |
| G. sulphuraria 074W* | | | | | | | | 13712004 | 36.89% | 433 | 172322 | 7177 | 83.8% | 87.4% | 2.3% | 10.3% |
| C. merolae 10D* | | | | | | | | 16728945 | 54.81% | 22 | 859119 | 5044 | 90.4% | 93.4% | 1.3% | 6.6% |
| G. phlegrea DBV009* | | | | | | | | 11413183 | 37.86% | 9311 | 1993 | 7836 | 68.3% | 88.1% | 3.6% | 11.9% |

DOI: https://doi.org/10.7554/eLife.45017.110

## Appendix 2

DOI: https://doi.org/10.7554/eLife.45017.108

### Archaeal ATPases and 'old' HGT

We compared the HGT results of this study to previous published claims of HGT in *G. sulphuraria* 074W (75 separate acquisitions followed by gene family expansion, 335 transcripts in total) (*Schönknecht et al., 2013*) and *G. phlegrea* DBV009 (13 genes from 11 acquisitions unique to this strain, excluding those shared with *G. sulphuraria* 074W and other red algae) (*Qiu et al., 2013*). Each HGT candidate was queried against our database, mapped to the existing OGs and phylogenetic trees were built for each sequence (where possible). The HGT candidates of *G. sulphuraria* 074W mapped into 100 different OGs, thus increasing the number of separate origins from 75 to 100 (more separate origins = less gene family expansion). 211 out of the 335 HGT candidates in *G. sulphuraria* 074W are 'archaeal STAND ATPases'. They clustered into OG0000000, OG0000003 and OG0000001 which are not classified as HGT. Thus, HGT origin for those gene families can be excluded. The remaining 124 *G. sulphuraria* 074W HGT candidates are spread across 98 OGs. Of those, 20 OGs overlap with our HGT findings, whereas 78 are OGs that do not have HGT origins (one was classified as EGT). All 13 HGT candidates in *G. phlegrea* DBV009 were found and their HGT origin could be confirmed. Some do not make the cut due to individual acquisitions by *G. phlegrea* DBV009 alone. However, considering the operon structures of the acquisition it seems plausible in this case.

In order to exclude the possibility that our database was 'missing' crucial non-eukaryotic species we queried all protein sequences against our own database and NCBI's uncurated nr database, including predicted models and environmental samples and implementing various search strategies. 219 out of the 335 HGT candidates in *G. sulphuraria* 074W did not report any hits outside the species itself (including the 211 'archaeal ATPases') and no functional evidence could be found besides the one obtained through manual curation of sequence alignments as reported by the author (*Schönknecht et al., 2013*).

As seen in the case of the human and the Tardigrade genome, the overestimation of HGT in eukaryotic genomes, followed by later re-correction, is not a new phenomenon (*Boothby et al., 2015*; *Crisp et al., 2015*; *Koutsovoulos et al., 2016*; *Salzberg, 2017*). There are several reasons that may have led to the drastic overestimation of HGT candidates in the case of *G. sulphuraria* 074W (100 OGs derived from HGT, instead of 58 OGs). Although published in 2013, the HGT analysis was performed in early 2007. By then, the RefSeq database contained 4.7 million accessions compared to 163.9 million accessions in May 2018. The low resolution regarding eukaryotic species may have led to many singletons, here defined as *Galdieria* being the only eukaryotic species in otherwise bacterial clusters, leading to the mislabelling of HGT. Further, the many small contigs derived from short read sequencing technologies of the last decade, combined with older assembly software [138] are known potential pitfalls (*Danchin, 2016*) for missassembly that may lead to the inclusion of bacterial contigs into the reference genome as a consequence of prior culture contamination. Lastly, this analysis occurred a decade prior to the tardigrade and human case that led to raised awareness and standards regarding HGT annotation as many claims of HGT were later refuted by further analyses. From a biological view the HGT origin of the Archaeal ATPases is disputable as a re-sequencing of the Genome using MinION technology (Rossoni, data unpublished) shows they always occur immediately adjacent to every single telomere, therefore adding another layer of complexity. The 'archaeal ATPase' was not only integrated into the genome, but also put under influence a non-random duplication mechanism responsible for spreading copies in a targeted manner to the subtelomeric region of each single contig (no exception!). Examples of similar cases may be found in the Variant Surface Glycoproteins (VSGs) of the Trypanosoma [139] and the Candidates for Secreted Effector Proteins (CSEPs) in the powdery mildew fungus Blumeria graminis [140]. As those genes are vital for the infection of the host, they are subjects of very strong natural selection and profit

from high evolutionary rates achieved at the subtelomeric regions. But the high evolutionary rates also made it impossible to correctly embed the aforementioned gene families in a phylogenetic tree. As such, it is not to be excluded that a similar case occurred regarding *Galdieria sulphuraria*'s 'archaeal ATPases', although a permissive search might indicate an archaeal origin of single protein domains. Also, as only a patchy subset of the ATPases reacts to temperature fluctuations, it cannot be determined that temperature is the driving factor.

# Appendix 3

DOI: https://doi.org/10.7554/eLife.45017.108

## %GC

**Appendix 3—table 1. %GC analysis of the Cyanidiales transcriptomes.** %GC content of HGT genes was compared to the %GC content of native genes using students test. Legend: HGT Genes: number of HGT gene candidates found in species. Avg. %GC Native: average %GC of native transcripts. Avg. %GC HGT: average %GC of HGT candidates. P-Val (T-test): significance value (p-value) of student's test. Delta: difference in %GC between average %GC of native genes and the average %GC of HGT candidates.

| | HGT genes | Avg. %GC Native | Avg. %GC HGT | p-Val (T-test) | Delta |
|---|---|---|---|---|---|
| Galdieria_sulphuraria_074W | 55 | 38.99 | 39.62 | 0.046 | 0.63 |
| Galdieria_sulphuraria_MS1 | 58 | 39.59 | 40.79 | 0 | 1.2 |
| Galdieria_sulphuraria_RT22 | 54 | 39.54 | 40.85 | 0 | 1.31 |
| Galdieria_sulphuraria_SAG21 | 47 | 40.04 | 41.47 | 0 | 1.43 |
| Galdieria_sulphuraria_MtSh | 47 | 41.33 | 42.48 | 0 | 1.15 |
| Galdieria_sulphuraria_Azora | 58 | 41.34 | 42.57 | 0 | 1.23 |
| Galdieria_sulphuraria_YNP55871 | 46 | 41.33 | 42.14 | 0.006 | 0.81 |
| Galdieria_sulphuraria_5572 | 53 | 39.68 | 40.5 | 0.002 | 0.82 |
| Galdieria_sulphuraria_002 | 52 | 40.76 | 41.35 | 0.016 | 0.59 |
| Galdieria_phlegrea_DBV08 | 54 | 39.97 | 40.58 | 0.016 | 0.61 |
| Galdieria_phlegrea_Soos | 44 | 39.57 | 40.73 | 0 | 1.16 |
| Cyanidioschyzon_merolae_10D | 33 | 56.57 | 56.57 | 0.996 | 0 |
| Cyanidioschyzon_merolae_Soos | 34 | 54.84 | 54.26 | 0.479 | −0.58 |

DOI: https://doi.org/10.7554/eLife.45017.113

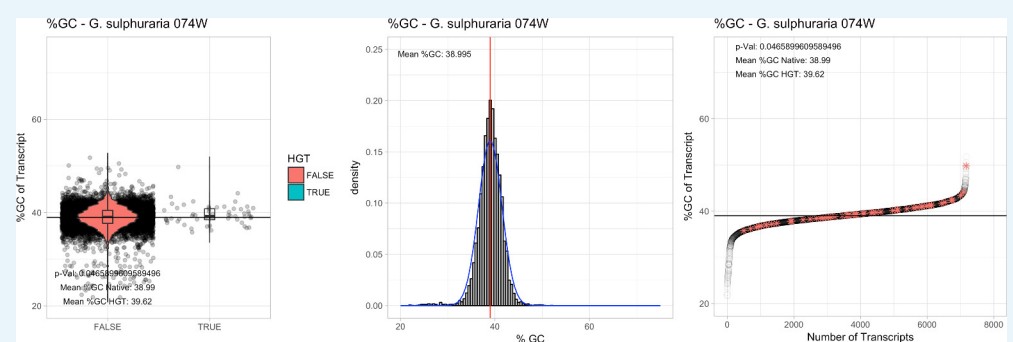

**Appendix 3—figure 1.** %GC – *Galdieria sulphuraria 074W*: (Left) Violin plot showing the %GC distribution across native transcripts and HGT candidates. (Mid) Cumulative %GC distribution of transcripts. Red line shows the average, blue line a normal distribution based on the average value. (Right) Ranking all transcripts based upon their %GC content. Red '*' demarks HGT candidates. As the %GC content was normally distributed, students test was applied for the determination of significant differences between the native gene and the HGT candidate subset.

DOI: https://doi.org/10.7554/eLife.45017.114

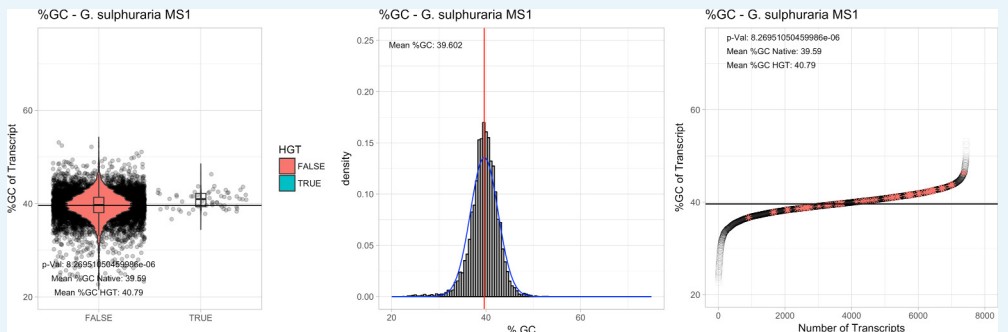

**Appendix 3—figure 2.** %GC – *Galdieria sulphuraria MS1*: (Left) Violin plot showing the %GC distribution across native transcripts and HGT candidates. (Mid) Cumulative %GC distribution of transcripts. Red line shows the average, blue line a normal distribution based on the average value. (Right) Ranking all transcripts based upon their %GC content. Red '*' demarks HGT candidates. As the %GC content was normally distributed, students test was applied for the determination of significant differences between the native gene and the HGT candidate subset.

DOI: https://doi.org/10.7554/eLife.45017.115

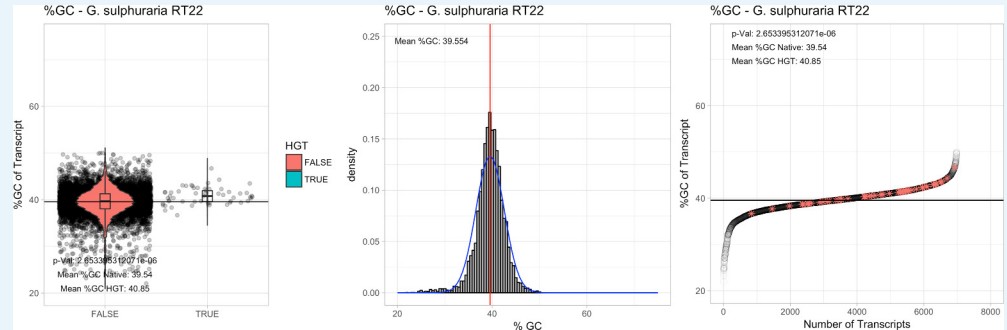

**Appendix 3—figure 3.** %GC – *Galdieria sulphuraria RT22*: (Left) Violin plot showing the %GC distribution across native transcripts and HGT candidates. (Mid) Cumulative %GC distribution of transcripts. Red line shows the average, blue line a normal distribution based on the

average value. (Right) Ranking all transcripts based upon their %GC content. Red '*' demarks HGT candidates. As the %GC content was normally distributed, students test was applied for the determination of significant differences between the native gene and the HGT candidate subset.

DOI: https://doi.org/10.7554/eLife.45017.116

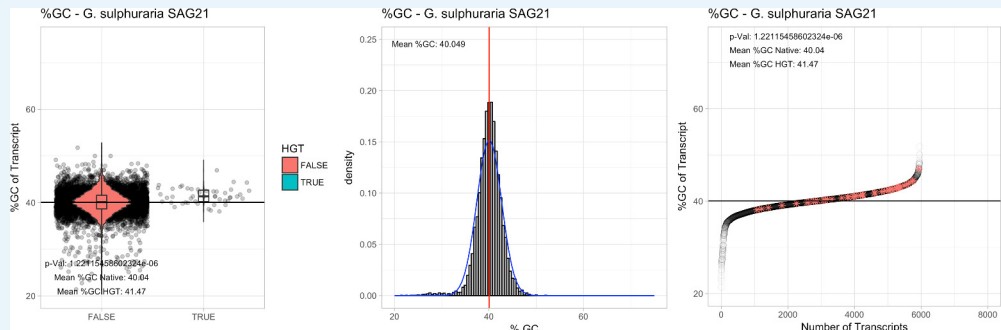

**Appendix 3—figure 4.** %GC – *Galdieria sulphuraria SAG21:* (Left) Violin plot showing the % GC distribution across native transcripts and HGT candidates. (Mid) Cumulative %GC distribution of transcripts. Red line shows the average, blue line a normal distribution based on the average value. (Right) Ranking all transcripts based upon their %GC content. Red '*' demarks HGT candidates. As the %GC content was normally distributed, students test was applied for the determination of significant differences between the native gene and the HGT candidate subset.

DOI: https://doi.org/10.7554/eLife.45017.117

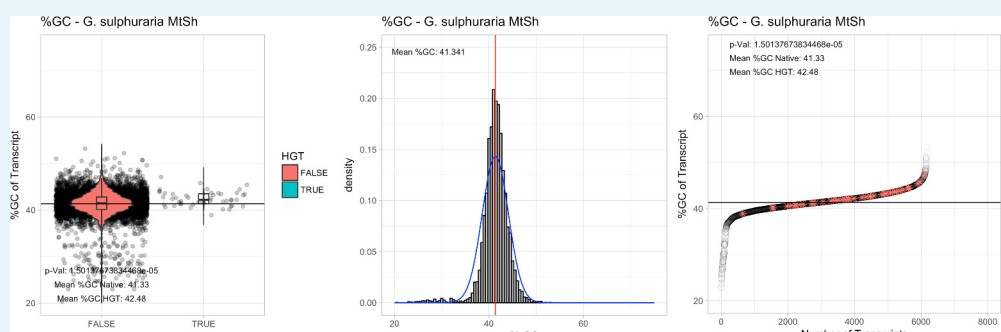

**Appendix 3—figure 5.** %GC – *Galdieria sulphuraria Mount Shasta (MtSh):* (Left) Violin plot showing the %GC distribution across native transcripts and HGT candidates. (Mid) Cumulative %GC distribution of transcripts. Red line shows the average, blue line a normal distribution based on the average value. (Right) Ranking all transcripts based upon their %GC content. Red '*' demarks HGT candidates. As the %GC content was normally distributed, students test was applied for the determination of significant differences between the native gene and the HGT candidate subset.

DOI: https://doi.org/10.7554/eLife.45017.118

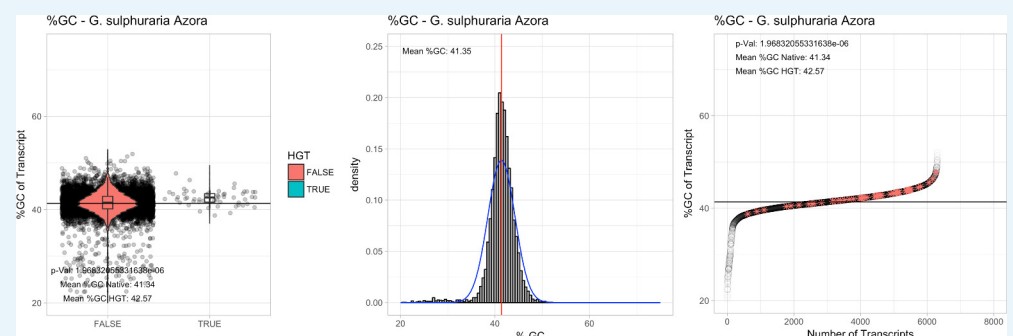

**Appendix 3—figure 6.** *Galdieria sulphuraria Azora:* (Left) Violin plot showing the %GC distribution across native transcripts and HGT candidates. (Mid) Cumulative %GC distribution of transcripts. Red line shows the average, blue line a normal distribution based on the average value. (Right) Ranking all transcripts based upon their %GC content. Red '*' demarks HGT candidates. As the %GC content was normally distributed, students test was applied for the determination of significant differences between the native gene and the HGT candidate subset.

DOI: https://doi.org/10.7554/eLife.45017.119

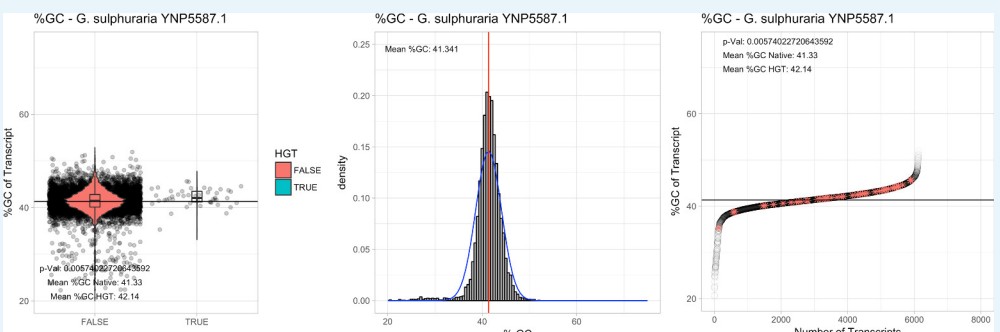

**Appendix 3—figure 7.** %GC – *Galdieria sulphuraria Mount Shasta YNP5578.1:* (Left) Violin plot showing the %GC distribution across native transcripts and HGT candidates. (Mid) Cumulative %GC distribution of transcripts. Red line shows the average, blue line a normal distribution based on the average value. (Right) Ranking all transcripts based upon their %GC content. Red '*' demarks HGT candidates. As the %GC content was normally distributed, students test was applied for the determination of significant differences between the native gene and the HGT candidate subset.

DOI: https://doi.org/10.7554/eLife.45017.120

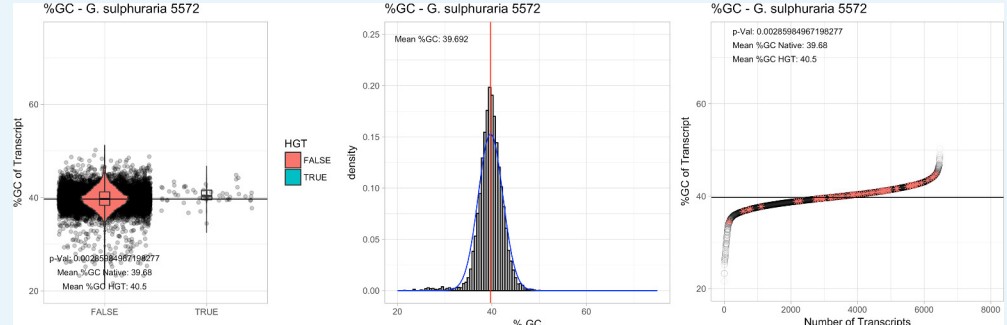

**Appendix 3—figure 8.** %GC – *Galdieria sulphuraria 5572:* (Left) Violin plot showing the %GC distribution across native transcripts and HGT candidates. (Mid) Cumulative %GC distribution of transcripts. Red line shows the average, blue line a normal distribution based on the

average value. (Right) Ranking all transcripts based upon their %GC content. Red '*' demarks HGT candidates. As the %GC content was normally distributed, students test was applied for the determination of significant differences between the native gene and the HGT candidate subset.

DOI: https://doi.org/10.7554/eLife.45017.121

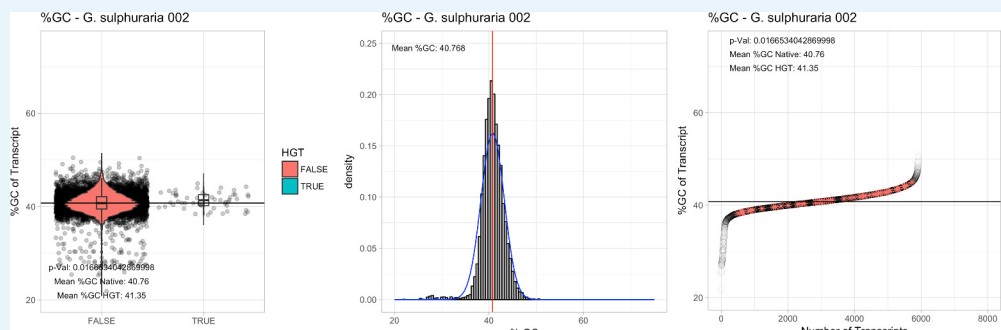

**Appendix 3—figure 9.** %GC – *Galdieria sulphuraria 002:* (Left) Violin plot showing the %GC distribution across native transcripts and HGT candidates. (Mid) Cumulative %GC distribution of transcripts. Red line shows the average, blue line a normal distribution based on the average value. (Right) Ranking all transcripts based upon their %GC content. Red '*' demarks HGT candidates. As the %GC content was normally distributed, students test was applied for the determination of significant differences between the native gene and the HGT candidate subset.

DOI: https://doi.org/10.7554/eLife.45017.122

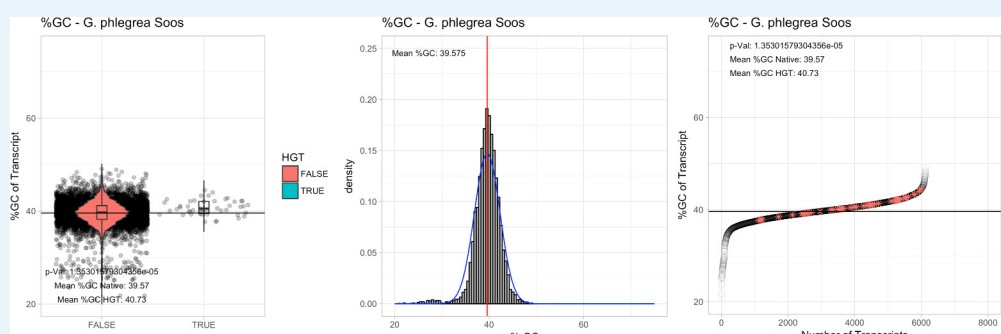

**Appendix 3—figure 10.** %GC – *Galdieria phlegrea Soos:* (Left) Violin plot showing the %GC distribution across native transcripts and HGT candidates. (Mid) Cumulative %GC distribution of transcripts. Red line shows the average, blue line a normal distribution based on the average value. (Right) Ranking all transcripts based upon their %GC content. Red '*' demarks HGT candidates. As the %GC content was normally distributed, students test was applied for the determination of significant differences between the native gene and the HGT candidate subset.

DOI: https://doi.org/10.7554/eLife.45017.123

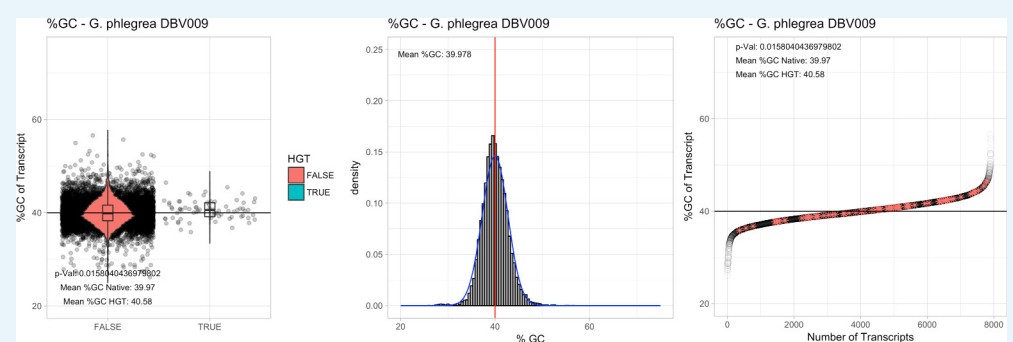

**Appendix 3—figure 11.** %GC – *Galdieria phlegrea DBV009:* (Left) Violin plot showing the %GC distribution across native transcripts and HGT candidates. (Mid) Cumulative %GC distribution of transcripts. Red line shows the average, blue line a normal distribution based on the average value. (Right) Ranking all transcripts based upon their %GC content. Red '*' demarks HGT candidates. As the %GC content was normally distributed, students test was applied for the determination of significant differences between the native gene and the HGT candidate subset.

DOI: https://doi.org/10.7554/eLife.45017.124

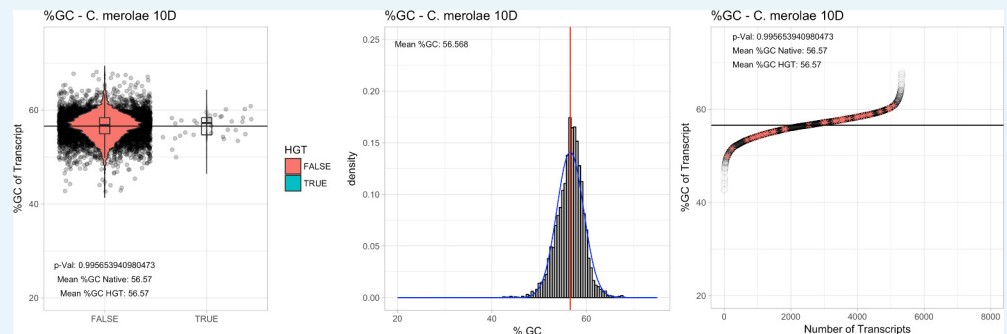

**Appendix 3—figure 12.** %GC – *Cyanidioschyzon merolae Soos:* (Left) Violin plot showing the %GC distribution across native transcripts and HGT candidates. (Mid) Cumulative %GC distribution of transcripts. Red line shows the average, blue line a normal distribution based on the average value. (Right) Ranking all transcripts based upon their %GC content. Red '*' demarks HGT candidates. As the %GC content was normally distributed, students test was applied for the determination of significant differences between the native gene and the HGT candidate subset.

DOI: https://doi.org/10.7554/eLife.45017.125

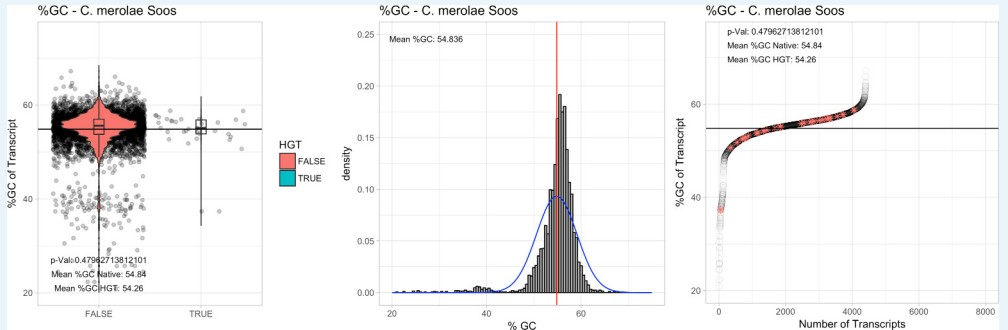

**Appendix 3—figure 13.** %GC – *Cyanidioschyzon merolae 10D:* (Left) Violin plot showing the %GC distribution across native transcripts and HGT candidates. (Mid) Cumulative %GC distribution of transcripts. Red line shows the average, blue line a normal distribution based

on the average value. (Right) Ranking all transcripts based upon their %GC content. Red '*' demarks HGT candidates. As the %GC content was normally distributed, students test was applied for the determination of significant differences between the native gene and the HGT candidate subset.

DOI: https://doi.org/10.7554/eLife.45017.126

# Appendix 4

DOI: https://doi.org/10.7554/eLife.45017.108

**Appendix 4—table 1. Single exon genes vs multiexonic.** The ratio of single exon genes vs multiexonic genes was compared between HGT candidates and native Cyanidiales genes (Fisher enrichment test). Legend: HGT Genes: number of HGT gene candidates found in species. Single Exon HGT: number of single exon genes in HGT candidates. Multi Exon HGT: number of multiexonic genes in HGT candidates. Single Exon Native: number of single exon genes in native Cyanidiales genes. Multi Exon Native: number of multiexonic genes in native Cyanidiales genes. HGT SM Ratio percentage of single exon genes within the HGT candidate genes. Native SM Ratio percentage of single exon genes within the native genes. Delta: difference in percent between the percentage of single exon genes between the native genes and HGT candidates. Fisher p-val: p-value of fisher enrichment test.

| | HGT genes | Single exon (HGT) | Multi exon (HGT) | Single exon (Native) | Multi exon (Native) | Fisher's p | Single exon % (HGT) | Single exon % (Native) | Multi exon % (HGT) | Multi exon % (Native) |
|---|---|---|---|---|---|---|---|---|---|---|
| Galdieria_ sulphuraria_ 074W | 55 | 29 | 26 | 1879 | 5240 | 4.05E-05 | 52.7% | 26.4% | 47.3% | 73.6% |
| Galdieria_ sulphuraria_ MS1 | 58 | 22 | 36 | 1224 | 6159 | 0.0001098 | 37.9% | 16.6% | 62.1% | 83.4% |
| Galdieria_ sulphuraria_ RT22 | 54 | 26 | 28 | 1756 | 5172 | 0.0004079 | 48.1% | 25.3% | 51.9% | 74.7% |
| Galdieria_ sulphuraria_ SAG21 | 47 | 8 | 39 | 901 | 5008 | 0.6852 | 17.0% | 15.2% | 83.0% | 84.8% |
| Galdieria_ sulphuraria_ MtSh | 47 | 17 | 30 | 1239 | 4874 | 0.01054 | 36.2% | 20.3% | 63.8% | 79.7% |
| Galdieria_ sulphuraria_ Azora | 58 | 14 | 39 | 966 | 5286 | 0.03558 | 24.1% | 15.5% | 75.9% | 84.5% |
| Galdieria_ sulphuraria_ YNP55871 | 46 | 21 | 25 | 1548 | 4524 | 0.00341 | 45.7% | 25.5% | 54.3% | 74.5% |
| Galdieria_ sulphuraria_ 5572 | 53 | 29 | 24 | 1389 | 5030 | 1.75E-07 | 54.7% | 21.6% | 45.3% | 78.4% |
| Galdieria_ sulphuraria_ 002 | 52 | 26 | 26 | 140 | 4720 | 8.75E-07 | 50.0% | 2.9% | 50.0% | 97.1% |
| Galdieria_ phlegrea_ DBV009 | 54 | na | na | na | na | na | na | na | na | na |
| Galdieria_ phlegrea_ Soos | 44 | 25 | 22 | 1369 | 4709 | 5.17E-06 | 56.8% | 22.5% | 43.2% | 77.5% |
| Cyanidio schyzon_ merolae_ 10D | 33 | 33 | 0 | 4744 | 26 | 1 | 100.0% | 99.5% | 0.0% | 0.5% |

*Appendix 4—table 1 continued on next page*

*Appendix 4—table 1 continued*

| | HGT genes | Single exon (HGT) | Multi exon (HGT) | Single exon (Native) | Multi exon (Native) | Fisher's p | Single exon % (HGT) | Single exon % (Native) | Multi exon % (HGT) | Multi exon % (Native) |
|---|---|---|---|---|---|---|---|---|---|---|
| Cyanidio schyzon_ merolae_ Soos | 34 | 33 | 1 | 3960 | 412 | 0.367 | 97.1% | 90.6% | 2.9% | 9.4% |

DOI: https://doi.org/10.7554/eLife.45017.128

**Appendix 4—table 2. Exon/Gene ratio.** The ratio of exons per gene was compared between HGT candidates and native Cyanidiales genes (Wilcox ranked test). Legend: HGT Genes: number of HGT gene candidates found in species. E/G All: average number of exons per gene across the whole transcriptome. E/G Native: average number of exons per gene across in native genes. E/G HGT: average number of exons per gene in HGT gene candidates. p-Val (Wilcox) SM Ratio p-value of non-parametric Wilcox test for significant differences. Delta: difference in average number of exons per gene the native genes and HGT candidates.

| | HGT genes | Mean exon per transcript (HGT) | Mean exon per transcript (Native) | Wilcox (p) | Delta |
|---|---|---|---|---|---|
| Galdieria_sulphuraria_074W | 55 | 2.25 | 3.2 | 9.40E-06 | 0.95 |
| Galdieria_sulphuraria_MS1 | 58 | 2.5 | 3.88 | 1.41E-05 | 1.38 |
| Galdieria_sulphuraria_RT22 | 54 | 2.63 | 3.95 | 3.42E-06 | 1.32 |
| Galdieria_sulphuraria_SAG21 | 47 | 4.02 | 5.03 | 0.0004 | 1.01 |
| Galdieria_sulphuraria_MtSh | 47 | 3.15 | 4.32 | 0.0011 | 1.17 |
| Galdieria_sulphuraria_Azora | 58 | 2.68 | 4.03 | 9.92E-05 | 1.35 |
| Galdieria_sulphuraria_YNP55871 | 46 | 2.61 | 3.65 | 2.30E-04 | 1.04 |
| Galdieria_sulphuraria_5572 | 53 | 2.15 | 3.53 | 2.25E-07 | 1.38 |
| Galdieria_sulphuraria_002 | 52 | 2.37 | 3.73 | 2.65E-06 | 1.36 |
| Galdieria_phlegrea_DBV009 | 54 | na | na | na | na |
| Galdieria_phlegrea_Soos | 44 | 2.19 | 3.33 | 1.19E-05 | 1.14 |
| Cyanidioschyzon_merolae_10D | 33 | 1 | 1.01 | 1.00E + 00 | 0.01 |
| Cyanidioschyzon_merolae_Soos | 34 | 1.06 | 1.1 | 2.10E-01 | 0.04 |

DOI: https://doi.org/10.7554/eLife.45017.129

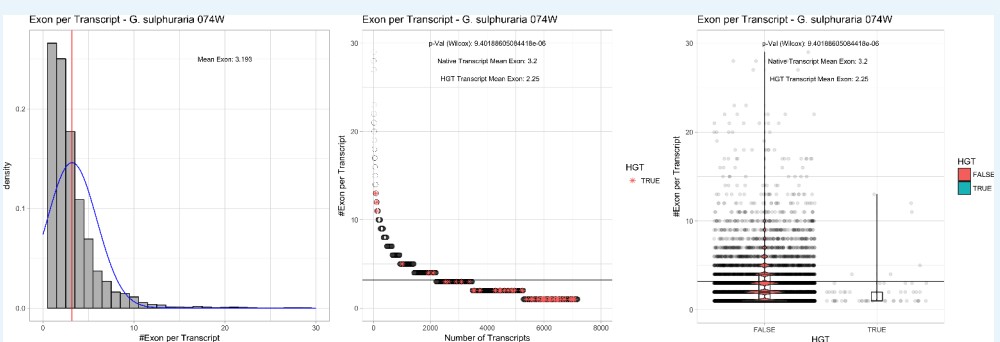

**Appendix 4—figure 1.** Exon/Intron – *Galdieria sulphuraria 074W:* (Left) Mid) Cumulative %GC distribution of transcripts. Red line shows the average, blue line a normal distribution based on the average value. The data is categorical (genes have either one, two, three etc. exons) and does not follow a normal distribution. (Mid) Ranking all transcripts based upon their

number of exons. Red '*' demarks HGT candidates. As the number of exons was not normally distributed, transcripts were ranked by number of exons. In order to resolve the high number of tied ranks (e.g. many transcripts have two exons) a bootstrap was implied by which the rank of transcripts sharing the same number of exons was randomly assigned 1000 times. Wilcoxon-Mann-Whitney-Test applied for the determination of significant rank differences between the native gene and the HGT candidate subset. (Right) Violin plot showing the number of exons per transcript distribution across native transcripts and HGT candidates.

DOI: https://doi.org/10.7554/eLife.45017.130

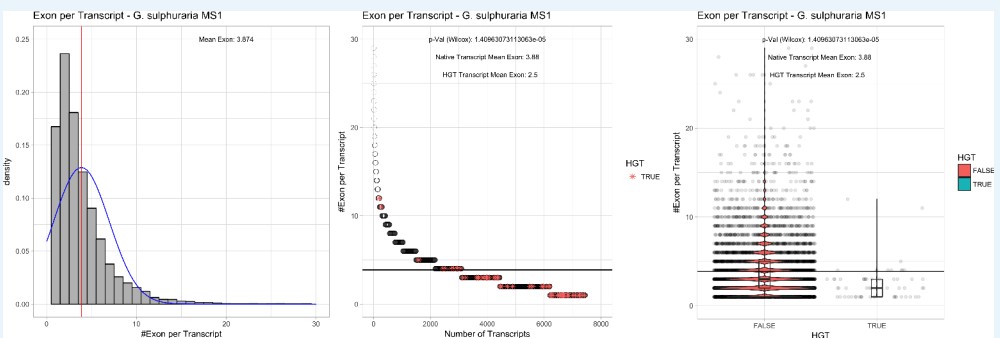

**Appendix 4—figure 2.** Exon/Intron – *Galdieria sulphuraria MS1:* (Left) Mid) Cumulative %GC distribution of transcripts. Red line shows the average, blue line a normal distribution based on the average value. The data is categorical (genes have either one, two, three etc. exons) and does not follow a normal distribution. (Mid) Ranking all transcripts based upon their number of exons. Red '*' demarks HGT candidates. As the number of exons was not normally distributed, transcripts were ranked by number of exons. In order to resolve the high number of tied ranks (e.g. many transcripts have two exons) a bootstrap was implied by which the rank of transcripts sharing the same number of exons was randomly assigned 1000 times. Wilcoxon-Mann-Whitney-Test applied for the determination of significant rank differences between the native gene and the HGT candidate subset. (Right) Violin plot showing the number of exons per transcript distribution across native transcripts and HGT candidates.

DOI: https://doi.org/10.7554/eLife.45017.131

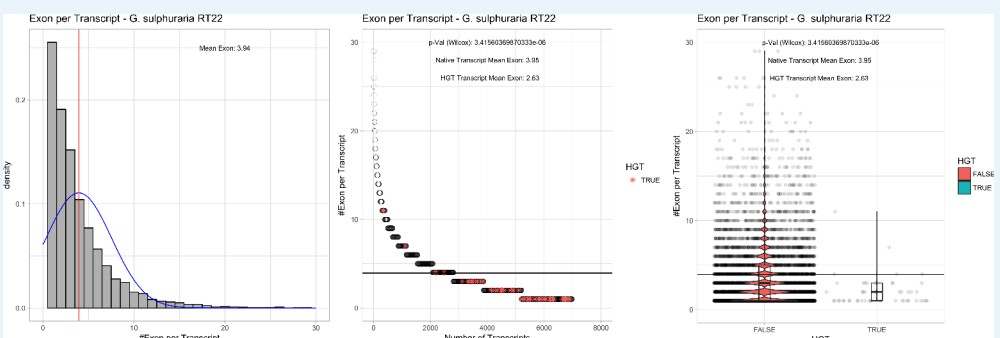

**Appendix 4—figure 3.** Exon/Intron – *Galdieria sulphuraria RT22:* (Left) Mid) Cumulative %GC distribution of transcripts. Red line shows the average, blue line a normal distribution based on the average value. The data is categorical (genes have either one, two, three etc. exons) and does not follow a normal distribution. (Mid) Ranking all transcripts based upon their number of exons. Red '*' demarks HGT candidates. As the number of exons was not normally distributed, transcripts were ranked by number of exons. In order to resolve the high number of tied ranks (e.g. many transcripts have two exons) a bootstrap was implied by which the rank of transcripts sharing the same number of exons was randomly assigned 1000

times. Wilcoxon-Mann-Whitney-Test applied for the determination of significant rank differences between the native gene and the HGT candidate subset. (Right) Violin plot showing the number of exons per transcript distribution across native transcripts and HGT candidates.

DOI: https://doi.org/10.7554/eLife.45017.132

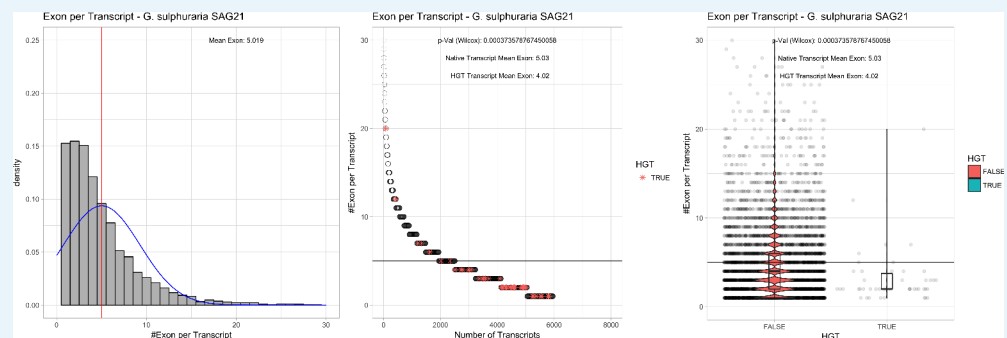

**Appendix 4—figure 4.** Exon/Intron – *Galdieria sulphuraria SAG21:* (Left) Mid) Cumulative % GC distribution of transcripts. Red line shows the average, blue line a normal distribution based on the average value. The data is categorical (genes have either one, two, three etc. exons) and does not follow a normal distribution. (Mid) Ranking all transcripts based upon their number of exons. Red '*' demarks HGT candidates. As the number of exons was not normally distributed, transcripts were ranked by number of exons. In order to resolve the high number of tied ranks (e.g. many transcripts have two exons) a bootstrap was implied by which the rank of transcripts sharing the same number of exons was randomly assigned 1000 times. Wilcoxon-Mann-Whitney-Test applied for the determination of significant rank differences between the native gene and the HGT candidate subset. (Right) Violin plot showing the number of exons per transcript distribution across native transcripts and HGT candidates.

DOI: https://doi.org/10.7554/eLife.45017.133

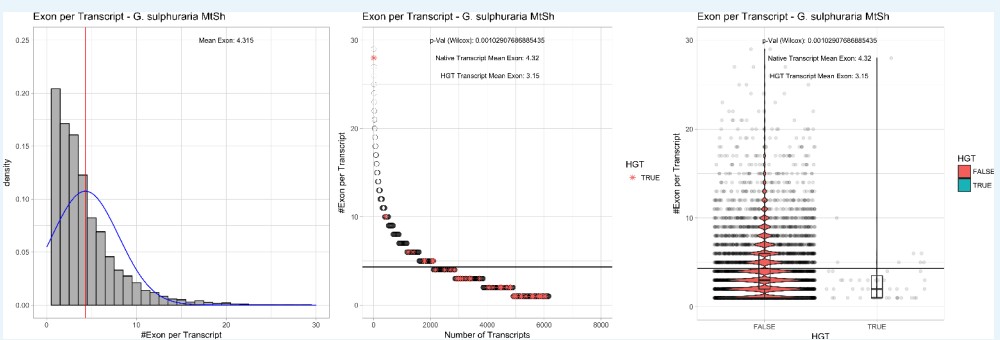

**Appendix 4—figure 5.** Exon/Intron – *Galdieria sulphuraria MtSh:* (Left) Mid) Cumulative %GC distribution of transcripts. Red line shows the average, blue line a normal distribution based on the average value. The data is categorical (genes have either one, two, three etc. exons) and does not follow a normal distribution. (Mid) Ranking all transcripts based upon their number of exons. Red '*' demarks HGT candidates. As the number of exons was not normally distributed, transcripts were ranked by number of exons. In order to resolve the high number of tied ranks (e.g. many transcripts have two exons) a bootstrap was implied by which the rank of transcripts sharing the same number of exons was randomly assigned 1000 times. Wilcoxon-Mann-Whitney-Test applied for the determination of significant rank differences between the native gene and the HGT candidate subset. (Right) Violin plot showing the number of exons per transcript distribution across native transcripts and HGT candidates.

DOI: https://doi.org/10.7554/eLife.45017.134

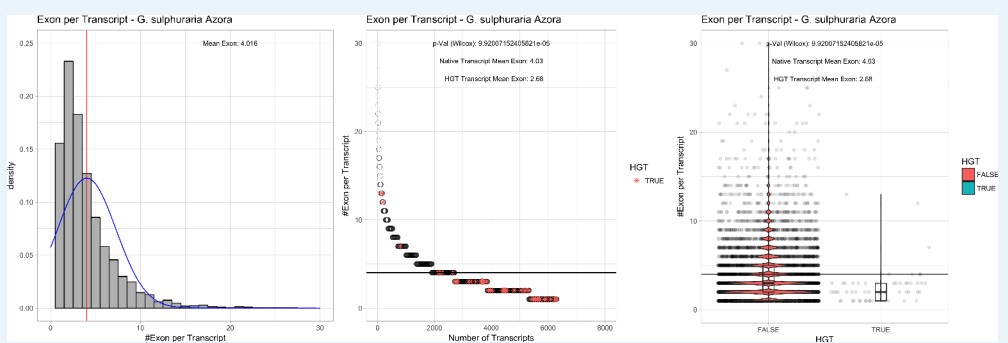

**Appendix 4—figure 6.** Exon/Intron – *Galdieria sulphuraria Azora:* (Left) Mid) Cumulative %GC distribution of transcripts. Red line shows the average, blue line a normal distribution based on the average value. The data is categorical (genes have either one, two, three etc. exons) and does not follow a normal distribution. (Mid) Ranking all transcripts based upon their number of exons. Red '*' demarks HGT candidates. As the number of exons was not normally distributed, transcripts were ranked by number of exons. In order to resolve the high number of tied ranks (e.g. many transcripts have two exons) a bootstrap was implied by which the rank of transcripts sharing the same number of exons was randomly assigned 1000 times. Wilcoxon-Mann-Whitney-Test applied for the determination of significant rank differences between the native gene and the HGT candidate subset. (Right) Violin plot showing the number of exons per transcript distribution across native transcripts and HGT candidates.

DOI: https://doi.org/10.7554/eLife.45017.135

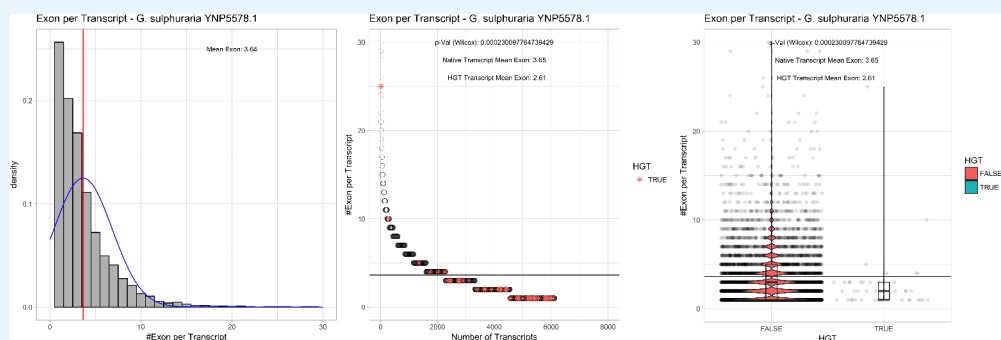

**Appendix 4—figure 7.** Exon/Intron – *Galdieria sulphuraria YNP5578.1:* (Left) Mid) Cumulative %GC distribution of transcripts. Red line shows the average, blue line a normal distribution based on the average value. The data is categorical (genes have either one, two, three etc. exons) and does not follow a normal distribution. (Mid) Ranking all transcripts based upon their number of exons. Red '*' demarks HGT candidates. As the number of exons was not normally distributed, transcripts were ranked by number of exons. In order to resolve the high number of tied ranks (e.g. many transcripts have two exons) a bootstrap was implied by which the rank of transcripts sharing the same number of exons was randomly assigned 1000 times. Wilcoxon-Mann-Whitney-Test applied for the determination of significant rank differences between the native gene and the HGT candidate subset. (Right) Violin plot showing the number of exons per transcript distribution across native transcripts and HGT candidates.

DOI: https://doi.org/10.7554/eLife.45017.136

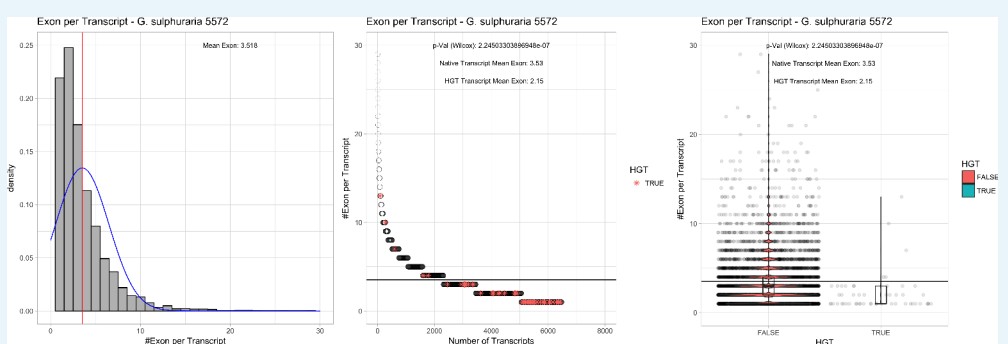

**Appendix 4—figure 8.** Exon/Intron – *Galdieria sulphuraria 5572:* (Left) Mid) Cumulative %GC distribution of transcripts. Red line shows the average, blue line a normal distribution based on the average value. The data is categorical (genes have either one, two, three etc. exons) and does not follow a normal distribution. (Mid) Ranking all transcripts based upon their number of exons. Red '*' demarks HGT candidates. As the number of exons was not normally distributed, transcripts were ranked by number of exons. In order to resolve the high number of tied ranks (e.g. many transcripts have two exons) a bootstrap was implied by which the rank of transcripts sharing the same number of exons was randomly assigned 1000 times. Wilcoxon-Mann-Whitney-Test applied for the determination of significant rank differences between the native gene and the HGT candidate subset. (Right) Violin plot showing the number of exons per transcript distribution across native transcripts and HGT candidates.

DOI: https://doi.org/10.7554/eLife.45017.137

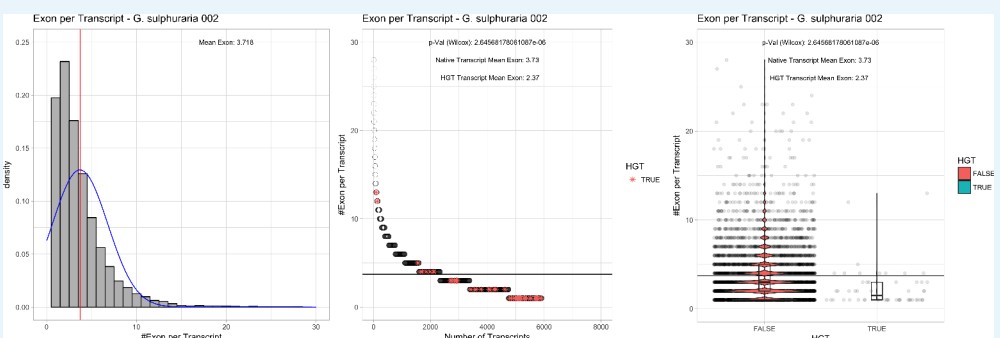

**Appendix 4—figure 9.** Exon/Intron – *Galdieria sulphuraria 002:* (Left) Mid) Cumulative %GC distribution of transcripts. Red line shows the average, blue line a normal distribution based on the average value. The data is categorical (genes have either one, two, three etc. exons) and does not follow a normal distribution. (Mid) Ranking all transcripts based upon their number of exons. Red '*' demarks HGT candidates. As the number of exons was not normally distributed, transcripts were ranked by number of exons. In order to resolve the high number of tied ranks (e.g. many transcripts have two exons) a bootstrap was implied by which the rank of transcripts sharing the same number of exons was randomly assigned 1000 times. Wilcoxon-Mann-Whitney-Test applied for the determination of significant rank differences between the native gene and the HGT candidate subset. (Right) Violin plot showing the number of exons per transcript distribution across native transcripts and HGT candidates.

DOI: https://doi.org/10.7554/eLife.45017.138

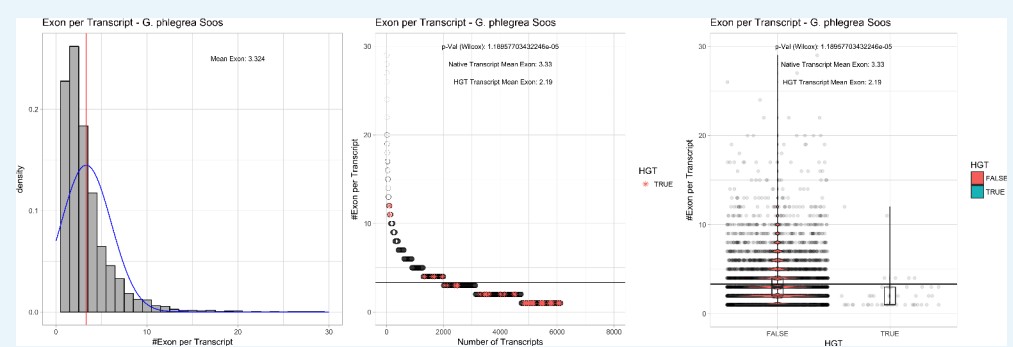

**Appendix 4—figure 10.** Exon/Intron – *Galdieria phlegrea Soos:* (Left) Mid) Cumulative %GC distribution of transcripts. Red line shows the average, blue line a normal distribution based on the average value. The data is categorical (genes have either one, two, three etc. exons) and does not follow a normal distribution. (Mid) Ranking all transcripts based upon their number of exons. Red '*' demarks HGT candidates. As the number of exons was not normally distributed, transcripts were ranked by number of exons. In order to resolve the high number of tied ranks (e.g. many transcripts have two exons) a bootstrap was implied by which the rank of transcripts sharing the same number of exons was randomly assigned 1000 times. Wilcoxon-Mann-Whitney-Test applied for the determination of significant rank differences between the native gene and the HGT candidate subset. (Right) Violin plot showing the number of exons per transcript distribution across native transcripts and HGT candidates.

DOI: https://doi.org/10.7554/eLife.45017.139

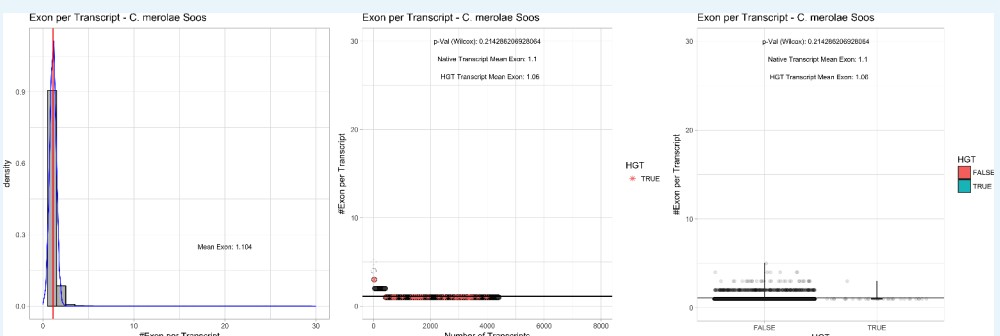

**Appendix 4—figure 11.** Exon/Intron – *Cyanidioschyzon merolae Soos:* (Left) Mid) Cumulative %GC distribution of transcripts. Red line shows the average, blue line a normal distribution based on the average value. The data is categorical (genes have either one, two, three etc. exons) and does not follow a normal distribution. (Mid) Ranking all transcripts based upon their number of exons. Red '*" demarks HGT candidates. As the number of exons was not normally distributed, transcripts were ranked by number of exons. In order to resolve the high number of tied ranks (e.g. many transcripts have two exons) a bootstrap was implied by which the rank of transcripts sharing the same number of exons was randomly assigned 1000 times. Wilcoxon-Mann-Whitney-Test applied for the determination of significant rank differences between the native gene and the HGT candidate subset. (Right) Violin plot showing the number of exons per transcript distribution across native transcripts and HGT candidates..

DOI: https://doi.org/10.7554/eLife.45017.140

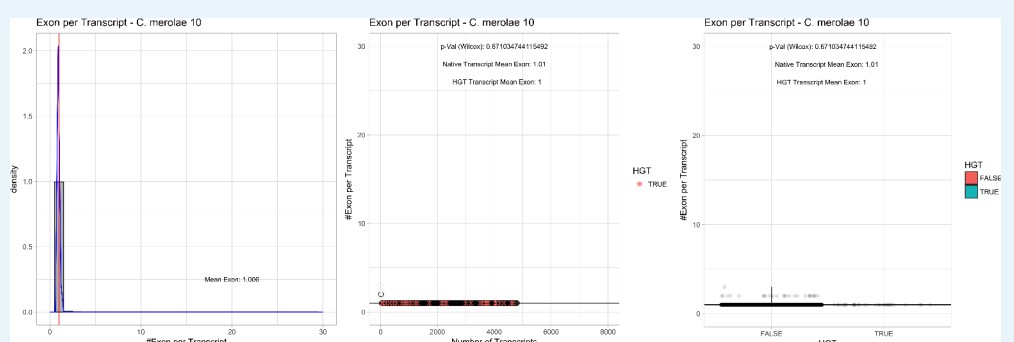

**Appendix 4—figure 12.** Exon/Intron – *Cyanidioschyzon merolae 074W:* (Left) Mid) Cumulative %GC distribution of transcripts. Red line shows the average, blue line a normal distribution based on the average value. The data is categorical (genes have either one, two, three etc. exons) and does not follow a normal distribution. (Mid) Ranking all transcripts based upon their number of exons. Red '*' demarks HGT candidates. As the number of exons was not normally distributed, transcripts were ranked by number of exons. In order to resolve the high number of tied ranks (e.g. many transcripts have two exons) a bootstrap was implied by which the rank of transcripts sharing the same number of exons was randomly assigned 1000 times. Wilcoxon-Mann-Whitney-Test applied for the determination of significant rank differences between the native gene and the HGT candidate subset. (Right) Violin plot showing the number of exons per transcript distribution across native transcripts and HGT candidates.

DOI: https://doi.org/10.7554/eLife.45017.141

## Appendix 5

DOI: https://doi.org/10.7554/eLife.45017.108

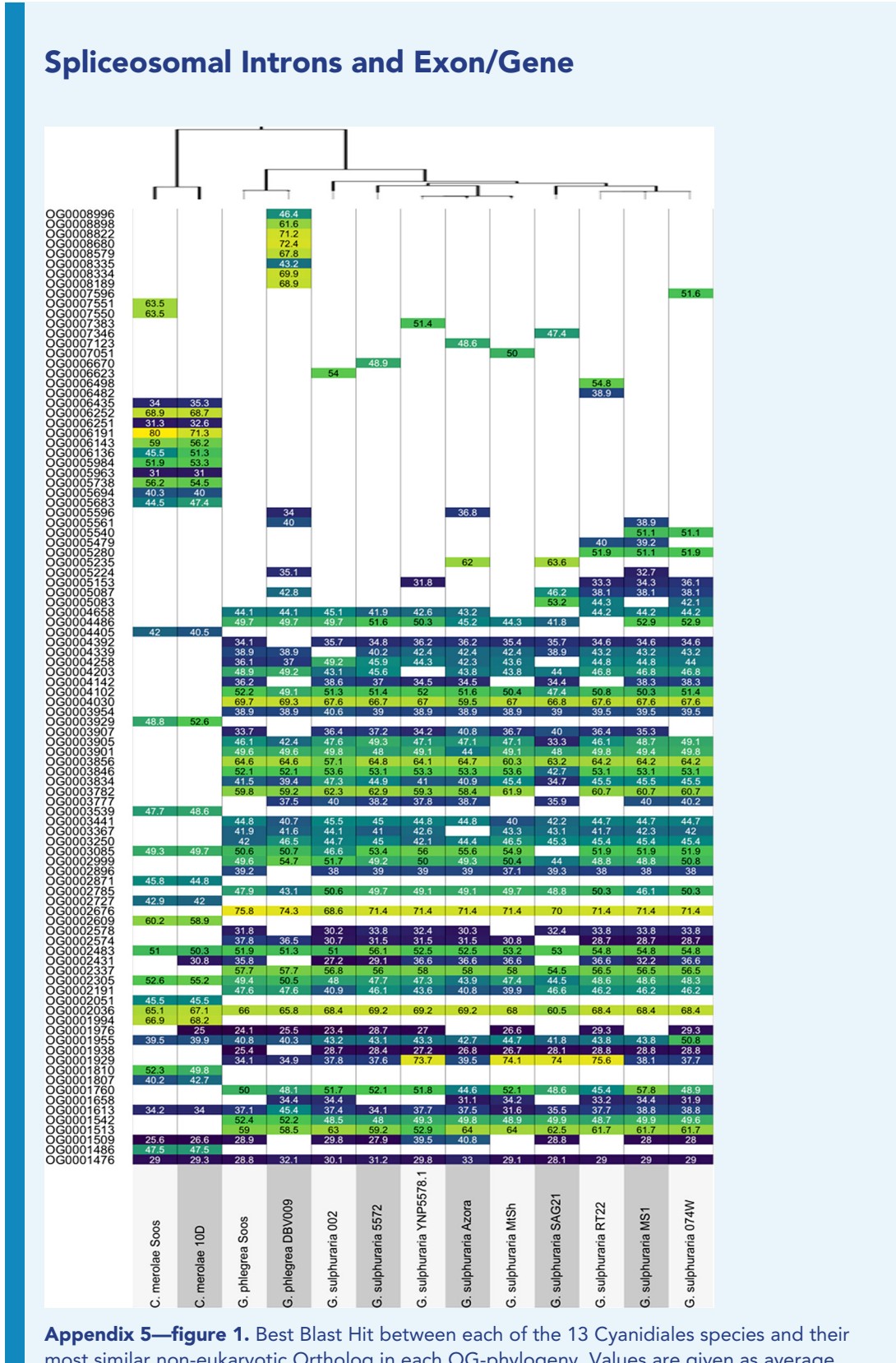

**Appendix 5—figure 1.** Best Blast Hit between each of the 13 Cyanidiales species and their most similar non-eukaryotic Ortholog in each OG-phylogeny. Values are given as average

percent protein identity between Cyanidiales and non-eukaryotic ortholog. White boxes represent missing Cyanidiales orthologs.

DOI: https://doi.org/10.7554/eLife.45017.143

