## [Decision Letter]

Thank you for submitting your article "The genomes of polyextremophilic Cyanidiales contain 1% horizontally transferred genes with diverse adaptive functions" for consideration by *eLife*. Your article has been reviewed by three peer reviewers, and the evaluation has been overseen by a Reviewing Editor and Diethard Tautz as the Senior Editor. The following individual involved in review of your submission has agreed to reveal their identity: Gregory Fournier (Reviewer #2).

The reviewers have discussed the reviews with one another and the Reviewing Editor has drafted this decision to help you prepare a revised submission.

You re-evaluate claims of widespread HGT in red algal lineages, in particular those found within extremophilic environments, the Cyanidiales. This work is largely in response to both previous published claims of extensive HGT in eukaryal genomes, as well as alternative claims that prokoaryote-eukaryote HGT is extremely rare, with most instances being due to contamination or other systematic biases in detection. You address these concerns by both expanding the genome sampling of red algal species, and by applying rigorous phylogenetic and statistical tests to detect HGT events. You further conclude that this limited, but clear signal of HGT within red algal lineages argues against there being a eukaryal "pan-genome" in the same way that pan-genomes likely exist within prokaryotes. The work is relevant to these debates, and the careful evaluation and examination of results, including G+C content bias, intron-exon bias, transcriptomic analysis, evolutionary distance, and manual inspection of inferred gene tree phylogenies all speak to this work being a valuable addition to the field. The testing of intron bias is especially interesting, and a powerful observation that matches prediction. You deftly point out the non-triviality of these types of analysis, for example, the difficulty of assigning HGT to a recipient group, when loss may have occurred in other clades following a more ancestral transfer. The use of PID to discriminate between HGT and DL scenarios is also convincing and intuitive.

A number of important corrections are necessary, as described below:

1) The tree in Figure 2 is mis-rooted. The root should be between the green algal clade and the red algal clade, rather than within the green algae. Also, the annotated dates for splits within this tree are potentially misleading; the ages of these groups remain highly uncertain, and should not be reported without the age ranges of uncertainty included, as given in the cited references.

2) "A core prerequisite of the HGT theory (and cumulative effects) is that horizontally acquired genes have different structural characteristics when compared to native genes". While this is a correct observation, it is not a "core prerequisite", as HGT between lineages with similar G+C contents and codon usages would still occur and be detectable by phylogenetic analysis. It simply is the case that transferred genes retain the composition of their donor lineage, for a time.

3) HGT specification and patterns of HGT acquisition. As illustrated by Figure 5, the authors describe how HGT events and putative HGT donors are observed within gene trees. These gene trees are an important result, and, in addition to the compiled table provided in the main text, an expanded version of Figure 5 should be included in the supplementary information that includes all putative HGTs. The total number of HGT identified by the study is small enough for this to be feasible.

4) Potential HGT donors share the same habitats with Cyanidiales. "To identify the potential sources of HGT, we counted the frequency at which any non-eukaryotic species shared monophyly with Cyanidiales.". The characterization of "donor" in this section and in the corresponding Figure 2 is incorrect. Sister groups to HGT recipient groups do constitute a monophyletic clade in a gene tree indicative of HGT, but they do not, on their own, identify the donor lineage, donor group, or the likely environment of the donor. Both the bacterial and eukaryal sister groups could have been recipients of HGT from an additional unsampled or extinct donor lineage, or the gene could have been distributed by stepwise HGT between the sister clades after a subsequent acquisition. The only way to clearly identify a donor group for an HGT is if the gene tree "nests" the HGT recipient group within a paraphyletic donor clade. This is not clearly the case in any of the example HGT gene trees shown in Figure 5. For example, in Figure 5B, *Galdieria* is sister to Actinobacteria, with representatives of Firmicutes as the outgroup. Since we do not have an established species-tree relationship uniting Actinobacteria and these firmicutes, we cannot determine if the Actinobacteria lineage transferred the gene to the *Galdieria* ancestor, or if both received the gene from a more ancient relative of the firmicutes, or another lineage not represented. Conversely, in Figure 5E, *Galdieria* nests within a clade of proteobacterial taxa. If these together represent a species tree clade of Proteobacteria, *Galdieria* can then be inferred to have been transferred from this group, and the ecology of this donor clade will constrain the ecology of the transfer event.

This also complicates the inference of a shared ecology between the "donor" and recipient groups. If there is a strong correlation with bacterial groups close to the HGT recipients, the correct inference to be drawn is that the HGT occurred AFTER the recipient lineage had become established in this environment. However, this need not be the case. HGT may occur from a donor lineage that does not share ecology with Cyanidiales today, if the HGT was ancient enough that the ancestor of the recipient group (Cyanidiales) had yet to evolve to become an extremophile. Either way, these inferences are valid independent of the ability to identify the specific HGT donor lineage within a bacterial species tree. The authors should therefore provide some additional nuance to the discussion of these results, and present the data in Table 2 to not reflect "potential HGT donors" in the absence of the polarization of an HGT from donor clade to recipient clade. A more agnostic "Natural habitat of prokaryotes with closely related orthologs" would be most accurate.

5) In the absence of detectable cumulative effects – which should definitely exist given that HGT appears to be real – the dynamics of gain and loss should be studied in more detail, first of all by producing heatmaps analogous to Figure 2 but instead showing (a) sharing of putative HGT OGs and (b) sequence identity amongst putatively shared HGT OGs. One hypothesis for both why few close matches are found is that the sequences undergo accelerated evolution upon arrival and therefore it should be interesting to compare the rate of sequence evolution between the core genome and these elements.

6) It would be interesting to extend the analysis separately to singletons, given that many of them can be presumably verified as real based on all the other criteria that the authors use and this should allow identification of recently acquired genes, some of which may show higher homology to prokaryotic homologues. If a set of high-quality singletons could be developed, this would allow much more to be said about the overall dynamic principles involved in gene gain and loss.

7) Sequence alignment and tree files must be included for all species and gene tree analyses within the manuscript, and made publicly available as an online resource, in addition to those genome and annotation files already identified as being deposited.

The following points should be given consideration

Accelerated evolution could be studied in more detail by looking at patterns of homology within genes, to see if particular trends could be found, e.g. in terms of particular biases that reflect adaptation to eukaryotic context.

Differential transcription. You show statistically significant differential expression of HGT genes under temperature changes. However, the test of differential expression they perform for all analyses compares HGT genes en masse to the overall expression of the full transcriptome (that is, non HGT genes). If we understand this method correctly, there is a slight flaw, in that HGT genes tend to be operational genes, which are more likely to be differentially expressed under different conditions. However, many genes in the transcriptome are housekeeping genes that would not be expected to be as responsive to differential expression. In order to account for this bias, differential expression of HGT genes should be compared to other genes only across major functional categories, to show that HGT has higher or lower expression than expected with respect to expected differential expression observed across gene functional categories. This will more clearly isolate the HGT-dependent signal in the data.

---

## [Author Response]

A number of important corrections are necessary, as described below:1) The tree in Figure 2 is mis-rooted. The root should be between the green algal clade and the red algal clade, rather than within the green algae. Also, the annotated dates for splits within this tree are potentially misleading; the ages of these groups remain highly uncertain, and should not be reported without the age ranges of uncertainty included, as given in the cited references.

Thank you for this comment. The reviewer is right, the tree was not properly rooted. This has been fixed in the revised version of Figure 2. Further, we have adjusted the age ranges for the splits according to the cited reference, following the reviewer’s suggestion.

2) "A core prerequisite of the HGT theory (and cumulative effects) is that horizontally acquired genes have different structural characteristics when compared to native genes". While this is a correct observation, it is not a "core prerequisite", as HGT between lineages with similar G+C contents and codon usages would still occur and be detectable by phylogenetic analysis. It simply is the case that transferred genes retain the composition of their donor lineage, for a time.

We agree with the reviewer’s point and have rephrased the sentence to address the reviewer’s comment as follows:

“One of the main consequences of HGT is that horizontally acquired genes may have different structural characteristics when compared to native genes (cumulative effects). HGT-derived genes initially retain characteristics of the genome of the donor lineage. Consequently, the passage of time is required (and expected) to erase these differences.”

3) HGT specification and patterns of HGT acquisition. As illustrated by Figure 5, the authors describe how HGT events and putative HGT donors are observed within gene trees. These gene trees are an important result, and, in addition to the compiled table provided in the main text, an expanded version of Figure 5 should be included in the supplementary information that includes all putative HGTs. The total number of HGT identified by the study is small enough for this to be feasible.

Thank you for this suggestion. We have followed the reviewer’s advice and now present the full set of gene trees as Figure 5—figure supplements 1–96). Further, we have also made available this and other data (single gene trees, orthogroup gene trees and alignments, orthogroup fasta-files) for download at < http://porphyra.rutgers.edu/bindex.php>.

4) Potential HGT donors share the same habitats with Cyanidiales. "To identify the potential sources of HGT, we counted the frequency at which any non-eukaryotic species shared monophyly with Cyanidiales.". The characterization of "donor" in this section and in the corresponding Figure 2 is incorrect. Sister groups to HGT recipient groups do constitute a monophyletic clade in a gene tree indicative of HGT, but they do not, on their own, identify the donor lineage, donor group, or the likely environment of the donor. Both the bacterial and eukaryal sister groups could have been recipients of HGT from an additional unsampled or extinct donor lineage, or the gene could have been distributed by stepwise HGT between the sister clades after a subsequent acquisition. The only way to clearly identify a donor group for an HGT is if the gene tree "nests" the HGT recipient group within a paraphyletic donor clade. This is not clearly the case in any of the example HGT gene trees shown in Figure 5. For example, in Figure 5 panel B, Galdieria is sister to Actinobacteria, with representatives of Firmicutes as the outgroup. Since we do not have an established species-tree relationship uniting Actinobacteria and these firmicutes, we cannot determine if the Actinobacteria lineage transferred the gene to the Galdieria ancestor, or if both received the gene from a more ancient relative of the firmicutes, or another lineage not represented. Conversely, in Figure 5E, Galdieria nests within a clade of proteobacterial taxa. If these together represent a species tree clade of Proteobacteria, Galdieria can then be inferred to have been transferred from this group, and the ecology of this donor clade will constrain the ecology of the transfer event.This also complicates the inference of a shared ecology between the "donor" and recipient groups. If there is a strong correlation with bacterial groups close to the HGT recipients, the correct inference to be drawn is that the HGT occurred AFTER the recipient lineage had become established in this environment. However, this need not be the case. HGT may occur from a donor lineage that does not share ecology with Cyanidiales today, if the HGT was ancient enough that the ancestor of the recipient group (Cyanidiales) had yet to evolve to become an extremophile. Either way, these inferences are valid independent of the ability to identify the specific HGT donor lineage within a bacterial species tree. The authors should therefore provide some additional nuance to the discussion of these results, and present the data in Table 2 to not reflect "potential HGT donors" in the absence of the polarization of an HGT from donor clade to recipient clade. A more agnostic "Natural habitat of prokaryotes with closely related orthologs" would be most accurate.

We take the reviewer’s point and have revised the text accordingly. We appreciate the reviewer’s suggestion for a more agnostic presentation of the results and have adopted her/his suggestion as the subheading for this paragraph. We hope that these changes satisfactorily address the reviewer’s point. The revised text reads as follows:

**“**Natural habitat of extant prokaryotes with closely related orthologs

We next set out to explore the natural habitats of extant prokaryotes that harbor the closest orthologs with candidate HGTs in the Cyanidiales. […] *Sulfobacillus thermosulfidooxidans* shares monophyly in 6/96 HGT-derived OGs and is followed in frequency by several species that are either thermophiles, acidophiles, or halophiles and share habitats in common with Cyanidiales (Table 2).”

5) In the absence of detectable cumulative effects – which should definitely exist given that HGT appears to be real – the dynamics of gain and loss should be studied in more detail, first of all by producing heatmaps analogous to figure 2 but instead showing (a) sharing of putative HGT OGs and (b) sequence identity amongst putatively shared HGT OGs. One hypothesis for both why few close matches are found is that the sequences undergo accelerated evolution upon arrival and therefore it should be interesting to compare the rate of sequence evolution between the core genome and these elements.

We thank the reviewer for this suggestion. To address this point, we have generated new figure, Appendix 5—figure 1. Here we show the% Protein Identity between best prokaryotic Blast-hit with each of the Cyanidiales species analyzed in our study. We appreciate the complex nature of sequence evolution after HGT, leading potentially to an accelerated divergence rate due to adaptation to a new genome and to any proteins that interact with the transferred gene product, as is found for symbiont encoded or transferred genes in any long-term interaction that can confound easy interpretation. Therefore, we present these data but did not look at this aspect in detail. We also note that the level of PID is high enough (35-45% in Appendix 5—figure 1) that we will pick up almost all of the candidates using standard BLAST. It is also hard to prove a negative result; i.e., we did not find the HGT because it is too diverged. Building HMMs for each HGT to scan all genomes may address this issue but would, in our opinion constitute a significant effort that focuses on post-HGT gene divergence and goes beyond the goals of this manuscript.

6) It would be interesting to extend the analysis separately to singletons, given that many of them can be presumably verified as real based on all the other criteria that the authors use and this should allow identification of recently acquired genes, some of which may show higher homology to prokaryotic homologues. If a set of high-quality singletons could be developed, this would allow much more to be said about the overall dynamic principles involved in gene gain and loss.

We respectfully disagree with this point. We intentionally avoided the inclusion of singletons because trees with singletons are never as stable as when there are multiple taxa that contain an HGT candidate. We do however take the reviewers point that a set of singletons might be useful resource for future work by others. We hence have generated a set of candidate singletons (see Appendix 5—figure 1) and made it available for download from <http://porphyra.rutgers.edu/bindex.php>.

7) Sequence alignment and tree files must be included for all species and gene tree analyses within the manuscript, and made publicly available as an online resource, in addition to those genome and annotation files already identified as being deposited.

Agreed. We have now made all of the requested data available for download from: http://porphyra.rutgers.edu/bindex.php.

The following points should be given considerationAccelerated evolution could be studied in more detail by looking at patterns of homology within genes, to see if particular trends could be found, e.g. in terms of particular biases that reflect adaptation to eukaryotic context.

We agree that these would be interesting aspects to study. We do however feel that these aspects are beyond the scope of the current study (see comments above).

Differential transcription. You show statistically significant differential expression of HGT genes under temperature changes. However, the test of differential expression they perform for all analyses compares HGT genes en masse to the overall expression of the full transcriptome (that is, non HGT genes). If we understand this method correctly, there is a slight flaw, in that HGT genes tend to be operational genes, which are more likely to be differentially expressed under different conditions. However, many genes in the transcriptome are housekeeping genes that would not be expected to be as responsive to differential expression. In order to account for this bias, differential expression of HGT genes should be compared to other genes only across major functional categories, to show that HGT has higher or lower expression than expected with respect to expected differential expression observed across gene functional categories. This will more clearly isolate the HGT-dependent signal in the data.

We are not convinced that we can “correct” for operational versus housekeeping genes to address the potential bias in the HGT expression data. By using all of the data we think that we best address the ideas that HGT genes are first, expressed, and second, respond to stress. Whether they are more or less responsive than “ancestral” genes seems to us to be a less informative question. Given that the HGT genes are resident in, and have been integrated in host cell metabolism, we expect that they will behave like any other differentially expressed, stress-related gene, and therefore the meta-analysis is probably the best approach. To restate this point in another way, assuming that novel genes need first to be “domesticated” and incorporated into host biology, the observation that many HGT-derived genes are differentially expressed under stress suggests to us that they are in fact true markers of stress.

To address the reviewers’ point, we have shortened the relevant section in the manuscript text.